# Inversions of landslide strength as a proxy for subsurface weathering

Stefano Alberti [1], Ben Leshchinsky [1] ✉, Josh Roering [2], Jonathan Perkins [3] & Michael J. Olsen [4]

Distributions of landslide size are hypothesized to reflect hillslope strength, and consequently weathering patterns. However, the association of weathering and critical zone architecture with mechanical strength properties of parent rock and soil are poorly-constrained. Here we use three-dimensional stability to analyze 7330 landslides in western Oregon to infer combinations of strength - friction angles and cohesion - through analysis of both failed and reconstructed landslide terrain. Under a range of conditions, our results demonstrate that the failure envelope that relates shear strength and normal stress in landslide terrain is nonlinear owing to an exchange in strength with landslide thickness. Despite the variability in material strength at large scales, the observed gradient in proportional cohesive strength with landslide thickness may serve as a proxy for subsurface weathering. We posit that the observed relationships between strength and landslide thickness are associated with the coalescence of zones of low shear strength driven by fractures and weathering, which constitutes a first-order control on the mechanical behavior of underlying soil and rock mass.

Landslide size distributions are fundamental for evaluating erosion patterns, landscape evolution, soil production, transport and sequestration of organic carbon, and hazard prediction[1–7]. Region-specific quantification of landslide size distributions are typically generalized through scaling laws that relate landslide area with volume and frequency, often using deposit or scar areas[8–17]. Despite the prevalence of these scaling laws, their association with landscape- and landslide-specific distributions of mechanical strength properties, which are intrinsically related to parent material and the magnitude of weathering, are poorly constrained. Thus, relationships that define strength at a landscape scale remain elusive. While connections between strength properties and landslide size have been made with a focus on regional weathering patterns[1–4,18–20], determination of hillslope strength is typically obtained through inversions based on limit equilibrium slope stability analyses of given landslides, which (1) presume planar, one- or two-dimensional kinematics of failure (e.g., Culmann's wedge, infinite slope, etc.) that neglect the potentially strong influence of a three-dimensional rupture surface, and (2) assume a fixed friction angle or cohesion (e.g., Bunn et al.[21]). Such approaches are practical considering that soil landslides tend to be shallow and constrained to more granular materials with limited mineral cohesion, and bedrock landslides occur under cohesive-frictional conditions that are related to partial fracturing and weathering of discontinuities[22]. However, these methods cannot isolate unique combinations of frictional and cohesive strength for a given landslide geometry. Thus, prior research has tended towards exploring various combinations of cohesion and friction[3,15,21,23], disturbance and forcing conditions[2,11], censoring and mapping subjectivity[24,25], local geomorphic features[7,14,19,24,26], as they relate to characteristic distributions of landslide size and frequency. As a result, further insight regarding (1) how weathering affects patterns in the distribution of unique mechanical strength properties in soil and bedrock, and (2) how strength distributions manifest as a covariate with landslide thickness remains poorly constrained.

[1]Department of Forest Engineering, Resources and Management, Oregon State University, Corvallis, OR 97331, USA. [2]Department of Earth Sciences, University of Oregon, Eugene, OR 97403, USA. [3]U.S. Geological Survey, Moffett Field, Mountain View, CA 94035, USA. [4]School of Civil and Construction Engineering, Oregon State University, Corvallis, OR 97331, USA. ✉e-mail: ben.leshchinsky@oregonstate.edu

Trends in scaling laws, typically landslide area-volume ($A − V$) and frequency-area relationships, have been associated with specific parent materials (e.g., Larsen et al.[7]) and reflect mean landslide thickness. However, an association with cohesive and frictional strength at depth remains poorly constrained. Landslide area-volume relationships provide an assessment of the volume or depth of moving material independent of frequency and are assumed to follow a power-law, typically in the form $V = \alpha A^{\gamma}$, where $\alpha$ and $\gamma$ represent a scaling intercept and exponent, respectively. Various scaling parameters have been proposed for specific landslide settings and parent materials[6,8,9,16]. The wide array of proposed scaling relationships reflect a variety of definitions (e.g., scar, deposit, both), including for different landslide mechanisms, and movement mechanics−in this study, we use both inferred source and deposit area as metrics for inversion of combined strength properties and scaling of inventoried landslides. Landslide area-frequency relationships define the likelihood of a given landslide size. A peak or "roll-over" is typically observed at a given threshold landslide size, thereafter following a negative power-law with increasing landslide size[8,10,27]. While the universality of parameters that define these scaling laws are not fully agreed upon[7,24], these relations serve as the primary metrics for quantifying the role of weathering and landslide erosion on a variety of earth surface processes.

We posit that there is an association between landslide thickness and the patterns of governing mechanical strength properties of soil, saprolite, and rock, which reflect the rate and magnitude of dominant subsurface weathering processes within a landslide-dense landscape. Through analysis of 7330 mapped landslides in Western Oregon, USA, we apply three-dimensional slope stability analyses to both deposits and reconstructed landslide source geometries to determine unique pairs of friction angle and cohesion. These distributions in strength are then used to create landscape-scale strength envelopes that we compare to mean landslide thickness. Our results demonstrate that under certain conditions, there is a negative correlation between proportional frictional resistance and landslide thickness. These conditions result in a nonlinear failure envelope commonly associated with fractured rock and to some extent, dilative soils. The decrease in absolute and proportional frictional strength with increasing landslide thickness may be directly accommodated by increased cohesive strength associated with partially fractured rock mass conditions, and consequently, depths sufficient to generate shear stresses that cause failure.

## Results

### Landslides in Western Oregon

Steep terrain, strong but infrequent earthquakes, weak geologic conditions (e.g., sedimentary rock), and considerable precipitation result in extensive landsliding in western Oregon, USA[28,29]. We use twelve inventories consisting of 7330 landslides mapped by the Oregon Department of Geology and Mineral Industries with high-resolution topographic data (0.91 m bare earth lidar, Burns and Madin[30]). The landslide inventories are primarily situated on the Oregon Coast, Oregon Coast Range, and the Western Cascade Range and were aggregated for our analysis (Figures SI.1). Landslides in the inventories are classified using the Varnes[31] system, where ~19.8% of features were classified as being comprised of soil and/or debris (herein termed "soil" for the rest of this study; 5.6% as earth/debris translational or rotational slides, 14.2% as earth flows), 44.5% classified as bedrock (36.9% as rock translational or rotational slides, 7.6% as rock flows), and 39.5% were categorized with complex landslide movements. Generally, the large range in landslide areas and volumes for complex landslides suggests material types that include soil, debris and particularly bedrock (larger, thicker landslides), but this is not certain and a well-known limitation of the Varnes system[32]. Summary statistics of landslide classifications are provided in Table SI.1. Mean annual precipitation at the sites range from 1010 to 3350 mm/yr[33] (Fig. SI.2). The sites are all subject to potentially strong seismic activity owing to the proximity of the Cascadia Subduction Zone 30–50 km from the Oregon Coast; modeled peak ground accelerations for a $M_w$9.0 rupture event at the sites range from 0.11 to 0.44 g for landslides at the sites[34] (Fig. SI.3). Geologic conditions vary across the sites, but mapped landslides occur predominantly sedimentary (47%), quaternary deposits (34%) and igneous rocks (18%), with a limited amount of landslides in metamorphic (0.4%) and unclassified (0.6%) units[30].

Using a topography-based rupture surface fitting technique (modified from Bunn et al.[35]) and an inpainting technique for landslide source area reconstruction[36,37], we calculate landslide volume for both mapped deposits and reconstructed source areas, enabling evaluation of respective area-volume and frequency-area relationships (Fig. 1; example of reconstruction in Fig. SI.4; landslide frequency-area and area-volume relationships for all movement mechanisms shown in Fig. SI.5). The technique fits a thin plate spline using the landslide headscarp and surrounding terrain to project an estimated, three-dimensional rupture surface (see Methods).

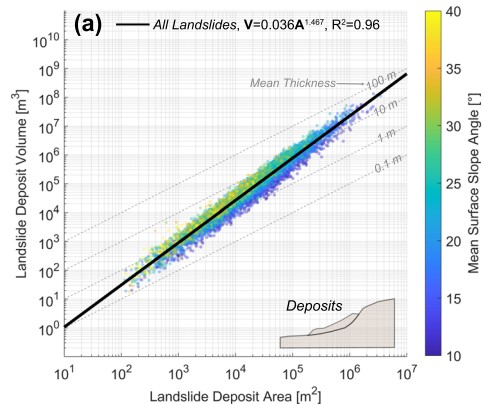
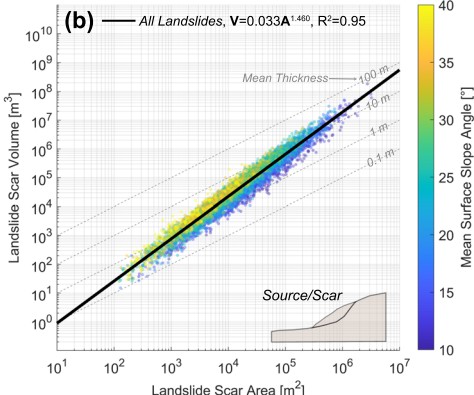

**Fig. 1 | Scaling relationships for aggregated landslide inventories in Western Oregon used in this study ($n$ = 7330 landslides). a** Landslide Area-Volume power law fits are shown for landslides based on existing deposit topography ($V = 0.036A^{1.467}$, $R^2$ = 0.96, RMSE = 0.217). Data points are sorted and colored by mean hillslope inclination, showing an association of steeper slopes within smaller landslide areas. **b** Landslide Area-Volume power law fits are shown for inferred

landslide source areas ($V = 0.033A^{1.460}$, $R^2$ = 0.95, 95% confidence intervals for exponent: 1.458–1.473). Data points are colored and sorted by mean hillslope inclination, showing steeper slope inclinations versus corresponding landslide deposits. Gray, dashed lines represent landslide area-volume relationships at fixed depths for reference. Power law coefficients and goodness-of-fit are shown in the legend.

The inpainting technique used to reconstruct landslide source surface geometry is based on satisfying a set of curvature-preserving differential equations based on surrounding topography—planform, profile and total curvature for deposits and source areas are shown in Figure SI.6. Frequency-area relationships (Fig. SI.5) of both deposits and landslide sources follow a similar inverse-gamma distribution observed for other inventories (e.g., Malamud et al.[10]), but differ between landslides classified as soil/debris (smaller magnitudes) and bedrock (larger magnitudes). The presence of distinct rollover points may reflect censoring of smaller landslides (typically soil), as described in several relevant studies (e.g., Bernard et al.[25], Tanyas et al.[24]); this censoring is likely reflected in some of the relatively modest morphological differences between landslides classified as "soil" or "bedrock" in this study. Source and deposit volumes show similar scaling (Fig. 1), maintaining a power-law exponent of $\gamma \approx 1.46$ and 1.47 and slightly different intercepts of $\alpha \approx 0.033$–0.036, respectively. Scars and deposits may have similar scaling exponents, as discussed in previous studies (i.e., Larsen et al.[7]). There is an evident sorting of data by mean landslide inclination ($\theta$), whereas there is an apparent inverse correlation between landslide inclination and area (Fig. 1a, b) and consequently, thickness. As expected, landslides reconstructed to their source area demonstrate steeper mean inclinations. Power-law fits of area and volume for landslide deposits and sources show different exponents for different movement mechanisms and material classifications (i.e., bedrock landslides with $\gamma \approx 1.40$–1.42; soil landslides $\gamma \approx 1.39$–1.42, complex movement $\gamma \approx 1.46$–1.47, Fig. SI.5). Scaling coefficients for soil rotational/translational landslides are similar to those observed in Oregon and Washington in previous studies (Larsen et al.[7]). However, the differences in area-volume relationship between landslide materials lie within 95% confidence intervals of one another, precluding differences of statistical significance. Median thickness for soil/debris and bedrock translational and rotational landslide sources are 0.92–1.17 m and 2.54–2.59 m, respectively. The observed median soil thicknesses are consistent with field observations (e.g., Heimsath et al.[38]). Nonetheless, potential censoring of smaller landslide sizes[8,24,27,39], and/or a bias in the original classification procedure of the landslide inventory (Burns and Madin[30]) is likely present. As conventional scaling laws are typically based on data either from deposits or scars, we choose source areas (i.e., landslides reconstructed to their unevacuated scars with inferred rupture surfaces) to evaluate relationships between landslide thickness and strength throughout this study.

## Patterns in strength distributions

Using landslide rupture surfaces in combination with existing and reconstructed topography, we use a three-dimensional slope stability analysis[21] to determine unique strength properties for all 7330 landslide features considering generalized geometry (i.e., no constraint on landslide shape). We use Mohr-Coulomb failure criterion to define governing strength:

$$\tau' = \sigma'\tan\phi' + c' \tag{1}$$

where $\tau'$ is effective shear strength at incipient failure, the effective friction angle is $\phi'$, and effective cohesion is $c'$. The dependency of frictional resistance on effective stress ($\sigma' = \sigma - u$, where $\sigma$ is total stress and $u$ is pore water pressure) reflects the strong control of groundwater on frictional strength, where the most drastic temporal changes are expected in the near-surface. The inferred friction angle is determined from the inferred rupture surface and landslide deposit with $c'$ is set to zero (reflecting the broken bonds of cementation and tensile strength, Skempton[40]); thereafter, this friction angle is fixed for the given landslide and used to determine requisite cohesion that yields equilibrium for the corresponding, reconstructed source

topography (see Methods). Some materials may demonstrate shear softening behavior upon sufficient strains. This shear softening can stem from exceedance of dilatative friction, weakening to residual friction, and/or loss of cementation (i.e., mineral cohesion)—we ascribe this strength loss to mineral cohesion as the mean proportional change in back-analyzed friction angle from source to deposit under cohesionless conditions is -33%, which would reflect very dense, dilatative granular materials with low fines content and low confining pressures[41]. Nonetheless, it is not possible to discount frictional shear softening which may be present, particularly in shallow and weathered residual soils (purely frictional behavior is presented in SI.14).

A larger uncertainty in this analysis owes to variability in groundwater flow-fields; thus, various groundwater conditions were evaluated as frictional strength is directly influenced by effective stress conditions associated with saturation (Figs. SI.7, SI.8) and described briefly herein, but "half-saturated" conditions (i.e., the thickness of saturated soil and/or rock for the landslide is half of total landslide thickness on a column-by-column basis, $m = 0.5$) and slope-parallel seepage are used for analyses described in the main text. We choose this condition as a conceptual "average" hydrological control as it is plausible to have perched groundwater in overlying soil or weathered rock stemming from intense rainfall, but saturated conditions are more likely to be persist deeper in bedrock[42]. The sensitivity of varied groundwater conditions is explored in the Figs. SI.11 and SI.12, but observed trends in comparative proportional frictional and cohesive hillslope strength are relatively insensitive to these conditions (e.g., <5% change in proportional strength).

We find that there are trends in strength properties of landslides when compared to mean thickness, and to a modest level, between subjective landslide classifications. Translational landslides, both in rock and soil/debris, tend to show greater friction angles and lower cohesion than their rotational counterparts (Figs. SI.9, SI.10). Bedrock landslides show greater median thickness and cohesion than soil landslides, and although differences of median friction angles when comparing soil/debris and rock classifications are modest, higher percentiles of friction angles of soil/debris landslides are notably larger than their bedrock counterparts (greater by -2–5°). Landslides with complex movement have the largest median thickness and cohesion, the former suggesting a bias towards being composed of bedrock. Median friction angles vary by mechanism for the given conditions but tends to be highest for translational landslides. Differences in median friction angle between landslide classifications are modest (19–25°) and are a strong covariate with mean landslide surface inclination. However, differences in landslide inclination, thickness and strength are usually more pronounced at the extremes of their distributions (e.g., 90th percentile). For example, at the 90th percentile, bedrock landslides exhibit cohesion values three to four times larger than those associated with soil failures at the same percentile. To further compare differences, we calculate statistical significance under the hypothesis that distributions of strength are different using a Wilcoxon rank sum test (Table SI.2). Distributions of cohesion are different with statistical significance ($p$-values < 0.01) between most landslide classifications. Differences in distributions of friction angles show modest statistical differences in many cases. However, mean landslide thickness may serve as a more objective means of constraining strengths as classification of landslide mechanism (Fig. SI.10) is subjective.

We generalize strength properties by creating a failure envelope from all landslide data, where mean effective normal stress ($\sigma'$) and commensurate mean limiting shear strength ($\tau'$) are compared (Fig. 2a). Power-law fits to moving 10th, 50th, and 90th percentiles (1% moving window) demonstrate that the failure envelopes diverge from a linear relationship between effective normal stress and limiting shear strength. That is, while individual landslide strengths are back-analyzed using Mohr-Coulomb criteria (an implicitly linear failure envelope), the failure envelope obtained from analyzing normal

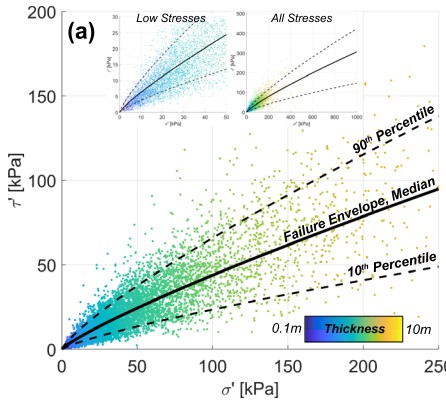
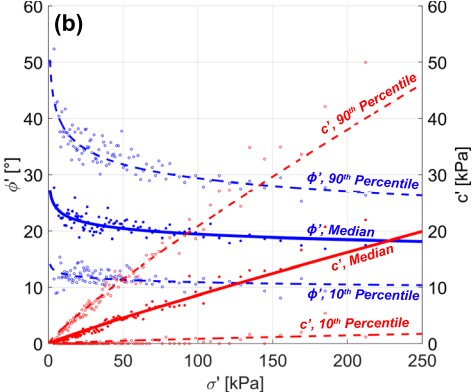

**Fig. 2 | Landslide shear strength (m=0.5 conditions) and associated stresses based on mean landslide thicknesses (n=7330 landslides). a** Effective normal stress ($\sigma'$) versus shear stress ($\tau'$) for landslide inventories colored by mean landslide thickness. Black lines represent moving median and 10 and 90% quantiles. Power-law fits to the moving quantiles for a 1% moving window to represent plausible landslide failure envelopes (median envelope: $\tau'=0.913\sigma'^{0.841}$, $R^2=0.969$,

RMSE = 1.39). The insets show failure envelopes at low and high effective stresses, primarily owing to sensitivity of friction with normal stress. **b** Moving quantiles for both friction angle and cohesion are shown versus normal stress, fitted with power law functions to median strength values for each percentile of binned effective normal stress ($\phi'=27.79\sigma'^{0.079}$, $R^2=0.612$, RMSE = 1.29; $c'=0.13\sigma'^{0.917}$, $R^2=0.969$, RMSE = 0.352), shown with solid and open circles.

stresses and limiting shear strength for all slides in our landslide inventory is nonlinear. A comparison of effective normal stress and both friction angle and cohesion demonstrates the source of the nonlinearity (Fig. 2b). There is an inverse correlation between $\sigma'$ and $\phi'$, and thus, a strong positive correlation between $\sigma'$ and $c'$. The median friction angles associated at very low $\sigma'$ are ~28° and have small cohesion ($\approx0.5$ kPa) for $m=0.5$ conditions. At low normal stresses, 10th and 90th percentiles for friction angle and cohesion are 14° and 42°, and 0.1 and 2 kPa, respectively. These low normal stresses include shallow failures that tend to be associated with steep terrain (Fig. 1a), further decreasing effective normal stress but may also include relatively gentle failures (e.g., small earthflows, weak remobilized landslide debris). On the contrary, deep, presumably bedrock landslides demonstrate decreased frictional resistance, decreased landslide inclinations, and higher normal stresses in comparison to smaller, shallower landslides, but have significantly more cohesion. This trend of decreasing landslide inclination (Fig. 1) with increasing landslide thickness results in two interdependent stress conditions associated with back-analyzed, incipient failure from a landslide source—(1) frictional resistance must be sufficiently reduced as to result in an excess of shear stress that is accommodated by cohesion, and (2) shear stresses must be sufficiently large to overcome the cohesive resistance associated within the rock mass, requiring sufficient landslide thickness. Under these conditions, and particularly with more gentle landslide inclinations (which decreases shear stress), an increasingly large thickness is required to overcome cohesive strength (e.g., Frattini and Crosta[14]). If purely frictional conditions are considered (i.e., no cohesion is considered for back-analysis at the source area), a nonlinear envelope also occurs although friction angles are larger, as expected (Fig. SI.14).

The nonlinearity of the inventory-derived failure envelopes, either for cohesive-frictional or purely frictional conditions is consistent with behavior of soils[41], rockfills[43], and fractured or intact rock[44,45]. Friction may be enhanced in soils (i.e., regolith) under low confining stresses owing to dilation and hardening stemming from grain interlocking[41]. In rockfills, which are effectively a fractured rock mass similar to saprolite, prior research has described a decrease in effective friction angle that spans several orders of magnitude of increasing normal stress[43,44,46] and is a function of density, grain angularity, crushing of asperities, and suppression of rolling resistance under shear. For rock masses, numerous studies have demonstrated a nonlinearity in failure envelopes[44,45,47]. A nonlinear failure envelope exists for a variety of rock mass states (e.g., jointed, fractured, intact), owing to infilling of joints

with weak suffused materials, diminished dilation along roughened joints, and crushing of asperities in joints[45]. The attenuation of frictional strength with normal stress (and indirectly, depth) may serve as a proxy for a variety of mechanisms: namely, (1) brittle rock may mobilize cohesive strength before fully mobilizing friction[48], (2) deep discontinuities are prone to suffusion of clay minerals that exhibit low frictional strength[49] and/or agglomeration of weaknesses[50,51], and (3) large effective stresses suppress frictional behavior[45,52].

Another plausible cause is a decrease in frictional resistance (but not friction angle) in the presence of saturation in bedrock but not in soil (e.g., frictional resistance is affected by pore water pressure, Eq. 1). This sensitivity of our failure envelope is tested for both fixed groundwater depths and a range of saturation ratios (Figs. SI.11 and SI.12), whereas most scenarios still maintain a nonlinear failure envelope of different magnitudes, although fixed, landscape-level groundwater depths of $\approx5-20$ m demonstrate approximately linear behavior and diminished changes in friction angle with effective normal stress. Nonetheless, there is still an increase in cohesion with mean landslide thickness and increasing normal stress. While persistent saturation may occur in bedrock at large depths, intense rainfall is more likely to result in perched groundwater in the near surface (i.e., within soil, saprolite and to some extent, weathered rock; e.g. Salve et al.[42]). As contrasts in hydraulic conductivity at the transition to bedrock may result in a distinct aquifer within regolith or the weathering front of bedrock (e.g., perched groundwater, Lebedeva and Brantley[53]) that is disconnected from seasonal groundwater variations, we assume that saturation ratios are a representative assumption in comparison to constant groundwater depth. Nonetheless, as groundwater conditions in bedrock may be extremely heterogenous (e.g., Lovill et al.[54]) and most of the analyzed landslides are deep-seated, it is possible that unconstrained, variable groundwater conditions over the large area considered in this study may dampen the observed shifts in friction angle with depth. Even so, we attribute the nonlinearity in the failure envelopes to a sensitivity of governing strength with landslide thickness as the observed nonlinearity in failure envelopes is consistent with observations of rock mass behavior (i.e., Hoek and Brown[47], Barton[45]). An approximation of landscape-scale regional flow patterns would enable more confident trends in shear strength as controlled by groundwater flow. However, such models rely on (1) unknown regional-scale boundary conditions and parameters (e.g., hydraulic conductivity), and (2) and homogenous conditions—at this stage, there is insufficient data as to apply such conditions for our analysis. Thus, we evaluate the sensitivity and bounds of groundwater assumptions

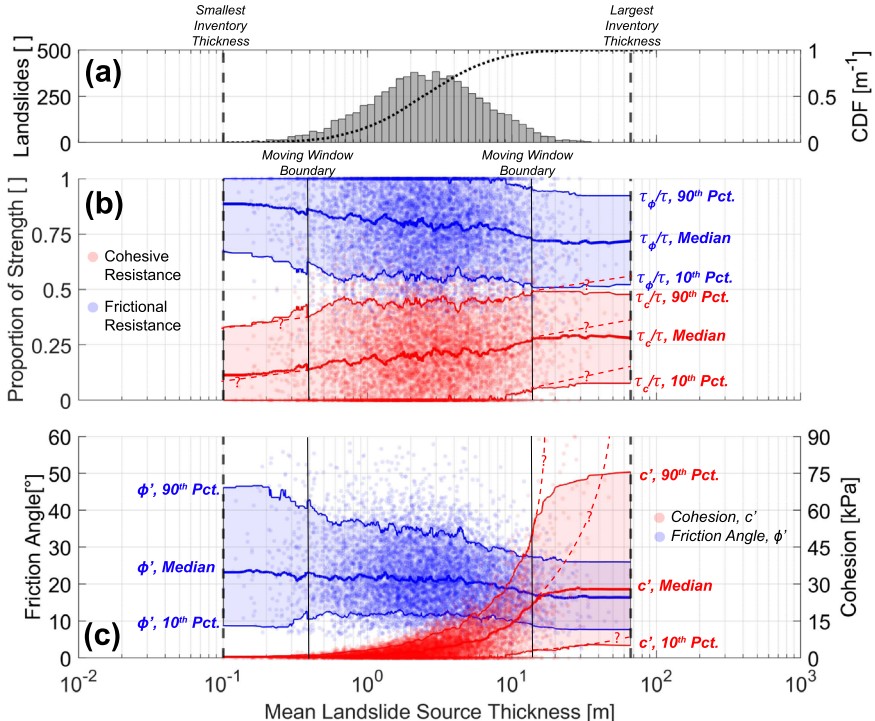

**Fig. 3 | Exchange in strength with landslide size. a** Histogram and cumulative distribution function (CDF) of landslides by mean thickness. Approximately 25% of landslides have a mean thickness of <1 m (predominantly soil), while 70% have mean thicknesses of 1–10 m (predominantly saprolite, weathered bedrock and potentially fresh bedrock). **b** Proportion of total shear strength attributed to cohesion and friction in comparison to landslide thickness. The shaded areas represent the bounds between 10th and 90th moving percentiles (300 point window) of cohesive and frictional resistance, respectively. A gradual transfer from frictional resistance to cohesive resistance is observed with increasing landslide thickness, the largest exchange occurring within weathered bedrock suggestive of a gradient of weathering and strength change. The moving window obfuscates trends at the bounds of the dataset, but hypothetical trajectories are shown with dashed lines for cohesion only (an alternative, binned comparisons are shown in Fig. SI.13). **c** Exchange in friction angle and cohesion with mean landslide thickness.

ranging from dry to fully saturated conditions, observing that the exchange of shear strength magnitude is sensitive to groundwater assumptions; however, the proportional exchange in strength with landslide thickness (described below) is less sensitive to groundwater assumptions. Seepage stresses may cause localized instability—for example, the near-surface (e.g., regolith and saprolite) is more sensitive than deeper flow regimes to changes in pore water pressures owing to rainfall infiltration and higher variations in seepage stresses are observed at shallow depths[55]. At greater depths, changes in effective stress fields induced by seepage are less pronounced and also exist at depths where cohesion may be appreciable, adding a nontrivial length scale to stability (i.e., seepage fields may be less important when cohesion plays a significant role in stability). At more shallow depths (the top few meters) where friction is slightly more dominant, these seepage fields and their influence on effective stresses may be more important, which supports the known sensitivity of shallower landslides to groundwater changes stemming from infiltration[56].

To express the relative contributions of friction and cohesion with landslide thickness, we normalize frictional and cohesive strength to total shear strength (Fig. 3b). A transition in frictional to cohesive resistance is observed with increasing landslide thickness. At depths associated with shallow landslides that typically occur in soil (<1 m), the median contribution of cohesion is ~11–17% of total strength, potentially reflecting factors that may appear as cohesion, such as roots and partial saturation (or the lack thereof). Beyond these depths in materials that would be comprised of saprolite, weathered bedrock, and fresh bedrock[57], the increase in median proportional cohesive strength continues to ~29%. The apparent plateau in this proportional exchange in cohesive strength results from the bounding values of our moving window and the large uncertainty reflects relatively few landslides having mean thicknesses beneath 0.5 m or beyond 10 m, but is

less pronounced using binned data (shown in Fig. SI.13). The known decreasing frequency of landslides with area and particularly depth may be a product of the trajectories of these strength exchanges, where significant cohesion (i.e., more intact rock) may preclude instability. Most observed landslides tend to occur within the range of landslide mean thicknesses where the strength transition is most pronounced, but the role of censoring from mapping cannot be excluded although it is not likely (i.e., larger landslides follow the expected negative power law for area-frequency). Further, there are still uncertainties as to whether landslides with small mean thicknesses are purely in soil. Nonetheless, the observed tradeoff in governing strength with landslide thickness may reflect the diminishing presence of weathering and fracture formation that control mechanical strength properties at depth. The exchange of strength with increasing thickness may suggest that there is a potential depth threshold where weathering diminishes or manifests at much lower rates (e.g., fresh bedrock[57]). The observed transition in strength behavior tends to cover upper bounds of weathered bedrock depths observed in Oregon[58] (~9 m) and northern California[42] (~20 m) although this thickness varies with channel distance and incision.

All models, particularly at this scale, have major uncertainties. Besides groundwater conditions and the influence of roots, other potential causes for the observed trends include different triggering conditions. While no specific triggering events are known for these inventories, we compare strength against indices for seismic or climatic disturbance (Fig. SI.18), using modeled peak ground acceleration for a $M_w$ 9.0 Cascadia Subduction Zone rupture event and mean annual rainfall. We found no clear relations with these potential controls on triggering with the exception of a very modest increase in friction angle and cohesion with mean annual rainfall, suggesting that hydrological drivers could be potentially dominant for these sites, consistent

with observations from Lahusen et al.[29] and Struble et al.[59]. Another uncertainty is the mobility of landslides post-failure, which is often a complex behavior controlled by the rate of deformation, parent material structure and stress path, and porewater pressures[60]. A lack of triggering information on the landslide inventory precludes an association between the mobility of the landslide deposits, strength, and evacuation of source material.

As rupture surfaces are a modeling output and ultimately, a first-order estimate of landslide geometry used to evaluate strength controls on instability, there is no direct means of directly isolating the observed trends in strength to ignore a bias from landslides with extensive evacuation and runout. Extensive runout or complete evacuation of scar could result in deposition that is largely a function of momentum. In such an instance, deposits could be in a more stable than a state of limiting equilibrium. We consider simple criteria by which landslides are evacuated, including the proportional overlapping deposit and modeled source area versus total landslide area (deposit and headscarp). We observe that the landslide inventory herein has a median areal overlap of 58% between source and deposit, which suggests that full evacuation for the landslides used in this analysis may not exert a strong bias associated with long runout landslides. We select a 25% overlap as a lower bound on non-evacuative landslides (approximately the 10th percentile for overlap) as it suggests that the area of deposition is not overly large in comparison to scar area (a consequence of runout). The role of evacuation may control the level of stability of landslide deposits, which are used to determine the friction angle in this study. That is, landslide deposits are treated to be a state of limiting equilibrium associated with continued activity. This is a significant and necessary assumption for the proposed inversions but has a basis in reality as landslide deposits (even relict deposits) are often prone to reactivation and continued instability Temme et al.[61] To explore the potential bias of deposits not being in a state of limiting equilibrium from extensive evacuation or runout, we perform analyses to evaluate the level of stability of deposits considering back-analyzed friction angles from source areas using a factor of safety ($FS$), which for purely frictional materials reduces to $FS = \tan(\phi'_{source})/\tan(\phi'_{deposit})$. Under these conditions, the median $FS$ is 1.26, almost equivalent to specified $FS$ values of designed, engineered slopes ($FS \approx 1.25$–1.3, Fig. SI.16). We also investigate strength trends through application of a $FS$ to deposit friction angles and subsequent recalculation of landslide source cohesion (Figs. SI.16, SI.17). We observe similar exchanges in proportional friction and cohesion, although the exchanges are offset. For some slow-moving failures that tend to be in a persistent residual state and generally be devoid of cohesive strength over long timescales, back-analyzed cohesion estimates may be overestimated. However, the genesis of these failures (which is approximated from this analysis) may have had significant cohesion from reconsolidation or lithification despite their post-failure creeping behavior. For a given landslide, the inversion of strength treats the landslide mass as a homogenous material, which is a simplification with potential significance in layered materials or interfaces between strata (e.g., soil-bedrock boundaries). However, this simplification is common for back-analyses in absence of stratigraphic information[2,26]. We acknowledge that while the three-dimensional slope stability analysis does account for localized topographic stresses within the landslide soil columns, it does not account for other stress controls, such as that of far-field tectonic stresses (e.g., Li and Moon[4]). More robust back-analyses may better incorporate how fractures and stresses stemming from tectonic strains might affect observed strength trends, as well as direct influences of discontinuities and stratigraphy.

## Discussion

The observed nonlinearity in our aggregated landslide failure envelopes, suggests a depth-dependent control on landslide initiation and indirectly, frequency. The observed inverse relationship between landslide inclination and landslide thickness does not preclude the potential for large and deep bedrock landslides to occur in steep terrain of significant relief; rather, it suggests that sufficiently diminished frictional resistance is requisite for the onset of failure in bedrock, at least at broad scales. These diminished frictional strengths may stem from clay-filled discontinuities, groundwater conditions, or suppressed friction with high normal stress, among other factors. In order to have sufficient stress to overcome rock mass strength, there must be sufficient relief and thickness[14,26], large normal stresses as to suppress friction[45], and sufficient weaknesses, fractures, and discontinuities where rock mass structure is reduced significantly. However, we note that cohesion values for bedrock landslides are still modest. This is perhaps expected, as numerous studies have described large discrepancies between rock mass strength at hillslope and laboratory scales[2,3,26], which have commonly been attributed to weakness or discontinuities in bedrock that are not captured when testing small specimens for strength. These discontinuities are often filled with suffused clay minerals[45], and groundwater[4], which in combination with large suppressed dilation from large overburden stresses, act to diminish frictional resistance. It is plausible that coherent, unfailed rock may have appreciably larger strength values than those observed from this inversion (Fig. 4c). However, the strength values described herein represent *incipient failure*; thus, strength values in bedrock are only representative of the weakened strength of bedrock at failure that presumably occurs from weathering. The observed decrease in friction and increase in proportional cohesive resistance with depth serve as a potential proxy for a weathering profile.

Weathering governs the strength of rock and soil and may stem from a variety of processes (e.g., hydrologic, tectonic, and chemical weakening)—it has been recognized as an increasingly important factor in earth surface processes[57,62,63], including landsliding[4]. Recent studies have described the importance—and uncertainty—of the boundary between weathered and "fresh' bedrock stemming from wetting and drying[57], which is considered to be the boundary of the "critical zone." We expect that while soil-bedrock boundary has often been treated as distinct, there is likely a gradient in the breakdown of bedrock with depth—localized and/or stochastic—that are intrinsically a control on the nonlinear relationship between strength and landslide thickness. The exchange of frictional and cohesive strength with mean landslide thickness may serve as a proxy for the bounds of this breakdown, particularly as extensive weathering is known to disaggregate bedrock (cohesive materials with tensile strength) to a more granular matrix (frictional materials). These weathering limits are particularly evident when comparing mean landslide thickness with the exchange of strength (Fig. 3b). When compared with the cumulative distribution of landslide thickness, it is evident that much—but not all—of the exchange of proportional strength occurs in the range of mean thicknesses where landsliding is most frequent (Fig. 4d); however, this trend may be muddled by the apparent censoring of smaller landslides, where strong controls on stability, such as vegetation, may persist. Approximately 33% of the observed exchange between frictional and cohesive strength occurs within depths that may reasonably be considered as soil (<1 m from Fig. 3). The other portion of this strength exchange occurs within depths associated within saprolite, weathered bedrock and "fresh" bedrock. At the larger end of mean landslide thickness, this exchange in strength becomes less clear as landslide frequency decreases. However, the well-known decreasing frequency of very deep landslides and the trajectories of the strength exchange (i.e., the conjectured dashed lines in Fig. 3b) could indicate that at depth, cohesions associated with more intact rock (i.e., less weathered conditions) potentially reflect the transition or gradient to fresh bedrock that has undergone limited subaerial weathering (e.g., wetting and drying) and is governed by slower weathering processes (e.g., tectonic, chemical weathering). Because layering and interfaces

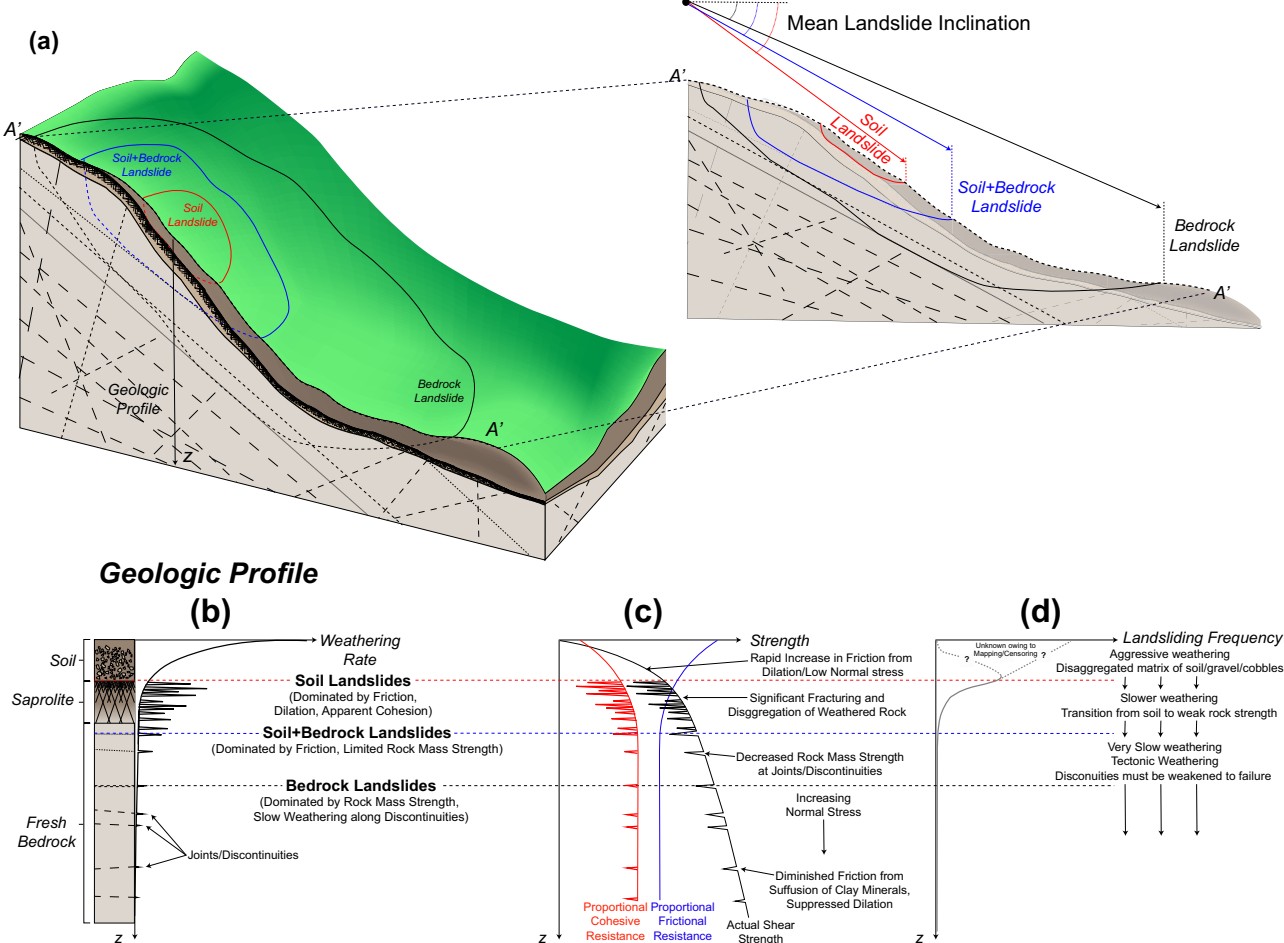

**Fig. 4 | Conceptual relationship between landslide size, strength, and weathering.** Weathering rates and processes control the relationship between landslide inclination, size, frequency, and strength. **a** Three general landslide characteristics are shown in the idealized geologic cross-section: deep, landslides predominantly in bedrock (black lines), landslides of moderate depth that encompass a mixture of soil and bedrock (blue lines), and shallow landslides, typically constrained within the soil (red lines). The landslide boundary and shear surface are shown in bold and dashed lines, respectively. A comparison of landslide inclinations shows (**b**) an idealized transition between steeper soil landslides to more gentle bedrock landslides with increasing depth. Soil landslides are often constrained to localized pockets of steep terrain. While thick bedrock landslides may encompass this same steep terrain, sufficient shear stress is necessary to overcome rock mass strength (predominantly cohesive), envelop localized weaknesses and/or suppress friction, and cause failure, consequently deepening the critical shear surface depth and encompassing more gentle terrain. **c** These conditions can only be accommodated

through diminished frictional resistance through suppressed friction angles (e.g., from large normal stresses, suffusion of clay minerals, polished rock asperities in fractures, lack of mobilized friction due to brittle cohesive strength) and/or elevated levels of saturation. **d** An idealized geologic profile is shown in the bottom of the figure, describing material, behavior, strength, and landslide frequency. Weathering is most rapid at the surface, quickly producing soil from underlying bedrock (rates depending on hillslope curvature and soil depth). This material, which is subject to low confining stresses at finite depths, is primarily frictional, has low cohesion and often dilative, exhibiting friction angles that are greater than those that would be encountered under higher normal stresses. The steep slopes, direct exposure to climatic disturbance, and rapid weathering in this regime results in the most frequent landslides. With increasing depth, landslide frequency diminishes owing to the stochastic distribution of discontinuities and arrested weathering rates deep within bedrock in comparison to the near-surface.

between strata are often assumed to be a control on landsliding, transitions in mechanical properties with depth may be gradual. Weathering beneath the critical zone is much slower than in the near-surface environment[63], but is likely the most rapid along discontinuities. Sufficient weakening of rock mass in discontinuities may take orders of magnitude more time than rapid weathering and production of regolith[7], resulting in the associated infrequency of large, deep bedrock landslides and sufficient loss of cohesive strength for yield (Figs. SI.5, 4d).

In the absence of strengthening from vegetation, shallow landslides in soil and/or bedrock tend to be dominated by frictional strength with small magnitudes of cohesion, if present. This frictional strength is associated with a relatively incoherent matrix of weathered material (Fig. 4b). The elevated friction angles for small and/or steep

landslides—and proportionally shallow depths—reflect a known sensitivity of friction angle to confining pressures (Fig. 2a). With decreasing normal stress, many soils and fractured rock mass may have enhanced friction angles resulting from dilation and interlocking of grains within its matrix[20,43,45]. While observed friction angles are rather low, materials found at lower elevations on hillslopes may have diminished frictional strength from continued weathering, loosening during transport and higher contents of clay minerals[64]—this behavior is reflected by the observation that not all shallow landslides have high friction angles (Fig. 2b). While many shallower landslides (e.g., $D < 1$ m) have limited cohesion, some cohesive strength is often still present. In many instances, cohesion accommodates over 50% of shear strength in some instances of small landslides ($D < 1$ m, Fig. 3). However, this proportion of strength does not reflect large magnitudes of cohesion owing to

small thicknesses and shear stresses. A mean cohesion of 0.83, 4.93 and 25.2 kPa is associated with thicknesses of $D < 1$ m, $1 \leq D < 10$ m, and $D \geq 10$ m, respectively. At shallow landslide depths, these cohesive strengths may result from the influence of lateral root reinforcement. As cohesion is back-analyzed uniformly over the entire rupture surface in this analysis, the influence of possible vegetation is evaluated with back-analyses with a variety of root cohesion at shallow depths (0.5 m) conditions reflecting forest management practices from Schmidt et al.[65], shown in Fig. SI.15. The presence of lateral root cohesion greatly reduces back-analyzed mineral cohesion at modest depths ($D < 1$ m), but much less so for moderate and only modestly so for landslides with significant thickness (i.e., bedrock landslides). This suggests that equivalent cohesion values back-analyzed at larger depths or thicknesses (applied over the entire shear surface) are greater than equivalent root cohesion only applied to the near-surface. At shallow depths (i.e., $D < 1$ m), equivalent cohesion values are at the lower-bound of "root cohesion" values observed in the Oregon Coast Range[65] and may owe to the aforementioned censoring of smaller landslides; however, it has been suggested that underlying assumptions that describe root cohesion values may overestimate the true strength from roots by as much as 75%[66–68], and potentially even more when considering sparse vegetation and the spatial heterogeneity of roots[69].

The presented trends do not contradict that mineral and root cohesion can be large in the soil mantle, and that friction may increase with landslide thickness; rather, they reflect a first-order set of trends that reflect the conditions that represent failure and the associated localized weaknesses that yield failure. For example, studies have made interesting observations that when a threshold slope gradient is exceeded in soil-mantled landscapes, there is less frequent shallow landsliding owing to cohesive boundary forces stemming from vegetation and/or mineral cohesion[70]. It is suggested that beyond these threshold slope gradients, soil erosion in these environments may owe more to creep, which prevents requisite thickening of the soil mantle for landsliding. The censoring of small landslides in these inventories may indeed show a strong cohesive control at very shallow depths. Other studies have described that friction angle may increase with depth, at least in the soil mantle[71]. These strengthening factors may be dominant in many landscapes, including the areas described herein. However, the landslide behaviors presented in this study do not refute those observations; rock mass may have very large friction angles, cohesion stemming from mineral behavior, roots, and capillary stresses may be significant at shallow depths in soil. This study focuses on landslides that have occurred and do not necessarily represent complete landscape strength conditions; rather, the landslides used to construct relationships here may reflect the exception: localized strength conditions that result in landsliding (i.e., low root density or mineral cohesion, weakened and slickensided fractures in soft rock mass with partially intact rock bridges–all of which result in landslide erosion). Still, this analysis does provide perspective as to the exchange of strength within a gradient of weathering at depths reflective of soil and particularly bedrock.

While the observations presented here derive from the climatic, tectonic, and geologic environment of western Oregon, we anticipate that these strength and landslide scale trends, particularly the non-linear strength envelope, are likely relevant to other mountainous settings. We observed different patterns in strength of the near-surface, soil-mantled environment compared to larger depths associated with weathered or fresh bedrock, which reflect a gradient in mechanical properties owing to weathering (i.e., more friction-dominated in saprolite, suppressed friction and more cohesion-dominated in deeper rock mass). For the described inventories, the transition in shear strength with increasing landslide thickness correlates with landslide inclination. Bedrock landslides may still encompass steep terrain, and shallow soil landslides may occur in moderately gentle terrain, but both scenarios show a propensity towards different magnitudes of

friction angles and cohesion. The frequency of landsliding is highest in the shallow, frictional materials and weathered bedrock owing to relatively rapid weathering and commensurately weakened rock mass strength. Deep-seated bedrock landslides are much less frequent as sufficient cohesive weakening takes much longer in comparison to the near-surface environment, although large normal stresses may diminish friction. Thus, we posit that models that rely on strength of subsurface materials should account for the nonlinearity in strength criteria, and that analysis of landslide terrain may serve as a proxy for landslide thickness and weathering.

## Methods

### Landslide inventory
Landslide inventories were selected to represent diverse geologic, topographic, climatic and tectonic environments that have been mapped as part of the Oregon Statewide Landslide Information Database for Oregon (SLIDO v 4.2 updated 10/30/2020; Franczyk et al.[72]). This landslide inventory was mapped by the Oregon Department of Geology and Mineral Industries (DOGAMI) using identification of distinct topography associated with a headscarp, internal scarps, material dislocated downslope (deposits) and a toe based on interpretation of bare earth lidar and its derivatives. A given landslide was mapped using polygons representing the interpreted headscarp and associated deposits. The landslide inventory was classified according to the Varnes[31] classification system[30]. The approximate headscarp height and topographic texture were used by the mappers to define material as rock, debris or earth–classifications consistent with the Varnes classification system. This landslide inventory contains metadata for most mapped landslides, including inferred landslide mechanism, material type, and lithology. More details on the mapping process can be found in the protocols outlined by Burns and Madin[30]. All landslide inventories used in this analysis were mapped using 0.9-m lidar-derived digital elevation models (DEMs) from the Oregon Lidar Consortium (OLC), and certified by the DOGAMI geologists. Based on the mapping protocols, Burns and Madin[30] state that the smallest mappable landslide was 100 m² in planform area.

### Rupture surfaces
We mapped landslide rupture surface geometry using a thin-plate spline technique applied to the boundary of each landslide, modified from the procedure proposed by Bunn et al.[35]. The depth of the rupture surface is dictated by boundary elevations within the landslide headscarp and just outside the deposits which serve as a constraint on a thin plate spline fitting method. Using a regularization value of 0.86[35], this method interpolates elevations across the thin-plate spline by fitting a smooth surface to the control coordinates by minimizing bending energy to solve for the coefficients and a set of weights. Inputs include a landslide polygon and vertices representing the extent of an estimated landslide rupture surface and the corresponding DEM. To adequately reflect various landslide sizes and maintain computational expediency, we use variable resolutions that ensure a minimum of at least 100 cells in the smallest mapped landslide (a sensitivity analysis of back-analyses was observed to be insensitive past this limit) and no smaller than the native DEM resolution (0.91 m). Similar tests showed limited back-analysis sensitivity for the biggest landslides when resolution was <6 m. Based on rupture surface fitting, the smallest landslide deposit area that could be resolved using these techniques was 121 m². These limits on landslide area censor smaller landslides and likely influence observed magnitude-frequency relationships as well as the boundaries of observed strength exchange. Examples of landslide geometries of varying sizes are shown in Fig. SI.4.

### Inpainting
To reconstruct the estimated *original* topography of the landslide prior to failure, we adopted an approach largely used in geomorphic

analyses where hole-filling is required[73,74], termed *inpainting*. The inpainting technique used is based on assumed boundary constraints and associated partial differential equations[36,75]. In this case, we used the Matlab® function *inpaint_nans.m*[36,37] under a simple plate constraint that adheres to the diffusion equation $(\partial z^2/\partial^2 x + \partial z^2/\partial^2 y = 0)$, computed using a finite difference solver over the landslide DEM grid (0.91 m to 6 m length scale). These techniques have shown great potential to preserve and propagate curvature[76], which serves as a means of reconstructing topography based on surrounding, presumably unfailed terrain. Examples of landslide geometries of varying sizes are shown in Fig. SI.4.

## Back-analysis and stability

Back-analyzed slope stability was evaluated using a three-dimensional adaptation of the three-dimensional force equilibrium method of columns approach[21,77] which evaluates the sum of forces in the direction of sliding and transverse to the direction of sliding. We used a modification of this approach with a rotational correction procedure to account for the governing direction of sliding in context the asymmetry and complex shapes of natural landslides and the convergence of force equilibrium conditions while maximizing strength properties. We chose a column-based slope stability model for its straightforward implementation on a gridded digital elevation model, enabling efficient performance at a regional scale and direct incorporation of topographic variation of the digital elevation model. The approach was applied using the rupture surface determined from the thin-plate spline, and first applied to existing landslide topography (i.e., deposits at rest, Fig. SI.4b) to determine the friction angle for each landslide in absence of cohesion (e.g., bonds of cohesion have been broken), excluding the headscarp. To obtain unique combinations of friction angle and cohesion, we then rerun the back analysis on all landslides using (1) the same rupture surface (Fig. SI.4d), (2) the friction angle determined from the existing landslide deposit (Fig. SI.4b) and (3) reconstructed landslide topography (Fig. SI.4c) to determine the cohesion that yields equilibrium. In scenarios where the friction angle for existing topography was smaller than that back-analyzed from reconstructed terrain, we assigned zero cohesion (-14% of all landslides for $m = 0.5$ conditions). Results excluding these values are shown in Fig. SI.19 and show similar trends to those in the narrative. We used 7330 landslides from twelve different inventories that met a variety of criteria necessary for this analysis (Fig. SI.1). Namely, this analysis required inventoried landslides that contained headscarp and deposit polygons and that contained sufficient metadata as to exclude falls, topples, and debris flows, which are mechanisms where the proposed methods would not appropriately capture failure kinematics. Further, we exclude data where estimates of volume change between source and deposits are poor—only considering bulking ranging from 0.5 (50% of source material has been eroded, possible in such landscapes) and 5.5 (90% confidence for bulking, Larsen and Montgomery[7]). As many landslides are likely ancient, sufficient erosion (through repeated failure or other geomorphic processes) could result in diminished post-failure volume and served as the lower bound for the bulking threshold. Large bulking can occur from entrainment of downslope debris. We test any bias from potential change in density from bulking in Fig. SI.19 and similar trends in strength with thickness hold. As evacuation may be a bias for the trends (i.e., low friction angles for long runout landslides), we place a minimum areal overlap threshold for deposit and source area versus total area (deposit and headscarp are combined) of 25% to reflect the lower bounds of non-evacuative landslides. This approach was deemed reasonable as most landslides did not exhibit complete evacuation based on areal extents (median areal overlap of deposit and source proportional to total area was 58%). To ensure that no sampling bias is introduced, we evaluate the aforementioned strength relationships without sampling thresholds (Fig. SI.20), which show similar trends to those already presented.

## Landslide stresses

We use mean landslide thickness ($D$, m), mean reconstructed landslide inclination ($\theta$, °), proportionally saturated thickness ($m$, dimensionless), mass density of soil/rock ($\rho$, 2,040 kg/m³), mass density of water ($\rho_w$, 1,000 kg/m³), gravity ($g$, 9.81 m/s²), and back-analyzed landslide effective cohesion ($c'$, kPa) and effective friction angle ($\phi'$, °) to determine effective normal and shear strength for failure envelopes. As normal and shear stresses may vary throughout the landslide slip surface, for comparative purposes we simplify these stresses into scalar values using mean landslide thickness for comparison on a landscape scale using Mohr-Coulomb relationships based on mean basal stresses. The effective strength parameters are representative of drained conditions (i.e., loading is slow and rate-dependent excess pore pressures are not appreciable, effective stress conditions are maintained). For simplification, scalar representation of landslide effective normal stress ($\sigma'$) is treated as:

$$\sigma' = (\rho g D - \rho_w g D m)\cos\theta \qquad (2)$$

accounting for slope-parallel seepage (per Reid[78]) and pore pressures are applied as a pressure within each landslide column as a function of saturated thickness and the inclination of the surface of each cell. Commensurate effective shear strength ($\tau'$) was calculated based on Mohr-Coulomb criteria as shown in Eq. (1). Scattered data of effective normal and shear stresses from all the analyzed landslides are used to create fits for failure envelopes as shown in Fig. 2a and Figures in the SI.

## Data availability

The data generated in this study have been deposited in a GitHub repository (https://github.com/benalesh/Landslide-Strength). The complete SLIDO inventory is publicly-available through DOGAMI (oregongeology.org/slido/data.htm), while lidar data is available through the Oregon Consortium Lidar (oregongeology.org/lidar/).

## Code availability

Data analysis and processing were conducted using the commercial software MATLAB and its associated functions. The various scripts used for data analysis are available from the corresponding author upon request.

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

## Acknowledgements
B.L. and M.J.O. acknowledge support from the Oregon Department of Transportation from SPR807 and SPR808 and National Science Foundation Grant CMMI-2050047. Mark Reid provided constructive comments on earlier versions of this work. Any use of trade, firm, or product names is for descriptive purposes only and does not imply endorsement by the U.S. Government.

## Author contributions
S.A., B.L., J.R., J.P., and M.O. contributed to writing and interpretation of results. S.A. and B.L. developed scripts, compiled data, ran analyses, and developed the study concept.

## Competing interests
The authors declare no competing interests.

## Additional information

 **11**

