## [Peer Review File · Nature Communications]

REVIEWER COMMENTS

Reviewer #1 (Remarks to the Author):

Review: Distributions of Landslide Size Controlled by Patterns in Hillslope Strength

By: Stefano Alberti, Ben Leshchinsky, Josh Roering, Jonathan Perkins & Michael Olsen

The paper is a novel geomorphic analysis of landslides in western Oregon. The authors have a unique detailed database of landslides with accurately knowledge of surface geodetical information. The authors use over 8000 landslides from the database to unravel geomorphic relationships between slope inclination and area/depth/volume of landslides, but also with the hillslope subsurface strength characteristics. Hereto they creatively use 3D slope stability back analysis to assess the effective geomechanical properties, i.e. cohesion and angle of internal friction values.

Their findings are convincing but also a bit surprising to me; deeper landslides have a larger contribution of cohesive strength over frictional strength. Based on a simple look at the Mohr-Coulomb equation, shallow landslides are more associated with their cohesion values (constant in the formula) and friction dominates when the landslide depth increases (stress * tan(phi) dominates). The authors continue to discuss in-depth possible rationale for the observed relative increase in cohesive resistance over friction resistance, linking the results to weathering regimes.

I applaud the authors for the work on their landslide database, terrain reconstruction and 3D back analysis. The data driven analysis are all excellently done and great to have the data shared. Also the inversely determined strength characteristics are clear. The work presented in this paper is innovative, timely and an impressive amount of work. I found the paper concise, information dense and well written and structured. To me it is interesting for a larger audience. I recommend publication but I have one main clarification to ask.

My main concern relates the analysis of the groundwater effect.

Pore pressure ratio (L159-161; L303 – 310), or hydrology of the landslide. In their analysis the authors study the influence of pore pressure on their findings by a sensitivity analysis of the pore pressure – soil thickness ratio (m) (results in Supplement SI6). They conclude that the range of groundwater conditions has limited effect on their findings of relative contribution cohesive – friction resistance as function of landslide thickness. However, it is a constant ratio independent of landslide thickness and I am not sure if that is a realistic assumption. Assuming similar 'perched GW' to fill up the hillslope equally in a 1m soil landslide and a 10m bedrock landslide seems debatable. With $m=0.5$ this means groundwater is 50cm below surface in the 1m thick example and 5m below surface in the bedrock example. I think (but please proof me wrong) that such low pore water pressure results in a relative decrease in estimated friction resistance. However, it could equally be that deep-seated landslides need larger saturation degree. So a 1m landslide can be triggered with $m=0.5$ and a deep-seated landslide with $m=0.8$ for example (GW 2 m below surface). High degrees of saturation are known to be required for deep-seated landslides. It also coincides nicely with the finding of the authors that deep-seated landslides typically occur on gentler slopes. And maybe steeper slopes show less pore pressure build up as there is an important downslope gradient

draining the slope whereas gentler slopes see relatively higher GW tables also having reduced drainage.

I suggest the authors look at the effect of increasingly larger degree of saturation with increasing landslide thickness. I hypothesize that this will (significantly?) dampen the effect on cohesive – friction resistance exchange with increasing landslide thickness. Even if the effect is modest, it would be good to add it in the discussion of the results.

Minor points:

- Equation 1 and Figure 5c: If looking at Mohr-Coulomb, should the limit not be that the proportion cohesive resistance should be 100% if the landslide thickness reaches 0?
- Can you elaborate on your definition of effective angle of internal friction and effective cohesion? As this also comes back in your discussion on the weathering hypothesis.
- The authors state a few times that the dominant soil geometry is approx.. 2 m soil and then saprolite/bedrock. If this is so dominant, why not doing the 3D geomechanical back analysis with this geometry? Can the authors quantify and discuss what would be the effect of using a two-layer geometry ?
- Is it possible to also look at the location of landslides on a slope: in the convex or concave part of the hillslope. Is there information on that?
- The same holds for possible diverging / converging slope geometries. Any information on that?
- L346-347: I do not entirely understand this argument: even for hillslope scale the effective cohesion values are modest. Can you compare to other reported hillslope scale cohesion values in the Oregon regions?
- L405: small or negligible magnitude of cohesion in shallow landslides. That is a strong statement, knowing that in your analysis also apparent root cohesion of small and larger vegetation is included in the effective cohesion value and generally seen as very large and important. Moreover, a few kPa on cohesion equals several decimetres of soil stress..... Maybe do some back of envelope calculations to proof your statement could be useful.

It has been a pleasure to read this paper and I wish the authors success with their work.

Dr Thom Bogaard,

Delft University of Technology, Delft, The Netherlands

Reviewer #2 (Remarks to the Author):

This is a very interesting and potentially important paper though it is currently a little difficult to follow in places. The paper is set up as an explanation for size distributions but I don't think that is really what the paper is about. Instead the authors use a back analysis of thousands of landslides to identify depth dependent changes in friction angle and cohesion and argue that these are connected to bedrock weathering. The large sample back analysis is a spectacular effort and the conclusions about friction angle and cohesion variation with depth are extremely surprising and thus very important. On the other hand, I don't find the connections to landslide size convincing at all.

My major concern with the paper is that the friction angle estimates rely on a method that generates spurious correlations with landslide inclination. I did not find sufficient information in the paper to fully understand the method for friction angle estimation and my comments below focus on this aspect of the work. Concerns around this aspect of the method and the consequent results, which have opposite trends to those from existing studies make me extremely nervous that there is an error here that propagates throughout the results.

Major comments

MC1: The paper needs a title that is more in line with the paper's claims and a more coherent argument to support these main claims. The paper isn't really about size it is about rock/soil strength back calculated from observed landslides. That doesn't make it unimportant! It is still a very interesting paper and its findings are surprising and therefore important. The claims are clearly outlined in the final quarter of the abstract but I get lost in the text. I need more help following your logic if I am to be convinced that the claims of your abstract are justified by your data.

MC2: The paper needs more detail on and a stronger justification for the method of estimating friction angle. The method used to calculate unique values for both friction angle and cohesion involves two back calculation procedures for each landslide. Both involve interpolating surfaces from sparse marginal data. My main concern with the paper is the first back calculation that is used to calculate friction angle under the assumption that deposits within the scar are in limiting equilibrium (along the pre-existing failure mechanism) and are cohesionless. My concerns are: 1) that for many of the landslides I have seen (in person or in pictures) deposits are either patchy or absent in the scar and I don't know how you can apply your procedure in either of these cases; 2) a significant fraction of the pre-existing failure surface will now be the ground surface and shouldn't be included in the analysis (I guess it isn't but this wasn't clear to me from the paper); 3) the deposits that remain in the scar are not necessarily in limiting equilibrium, they can't be unstable but we have no indication of how stable they might be, as a result the first back calculation is a lower limit on friction angle but one that is also very strongly influenced by the scar topography (i.e. deposit angles do not reflect their friction angle but simply the angle of the scar within which they find themselves). This final point is the one I'm most concerned about because it would introduce a spurious correlation between landslide slope and friction angle. This would propagate through the known (and observed here) negative correlation between landslide slope and landslide depth to generate the relationship that you observe were friction declines with depth.

To me the depth dependent reduction in friction angle is the most important finding of the paper and the basis for the other ideas that follow so it is really important that your method to estimate friction angle is clear and that you deal with these concerns around possible errors in the estimate. The magnitude of the reduction in friction angle with depth is really surprising and since the

reduction is most rapid at shallow depths it is shallow landslides that should demonstrate this behaviour most clearly.

For friction (L390-91): 'Approximately half of the observed exchange between frictional and cohesive strength occurs within depths considered to be soil (<1.9 m)'. From the Figures it looks as though estimated friction angle decreases by about (40 – 30 deg) as mean depth increases from 0.1 to 1 m, which is a huge reduction in friction angle with depth. The only studies I have come across that discuss depth dependent friction angle in soil actually suggest the opposite relationship - i.e. friction angle increases with depth (Lu and Godt, 2008; and later papers by these authors). They argue that porosity decreases with depth (Selby, 1993; Cornforth, 2005) and friction angle decreases with porosity (Rowe, 1969; Marachi et al., 1969; Cornforth, 1973, 2005) thus friction angle should increase with depth (references all from: Lu, N. and Godt, J., 2008. Infinite slope stability under steady unsaturated seepage conditions. *Water Resources Research*, 44(11)).

You find that cohesion increases with depth in the soil (e.g. L417-8, Fig 3b). Again, I do not know of any research suggesting that cohesion should increase with depth in the soil, though I do know of several papers that suggest the opposite, usually on the basis that both root density and root diameter distributions decrease with depth (e.g. Bischetti et al., 2007, Montgomery et al., 2009).

Bischetti, G.B., Chiaradia, E.A., Simonato, T., Speziali, B., Vitali, B., Vullo, P. and Zocco, A., 2007. Root strength and root area ratio of forest species in Lombardy (Northern Italy). In *Eco-and ground bio-engineering: The use of vegetation to improve slope stability* (pp. 31-41). Springer, Dordrecht.

Montgomery, D.R., Schmidt, K.M., Dietrich, W.E. and McKean, J., 2009. Instrumental record of debris flow initiation during natural rainfall: Implications for modeling slope stability. *Journal of Geophysical Research: Earth Surface*, 114(F1).

These differences to the existing literature do not demonstrate that your findings are incorrect. In fact they show how important the findings are if correct. However, because they are so surprising, different and important, the method that underpins them is a particular focus for scrutiny.

You do try to provide some support from the literature for the idea that friction declines with depth but when I followed up these citations it wasn't obvious that their findings supported your claims. You cite Alejano and Alonso (2005) as evidence that soil friction angles decline with depth but their observations are at larger confining stresses >100 kPa and show much smaller reduction in friction angle and much larger friction angle than observed here (67-50 deg across the range 100-1000 kPa). You say (L407) 'reflect a known sensitivity of friction angle to confining pressures' but this needs supporting with citations. You say (L409) 'With decreasing normal stress, many soils may have enhanced friction angles stemming from dilation and interlocking of grains within its matrix (Townsend et al. 2020). However, Townsend et al. don't mention dilation, or interlocking and don't appear to show results in support of the statement that friction angles increase with decreasing normal stress. I couldn't find any reference to depth-dependent friction angles in Bellugi et al. (2021) either (L409-10). You make an argument around colluvial materials on L410-11, but geomorphologists typically count the entire soil mantle as colluvial, particularly in places like the Oregon Coast Range. Are you working from an alternative definition?

In the explanation of your back-calculation method, more information is needed on: 1) how you identify deposits within the scar and define the potential failure surfaces associated with those deposits; and 2) how many scars do not contain identifiable deposits in the scar, and their characteristics (e.g. size, depth, slope).

In the landscapes I have worked in 90% of the landslide scars were empty of deposit. You would be unable to resolve friction angle for any of these slides but if you assumed they were cohesionless that would also be incorrect, roots typically provide considerable additional strength. Perhaps the types of landslides that you are examining here are different to those I have in mind. This would be important to state in your early definition of landslides and in your explanation of what is and is not included in your landslide inventories.

Specific comments

L15: The 1st sentence, is stated as a fact but I think this is really a hypothesis.

L29-30: do gentle inclinations accompanied with 'proportionally lower frictional resistance' guarantee a need for 'compensation from cohesion'. The sentence reads as though one is a logical consequence of the other.

L30: 'requiring' what is it required in order to achieve? Stability?

L29-31: The argument of the sentence appears circular, it starts with landslide depth, associates depth with properties then finishes by saying that these properties result in the observed depth.

L32: not clear which particular 'strength relationships' you mean.

L33: not clear what you mean by 'geomorphic scaling laws'.

L38: 'landslide size distributions are a reflection of weathering patterns at a landscape scale'. This is too strong. It is stated as a fact but there is not consensus on this. Tanyas (2018) gives a nice review of many of the possible explanations.

L38: It would be useful to start by defining what you mean by landslides in this paper what are the limits to your definition in terms of speed, size, and movement mechanics? This will be important when you seek to claim that your landslide inventories are representative.

L38: When you talk about landslide size do you mean scar alone or scar and runout combined?

L58-63: Some researchers continue to argue that censoring is responsible for the rollover.

L67: 'long been associated with...' I think this claim needs citations to support it.

L71-74: I don't think this detail is necessary.

L74-75: This clause is out of place, it relates to the next sentence.

L77-78: How many of the distributions cited here are for landslide scars only?

L84: what does relevant mean in this context?

L85: mapped landslides: what are the mapping criteria and methods used to map these landslides? What is the definition of landslide used by the mappers? What do the shapes outline (scar or scar and runout)? These details will influence your sample of landslides, the particular properties of this sample is very likely to exert a strong influence on your back-calculated strength-depth relationships.

L92-96: This sentence is very difficult to understand it lists multiple interacting components but doesn't explain how they interact. It also relies on results that the reader hasn't yet seen so my suggestion would be to remove it.

L98: I'm not sure what you mean by 'adverse geologic conditions'.

L104: How were landslides classified as soil or bedrock?

L108: More information is needed on these inventories. What was their definition of a landslide? How small were the smallest landslides they could resolve? How did they deal with scar and runout? How do you deal with rockslide and rockfall? Bernard et al. (2021) show that if these rockfalls may be severely undersampled in existing inventories and can considerably alter the size distribution.

Bernard, T.G., Lague, D. and Steer, P., 2021. Beyond 2D landslide inventories and their rollover: synoptic 3D inventories and volume from repeat lidar data. *Earth Surface Dynamics*, 9(4), pp.1013-1044.

Fig 1b: There are surprisingly few small landslides in this inventory. How much censoring do you expect in your mapping and at what size?

L123: 'rupture surface fitting' introduces a dependence on both resolution and precision of topographic data. How does this censor your landslide distribution?

L126: the 'example of reconstruction' is useful, you should also include an example for one of the smallest landslides in your inventory to demonstrate (qualitatively) that it is appropriate to apply these techniques at this scale.

L142: I don't understand what 'sorted' means in this context. Do you mean the distributions differ between soil and bedrock landslides? Or are you being more specific e.g. they are shifted?

L150: How confident can you be that you have captured the geometry of 'incipient failure' for your landslides? Is there a size (and therefore depth) dependence to this confidence? How does uncertainty in incipient failure propagate into your results?

L164: 'Soil landslides demonstrate median thicknesses of 1.9-2.1m': These are extremely thick for soil landslides suggesting under-sampling of smaller shallower landslides in the inventory.

L166: 'friction angles are a strong covariate with mean landslide inclination'. Isn't this unavoidable given the method that you use to estimate friction angle? You should certainly including scatter plots of friction angle and mean landslide inclination.

L203-204: (e.g. "angle of repose", Alejano and Alonso 2005). Alejano and Alonso 2005 don't discuss 'angle of repose'.

L214-215: 'supporting arguments that': citations needed for the arguments you refer to here.

L248-254: 'decreasing landslide inclination with larger landslide area results in two interdependent stress requirements for failure': An increased thickness required to overcome cohesive strength makes sense but it is not clear why this follows from the preceding requirements: the first requirement (frictional resistance reducing so that cohesion can increase) isn't a requirement it is a restatement of your finding; the second (shear stress being sufficient large to overcome cohesion) is guaranteed from back-calculation.

L298: 'The attenuation of frictional strength with landside area and consequently depth...': none of the explanations that follow involve area, all involve depth; while you can use area, depth or volume as your independent variable, only depth is a candidate for mechanistic explanations. In your method area controls depth (through spline and inpainting) but you don't explain the physical connection between area and depth.

L330-331: I suggest removing landslide relief it is clearly a result of the increased length scale over which relief is being measured as area increases (demonstrated by the reduction in inclination with area).

L421: It is not clear what you mean by 'residence times' here.

L427-428: Don't these 'other strengthening factors, such as vegetation and partial saturation' provide strength as cohesion?

L456: It is not clear what you mean by flank polygons, it would help to label this on Fig S14

L457: How were the landslides identified in lidar? Can explain the method or provide a reference?

L457: What metadata were used to exclude landslide complexes?

L460: 'mapped using 0.9-meter lidar-derived DEMs': What censoring do you expect in your inventory on this basis?

L470: What is the minimum landslide area for which the method can resolve a rupture surface geometry? Did you test this for some of your smallest landslides?

L481: What length scale is curvature calculated over? I guess uncertainty in this length parameter has little impact on your findings but it should probably be recognised.

L481: Why is preserving and propagating curvature 'key for infilling landslide scars'?

L491-492: It is not clear to me that 'existing landslide topography' and 'deposits are rest' are equivalent. Two examples that I have seen at many landslides are: 1) The landslide topography will include part of the scar that is now free of deposit; 2) the deposit within the scar is discontinuous and patchy; 3) the deposit extends far beyond the scar (e.g. due to avalanche or flow-like runout). How do you handle these situations?

L493: 'most landslides did not exhibit complete evacuation': What was your estimate for the proportion of landslides that did exhibit complete evacuation?

L502: 'less than 10%': give the nearest integer value.

L508: Equation 2: What assumptions are required to construct these equations for normal and shear stress? I was expecting a $\cos(\theta)$ in the pore pressure term. Are you assuming hydrostatic rather than slope parallel seepage? If so, why make that choice?

L512: I'm not clear what you mean by 'arbitrary landslide inclinations' here.

Reviewer #3 (Remarks to the Author):

This paper by Alberti et al., "Distributions of Landslide Size Controlled by Patterns in Hillslope Strength" examined the influence of the hillslope strength on the landslide size scaling relation with volume and frequency. They analyzed ~ 8,000 landslides, quantified strength metric – friction angle and cohesion in western Oregon, and showed how the strength estimation change with landslide

size. This paper presented the applications of a novel 3D slope stability analysis on large actual dataset of landslides, and provide impressive amount of work on statistical tests and analysis. The 3D slope stability methods are based on the previously published studies but are applied to new large dataset in western Oregon. They show that friction and cohesion dominate soil and bedrock landslides, respectively, and hillslope strength affected by weathering patterns affect the geomorphic scaling law, especially the landslide frequency-area relationship.

The paper is well written and provides important results on demonstrating the weathering controls landslide scaling factors. However, some parts need to be improved by clarifying and presenting slightly differently before publication. The authors can improve the manuscript by 1) analyzing strength results separated for soil and bedrock landslides and 2) revising the discussion to acknowledge limitations of the methods and clarify the influence of "weathering." I recommend considering the publication after the significant revision of the paper. Below are the important comments on the paper.

First, the authors may consider currently presenting the strength estimation results separately for soil and bedrock landslides. I am wondering whether the relationship between strength and landslide sizes is somewhat influenced by mixing between abundances of soil vs. bedrock landslides for a given landslide size range. Smaller landslides are more soil landslides, and bigger landslides are more bedrock landslides. The increase of landslide sizes tends to have more bedrock landslides, so just higher cohesion values. The smooth transition in Figure 3 may not reflect the bedrock weathering degree difference but may show the relative mixing trends between those two landslide types. If bedrock weathering degree indeed influences the scaling factor, this trend should be evident when analyzed only for bedrock landslides.

Typically, soil landslides tend to occur in the soil-bedrock boundary, so the failure planes may likely represent soil depth variation. The soil depths are probably influenced by in-situ weathering but also can be affected by curvature, soil transport and thus landscape position, etc. If you separate the analysis, you may see some trends differ in bedrock and soil landslides. For example, the influence of cohesion in soil landslides is more significant with shallower depth. Because there may be different potential controls on soil and bedrock landslides, I suggest dividing the results for those and clearly showing the existence of trends in soil and bedrock landslides separately. Adding the figures for only soil vs only bedrock landslides in the supplementary figure may be sufficient.

Second, the discussion can be improved by a clear explanation of 'weathering' and the limitation of the methods. The discussion regarding weathering is somewhat unclear to me. Does "weathering regimes" imply dichotomy boundaries between soil vs. bedrock, or in situ, continuous physical or chemical transition from bedrock, saprolite, to the soil. Those are different, and current analysis based on all landslides together may be hard to distinguish those two. Clarification on the influence 'weathering' – boundary, degree, or others, based on separated analysis (see the first comment) will be helpful. In addition, limitations of methods can be more clearly explained, and some quantitative comparison on strength estimates can be helpful. Although the authors used novel 3D methods for fitting failure planes, the slope stability analysis is simply based on the weight of the vertical column (Methods). This doesn't consider the effective stress from 3D topography or groundwater flow patterns, and the currently estimated cohesion in bedrock may influence by those other factors. In fact, the estimated cohesion in soil or bedrock are quite lower than the values observed by field measurement. Authors already discussion those potential influences qualitatively, but may consider providing a more quantitative comparison of results. It will be interesting to know whether the magnitudes of estimated cohesion differences (10 – 100 kPa) are similar in magnitudes of 3D effective stress variation from topography or groundwater flow.

Detailed comments are below.

Line 38. Consider adding references to Clarke and Burbank 2010

Clarke, B. A., and D. W. Burbank (2010), Bedrock fracturing, threshold hillslopes, and limits to the magnitude of bedrock landslides, *Earth Planet. Sci. Lett.*, 297(3–4), 577–586, doi:10.1016/j.epsl.2010.07.011.

Line 43. Consider adding references to Hovius et al. (1997)

Hovius, Niels, Colin P. Stark, and Philip A. Allen. "Sediment flux from a mountain belt derived by landslide mapping." *Geology* 25.3 (1997): 231-234.

Line 49. Clarke and Burbank's 2010 and 2011 papers

Line 53. three-dimensional geometry of "failure plane?"

Line 100 high-resolution topography means what resolution?

Lines 219 – 221. The range of measured lateral root cohesion in soil landslides in Oregon varies from 6.8 to 94.3 kPa (primarily measured < 1 m, Schmidt et al. 2001), which seem to be higher than the estimate presented here. Why do you think there are some differences?

Line 225 – 226. This is slight confusion to me. To see whether this trend is due to the in-situ weathering process within bedrock, I think that it will be helpful to see the analysis done only using bedrock landslides, not from combined soil and bedrock landslides. Currently, the results are shown for both soil and bedrock landslides, so it is hard to tell whether the transition is due to variations of soil thickness, the difference in relative abundances of landslides, or the in-situ weathering differences.

Line 306. Are groundwater depths consistent with all surface? In reality, it may be deep under the ridge and shallow in the channel and 3D flows. Can this 3D effect be significant enough to change the trends?

Line 310. Groundwater flow itself can generate differences in effective stress (Iverson and Reid, 1992, Reid and Iverson 1992). May check the magnitude of stress changes can induced by this effect.

Iverson, R. M., and Reid, M. E. (1992), Gravity-driven groundwater flow and slope failure potential: 1. Elastic Effective-Stress Model, *Water Resour. Res.*, 28(3), 925– 938, doi:10.1029/91WR02694.

Reid, M. E., and Iverson, R. M. (1992), Gravity-driven groundwater flow and slope failure potential: 2. Effects of slope morphology, material properties, and hydraulic heterogeneity, *Water Resour. Res.*, 28(3), 939– 950, doi:10.1029/91WR02695.

Line 313: "but more likely simply add noise to the observed trends." Why? Is it due to the stress magnitude being small?

Line 353 delete large

Line 419-420 The cohesion values (10 – 100 ka ranges) in the Roering et al. 2003 and Schmidt et al. 2001 seem quite higher than the presented.

Reviewer #1

Reviewer comments are in blue.

Author response is in black.

Modified text in manuscript or SI is black and italics.

The paper is a novel geomorphic analysis of landslides in western Oregon. The authors have a unique detailed database of landslides with accurately knowledge of surface geodetical information. The authors use over 8000 landslides from the database to unravel geomorphic relationships between slope inclination and area/depth/volume of landslides, but also with the hillslope subsurface strength characteristics. Hereto they creatively use 3D slope stability back analysis to assess the effective geomechanical properties, i.e. cohesion and angle of internal friction values.

Their findings are convincing but also a bit surprising to me; deeper landslide have a larger contribution of cohesive strength over frictional strength. Based on a simple look at the Mohr-Coulomb equation, shallow landslides are more associated with their cohesion values (constant in the formula) and friction dominates when the landslide depth increases (stress * tan(phi) dominates). The authors continue to discuss in-depth possible rationale for the observed relative increase in cohesive resistance over friction resistance, linking the results to weathering regimes. I applaud the authors for the work on their landslide database, terrain reconstruction and 3D back analysis. The data driven analysis are all excellently done and great to have the data shared. Also the inversely determined strength characteristics are clear. The work presented in this paper is innovative, timely and an impressive amount of work. I found the paper concise, information dense and well written and structured. To me it is interesting for a larger audience. I recommend publication but I have one main clarification to ask.

The authors sincerely thank the reviewer for their excellent comments and recommended clarifications. We have found all of this feedback exceptionally helpful and we are very grateful for the reviewer's time and effort.

My main concern relates the analysis of the groundwater effect.

R1.1. Pore pressure ratio (L159-161; L303 – 310), or hydrology of the landslide. In their analysis the authors study the influence of pore pressure on their findings by a sensitivity analysis of the pore pressure – soil thickness ratio (m) (results in Supplement SI6) . They conclude that the range of groundwater conditions has limited effect on their findings of relative contribution cohesive – friction resistance as function of landslide thickness. However, it is a constant ratio independent of landslide thickness and I am not sure if that is a realistic assumption. Assuming similar 'perched GW' to fill up the hillslope equally in a 1m soil landslide and a 10m bedrock landslide seems debatable. With $m=0.5$ this means groundwater is 50cm below surface in the 1m thick example and 5m below surface in the bedrock example. I think (but please proof me wrong) that such low pore water pressure results in a relative decrease in estimated friction resistance. However, it could equally be that deep-seated landslides need larger saturation degree. So a 1m landslide can be triggered with $m=0.5$ and a deep-seated landslide with $m=0.8$ for example (GW 2 m below surface). High degrees of saturation are known to be required for

deep-seated landslides. It also coincides nicely with the finding of the authors that deep-seated landslide typically occur on gentler slopes. And maybe steeper slopes show less pore pressure build up as there is an important downslope gradient draining the slope whereas gentler slopes see relatively higher GW tables also having reduced drainage.

I suggest the authors look at the effect of increasingly larger degree of saturation with increasing landslide thickness. I hypothesize that this will (significantly?) dampen the effect on cohesive – friction resistance exchange with increasing landslide thickness. Even if the effect is modest, it would be good to add it in the discussion of the results.

This is an excellent comment – this exact notion is one that we dwelled upon for quite a while. While there are many uncertainties relating to landslide reconstruction and inventory mapping, the groundwater conditions controlling triggering may be the largest source of uncertainty. To explore the sensitivity of our groundwater assumptions, we performed the analysis on a range of groundwater conditions, ranging from fully-saturated ($m=1$) to dry conditions ($m=0$) as well as with presumed groundwater depths in between ($D_w=1\text{m}, 2\text{m}, 5\text{m}, 10\text{m}, 20\text{m}$), as shown in Figures SI.7, SI.8, SI.11, and SI.12. The reviewer is completely correct that this does dampen the exchange of the magnitude of friction *angle* and cohesion with mean landslide thickness, particularly at groundwater depths of approximately 5-20m (which spans sizeable proportion of inventoried landslides). However, the *exchange of proportional frictional and cohesive resistance* with mean landslide thickness is not as sensitive the groundwater location. While somewhat counterintuitive, this behavior owes to a close association between the length scale of landslide thickness and cohesion. While frictional resistance is influenced by effective stress conditions (i.e. groundwater), the systematic decrease in landslide inclination with landslide size exceeds the sensitivity of frictional shear resistance with increasing groundwater. Cohesion is largely sensitive to the length scale of mean landslide thickness, with or without frictional resistance. This means that while saturated conditions below the surface may result in larger friction angles for deep landslides compared to shallow failures, the gentle inclinations attenuate this increased friction angle even under saturated conditions. In summary, the change in friction angle with landslide thickness is certainly sensitive to groundwater assumptions, but the insensitivity of cohesion to effective stress conditions maintains a systematic exchange of proportional cohesive and frictional strength that is largely insensitive to groundwater.

Another excellent point by the reviewer is the consideration of drainage and hydraulic gradient relating to landslide inclination. In the prior analysis, we considered hydrostatic conditions for all landslides, which is a less consequential assumption for very gentle hillslopes, but of significance for steep hillslopes where the seepage pressures may be vastly different from hydrostatic (Reid 1997). We apply the correction for slope-parallel seepage proposed by Reid (1997) where porewater pressure (u) is no longer hydrostatic ($u = \gamma_w h_w$), but rather considers approximate slope-parallel seepage conditions ($u = \gamma_w h_w \cos^2(\theta)$). Consideration of these conditions does result in a decrease in friction angle for steep, shallow landslides with a rather modest decrease for deep-seated gentle landslides. Nonetheless, the systematic exchange between frictional and cohesive

strength with landslide thickness still remains, as shown in the figures below. The text has been modified as:

“A larger uncertainty in this analysis owes to variability in groundwater flow-fields; thus, various groundwater conditions were evaluated as frictional strength is directly influenced by effective stress conditions associated with saturation (Figure SI.7, SI.8) and described briefly herein, but “half-saturated” conditions (i.e., the thickness of saturated soil and/or rock for the landslide is half of total landslide thickness on a column-by-column basis, $m=0.5$) and slope-parallel seepage are used for analyses described in the main text. We choose this condition as a conceptual “average” hydrological control as it is plausible to have perched groundwater in overlying soil or weathered rock stemming from intense rainfall, but saturated conditions are more likely to be persist deeper in bedrock (Salve et al. 2012). The sensitivity of varied groundwater conditions is explored in the Figures SI.11 and SI.12, but observed trends in comparative proportional frictional and cohesive hillslope strength are relatively insensitive to these conditions (e.g., <5% change in proportional strength).”

and:

“An approximation of landscape-scale regional flow patterns would enable more confident trends in shear strength as controlled by groundwater flow. However, such models rely on (1) unknown regional-scale boundary conditions and parameters (e.g., hydraulic conductivity), and (2) and homogenous conditions – at this stage, there is insufficient data as to apply such conditions for our analysis. Thus, we evaluate the sensitivity and bounds of groundwater assumptions ranging from dry to fully saturated conditions, observing that the exchange of shear strength magnitude is sensitive to groundwater assumptions; however, the proportional exchange in strength with landslide thickness (described below) is less sensitive to groundwater assumptions. Seepage stresses may cause localized instability – for example, the near-surface (e.g., regolith and saprolite) is more sensitive than deeper flow regimes to changes in pore water pressures owing to rainfall infiltration and higher variations in seepage stresses are observed at shallow depths (Reid and Iverson, 1992). At greater depths, changes in effective stress fields induced by seepage are less pronounced and also exist at depths where cohesion may be appreciable, adding a nontrivial length scale to stability (i.e., seepage fields may be less important when cohesion plays a significant role in stability). At more shallow depths (the top few meters) where friction is slightly more dominant, these seepage fields and their influence on effective stresses may be more important, which supports the known sensitivity of shallower landslides to groundwater changes stemming from infiltration.”

Figure SI.7. Relationships between landslide area versus strength and mean landslide thickness versus strength for a suite of groundwater conditions. We consider dry conditions, groundwater depths of 20m, 10m, 5m, 2m, 1m, 0.5m, and full saturation (0m depth to groundwater). The exchange in proportional frictional and cohesive strength is largely insensitive to the groundwater conditions shown, although a modest increase in cohesive resistance is observed with decreasing groundwater depth. No evident exchange in proportional strength is observed for the range of landslide areas shown, but the trends are more apparent when comparing proportional strength to mean landslide thickness. Magnitudes of shear strength are sensitive to groundwater conditions. Generally, a modest decrease in friction angle and significant increase in cohesion is observed with increasing mean landslide thickness, but is diminished for discrete groundwater depths. A decrease in friction angle and modest gain in cohesion is shown with increasing landslide area for all groundwater conditions.

Figure SI.8. Relationships between mean landslide thickness versus strength for a suite of saturation ratios (m). We consider dry conditions, pore pressure ratios of 0.25, 0.5, 0.75, and full saturation. Comparatively, the exchange in proportional frictional and cohesive strength is largely insensitive to the pore pressure ratios shown, although a modest increase in cohesive resistance is observed with increasing r_u . Magnitudes of shear strength are sensitive to m . Generally, a modest decrease in friction angle and significant increase in cohesion is observed with increasing mean landslide thickness.

Minor points:

R1.2. Equation 1 and Figure 5c: If looking at Mohr-Coulomb, should the limit not be that the proportion cohesive resistance should be 100% if the landslide thickness reaches 0?

Many thanks for the observation – I can see how this is confusing. Equation 1 is the general form of the Mohr-Coulomb strength criteria. This statement would certainly be correct if cohesion was constant with depth. However, our inversion does not show

cohesion to be constant with depth as it tends to decrease with landslide thickness. Further, there is often still cohesion at the thickness of very shallow landslides (Figure 2, Figure 3b). As Figure 5c is a “cartoon” and the fact that there may be some apparent cohesion (e.g. from roots) at shallow depths, we intentionally made the intercept for proportional cohesive resistance as a nonzero value, although the reviewer is correct that this only applies for *finite* depths. The caption for Figure 5 has been modified to clarify this:

“...This material, which is subject to low confining stresses at finite depths, is primarily frictional, has low cohesion and often dilative, exhibiting friction angles that are greater than those that would be encountered under higher normal stresses...”

R1.3. Can you elaborate on your definition of effective angle of internal friction and effective cohesion? As this also comes back in your discussion on the weathering hypothesis.

Excellent query. These nomenclature of “effective” strength parameters reflect mobilized strength under effective stress or “drained” conditions. That is, volume change is possible (not undrained loading), excess pore pressure generation is not significant, and rate-dependent strength is not considered. The following text was added to the Methods section:

“...back-analyzed landslide effective cohesion (c' , kPa) and effective friction angle (ϕ' , °) to determine effective normal and shear strength for failure envelopes. As normal and shear stresses may vary throughout the landslide slip surface, for comparative purposes we simplify these stresses into scalar values using mean landslide thickness for comparison on a landscape scale using Mohr-Coulomb relationships based on mean basal stresses. The effective strength parameters are representative of drained conditions (i.e., loading is slow and rate-dependent excess pore pressures are not appreciable, effective stress conditions are maintained).”

R1.4. The authors state a few times that the dominant soil geometry is approx.. 2 m soil and then saprolite/bedrock. If this is so dominant, why not doing the 3D geomechanical back analysis with this geometry? Can the authors quantify and discuss what would be the effect of using a two-layer geometry?

This is a fair question. However, in absence of site-specific information about stratigraphy, this is a difficult assumption to make. Further, as is common in absence of this information, conventional back-analyses often rely on the simplification of evaluating the strength of a homogenous medium, particularly at this scale (e.g., Gallen et al. 2015, Schmidt and Montgomery 1995). In the case of say, soil overlying bedrock, it would be plausible that the “smeared” cohesion value of a given landslide that shears through both strata would realistically have to overcome cohesion more concentrated in rock than in soil (ignoring other variables like suction and root strength). However, the range of depths considered, the generally shallow depths of soils (in these revisions, between 0.9-1.2m), and sensitivity of landslide thickness and cohesive length scales would likely not significantly affect the observed overall exchange in proportional

strength regimes. The following has been added to a new “Limitations and Uncertainties” section:

“For a given landslide, the inversion of strength presented treats the landslide mass as a homogenous material, which is a simplification with potential significance in layered materials or interfaces between strata (e.g. soil-bedrock boundaries). However, this simplification is common for back-analyses in absence of stratigraphic information is common (e.g. Schmidt and Montgomery 1995, Gallen et al. 2015).”

R1.5. Is it possible to also look at the location of landslides on a slope: in the convex or concave part of the hillslope. Is there information on that?

A good question – we thank the reviewer for the suggestion. We have performed an analysis to calculate the mean planform curvature, profile curvature and total curvature (planform+profile curvature) of each of the analyzed landslide deposits. There are many forms in which curvature may be calculated, but we evaluate planform and profile curvature calculated as the second derivative along and perpendicular to the steepest downward gradient, respectively. In general, modest positive planform curvature (i.e. convergent terrain) was observed for most landslide deposit areas (93%), with a relatively small median planform curvature of $1.9 \times 10^{-3} \text{ m/m}^2$. Similarly, modest convex (concave down) profile curvature was found (median of $-1.5 \times 10^{-3} \text{ m/m}^2$) with 90% of landslide deposits being convex. The observed distributions of curvature for landslide deposits do not tend to exhibit very extreme curvature (e.g. planform curvature in the order of $-1 \times 10^{-2} \text{ m/m}^2$ and $1 \times 10^{-2} \text{ m/m}^2$ for channels and ridges in the Oregon Coast Range, Roering et al. 1999). Reconstructed landslides have much surface lower curvature. Of course, mean curvature within the deposit is sensitive to extreme values, resolution, and potentially dislocated topographic features. We have added the following figure to the SI document:

Figure SI.6. Distributions of (a) planform, (b) profile, and (c) total surface curvature for landslide deposits. Distributions of (d) Planform, (e) profile, and (f) total surface curvature for landslide source. The median curvature is shown with a red, dashed line. A CDF is shown to reflect the distribution of curvatures.

We have also added the following text to the narrative:

“The inpainting technique used to reconstruct landslide source surface geometry is based on satisfying a set of curvature-preserving differential equations based on surrounding topography - planform, profile and total curvature for deposits and source areas are shown in Figure SI.6.”

R1.6. The same holds for possible diverging / converging slope geometries. Any information on that?

Excellent comment. Please see above.

R1.7. L346-347: I do not entirely understand this argument: even for hillslope scale the effective cohesion values are modest. Can you compare to other reported hillslope scale cohesion values in the Oregon regions?

Apologies for the lack of clarity here – our intention is to suggest that our observations (that hillslope- and mountain-scale cohesion values are modest, particularly in

comparison to those derived from laboratory tests) generally align with the findings of others, particularly when it comes to “cohesion”. To the best of our knowledge there is no large database for back-analyzed shear strength values *at a hillslope* scale in Oregon. There are several potential reasons for this, including that two-dimensional analyses may significantly overestimate strength, particularly cohesion (Stark and Eid 1998). Thus, the seemingly modest cohesion values may seem smaller in comparison to two-dimensional methods owing to a more comprehensive consideration of resistance along the boundaries of landslide rupture surfaces that do not align with the direction of sliding. Another reason is that the analysis presented herein describes incipient failure (i.e., hillslopes at the cusp of failure, where significant fractures may have already formed). Lastly, landslides may be the “exception to the rule” for landscape-scale strength; that is, they likely occur where significant weaknesses (joints, discontinuities, high clay contents) are dominant controls on strength. We have added the following text to the narrative:

“However, we note that cohesion values for bedrock landslides are still modest. This is perhaps expected, as numerous studies have described large discrepancies between rock mass strength at hillslope and laboratory scales (Schmidt and Montgomery 1995, Gallen et al. 2015, Medwedeff et al. 2020), which have commonly been attributed to weakness or discontinuities in bedrock that are not captured when testing small specimens for strength. These discontinuities are often filled with suffused clay minerals (Barton 2016), and groundwater (Moon et al. 2017), which in combination with large suppressed dilation from large overburden stresses, act to diminish frictional resistance.”

And:

“The presented trends do not contradict that mineral and root cohesion can be large in the soil mantle, and that friction may increase with landslide thickness; rather, they reflect a first-order set of trends that reflect the conditions that represent failure and the associated localized weaknesses that yield failure. For example, studies have made interesting observations that when a threshold slope gradient is exceeded in soil-mantled landscapes, there is less frequent shallow landsliding owing to cohesive boundary forces stemming from vegetation and/or mineral cohesion (Prancevic et al. 2020). It is suggested that beyond these threshold slope gradients, soil erosion in these environments may owe more to creep, which prevents requisite thickening of the soil mantle for landsliding. The censoring of small landslides in these inventories may indeed show a strong cohesive control at very shallow depths. Other studies have described that friction angle may increase with depth, at least in the soil mantle (e.g., Lu and Godt 2008). These strengthening factors may be dominant in many landscapes, including the areas described herein. However, the landslide behaviors presented in this study do not refute those observations; rock mass may have very large friction angles, cohesion stemming from mineral behavior, roots, and capillary stresses may be significant at shallow depths in soil. This study focuses on landslides that have occurred and do not necessarily represent complete landscape strength conditions; rather, the landslides used to construct relationships here may reflect the exception: localized strength conditions that result in landsliding (i.e., low root density or mineral cohesion, weakened and slickensided fractures in soft rock mass with partially intact rock bridges – all of which result in landslide erosion). Still, this analysis does provide perspective as to the exchange of strength within a gradient of weathering at depths reflective of soil and particularly bedrock.”

Stark, T. D., & Eid, H. T. (1998). Performance of three-dimensional slope stability methods in practice. *Journal of Geotechnical and Geoenvironmental engineering*, 124(11), 1049-1060.

R1.8.L405: small or negligible magnitude of cohesion in shallow landslides. That is a strong statement, knowing that in your analysis also apparent root cohesion of small and larger vegetation is included in the effective cohesion value and generally seen as very large and important. Moreover, a few kPa on cohesion equals several decimetres of soil stress..... Maybe do some back of envelope calculations to proof your statement could be useful.

This is a valuable comment and we have toned this statement down. Certainly roots will add a strengthening factor and cohesion is present in small landslides, but at a lower magnitude. We do, however, want to emphasize that estimates of root strength are highly variable and have the potential to be overestimated. The seminal works (Waldron 1977, Wu et al. 1979, Schmidt et al. 2001) focused on lateral root cohesion are based on a variety of assumptions that may amplify the perceived stabilizing effects of root structures. The largest overestimate of root reinforcement *likely* owes to the assumption is that all roots fail at once. Another source of overestimated root strength is the assumption that all roots break at failure, as opposed to pullout of sufficiently strong but weakly anchored roots. However, numerous more recent studies (e.g., Pollen et al. 2005, Cohen et al. 2011, Cronkite-Ratcliff et al. 2018, Giadrossich et al. 2019) have demonstrated that classical root reinforcement models may overestimate root lateral strength by 50-80%. Further, root lateral reinforcement is a strong function of root density, which decays with distance from the stem of a tree (Sakals and Sidle 2004, Roering et al. 2003). Studies using the “root cohesion” metric for stability (e.g. Wu et al. 1979) but evaluating the spatial distribution of roots (e.g., Sakals and Sidle 2004) suggest that root cohesion can regularly be in the range of ~1kPa, particularly in managed forestland (as is much of the area of these studies). Further, as shown in Table 2 and Table 4 (literature synthesis) of Schmidt et al. 2001), root cohesion values can regularly be in the range of ~1kPa (or less) at modest depths (~1-2m) *without accounting for* the spatial distribution of roots, pullout, or the progressive breakage of different size roots at failure. All of these elements suggest that the traditional (pre-2005) definition of root cohesion is potentially overestimated by 1-2 orders of magnitude and generally consistent with plausible ranges suggested in this study. We have added the following text to the manuscript to expound on this point:

“While many shallower landslides (e.g., $D < 1$ m) have limited cohesion, some cohesive strength is often still present. In many instances, cohesion accommodates over 50% of shear strength in some instances of small landslides ($D < 1$ m, Figure 3). However, this proportion of strength does not reflect large magnitudes of cohesion owing to small thicknesses and shear stresses. A mean cohesion of 0.83, 4.93 and 25.2 kPa is associated with thicknesses of $D < 1$ m, $1 \leq D < 10$ m, and $D \geq 10$ m, respectively. At shallow landslide depths, these cohesive strengths may result from the influence of lateral root reinforcement. As cohesion is back-analyzed uniformly over the entire rupture surface in this analysis, the influence of possible vegetation is evaluated with back-analyses with a variety of root cohesion at shallow depths (0.5 m) conditions reflecting forest management practices from Schmidt et al. (2001), shown in Figure SI.15. The presence of lateral root cohesion greatly reduces back-analyzed mineral cohesion at modest depths ($D < 1$ m), but much less so for moderate and only modestly so for landslides with significant thickness (i.e., bedrock landslides). This suggests that that equivalent cohesion values back-analyzed at larger depths or thicknesses (applied over the entire shear surface) are greater than equivalent root cohesion only applied to the near-surface. At shallow depths (i.e., $D < 1$ m), equivalent cohesion values are at the lower-bound of “root cohesion” values observed in the Oregon Coast Range

(Schmidt et al., 2001) and may owe to the aforementioned censoring of smaller landslides; however, it has been suggested that underlying assumptions that describe root cohesion values may overestimate the true strength from roots by as much as 75% (Pollen et al. 2005, Cohen et al. 2011, Giardrossich et al. 2019), and potentially even more when considering sparse vegetation and the spatial heterogeneity of roots (Sakals and Sidle 2004).”

- Waldron, L. J. (1977). The shear resistance of root-permeated homogeneous and stratified soil. *Soil Science Society of America Journal*, 41(5), 843-849.
- Wu, T. H., McKinnell III, W. P., & Swanston, D. N. (1979). Strength of tree roots and landslides on Prince of Wales Island, Alaska. *Canadian Geotechnical Journal*, 16(1), 19-33.
- Schwarz, M., Cohen, D., & Or, D. (2012). Spatial characterization of root reinforcement at stand scale: theory and case study. *Geomorphology*, 171, 190-200.
- Pollen, N., & Simon, A. (2005). Estimating the mechanical effects of riparian vegetation on stream bank stability using a fiber bundle model. *Water Resources Research*, 41(7).
- Cohen, D., Schwarz, M., & Or, D. (2011). An analytical fiber bundle model for pullout mechanics of root bundles. *Journal of Geophysical Research: Earth Surface*, 116(F3).
- Cronkite-Ratcliff, C., Schmidt, K. M., & Wirion, C. (2018, December). Revisiting apparent root cohesion at the Coos Bay, Oregon landslide: A comparison of three models. In *AGU Fall Meeting Abstracts* (Vol. 2018, pp. NH21B-0820).
- Giardrossich, F., Cohen, D., Schwarz, M., Ganga, A., Marrosu, R., Pirastru, M., & Capra, G. F. (2019). Large roots dominate the contribution of trees to slope stability. *Earth Surface Processes and Landforms*, 44(8), 1602-1609.
- Sakals, M. E., & Sidle, R. C. (2004). A spatial and temporal model of root cohesion in forest soils. *Canadian Journal of Forest Research*, 34(4), 950-958.
- Roering, J. J., Schmidt, K. M., Stock, J. D., Dietrich, W. E., & Montgomery, D. R. (2003). Shallow landsliding, root reinforcement, and the spatial distribution of trees in the Oregon Coast Range. *Canadian Geotechnical Journal*, 40(2), 237-253.

It has been a pleasure to read this paper and I wish the authors success with their work.

Dr Thom Bogaard,
Delft University of Technology, Delft, The Netherlands

Once again, we thank the reviewer for sharing their expert feedback and guidance – it has certainly helped improve this work and better highlight future aspects to explore.

Reviewer #2

This is a very interesting and potentially important paper though it is currently a little difficult to follow in places. The paper is set up as an explanation for size distributions but I don't think that is really what the paper is about. Instead the authors use a back analysis of thousands of landslides to identify depth dependent changes in friction angle and cohesion and argue that these are connected to bedrock weathering. The large sample back analysis is a spectacular effort and the conclusions about friction angle and cohesion variation with depth are extremely surprising and thus very important. On the other hand, I don't find the connections to landslide size convincing at all.

My major concern with the paper is that the friction angle estimates rely on a method that generates spurious correlations with landslide inclination. I did not find sufficient information in the paper to fully understand the method for friction angle estimation and my comments below focus on this aspect of the work. Concerns around this aspect of the method and the consequent results, which have opposite trends to those from existing studies make me extremely nervous that there is an error here that propagates throughout the results.

The authors appreciate the incredibly thorough, thoughtful, and enlightening feedback of the reviewer. Indeed, added clarity regarding some of the observations and connections to landscape-scale behavior was needed.

We have attempted to address these concerns, described herein, through rerunning analyses, modifying the model for a variety of conditions, adding numerous sensitivity analyses regarding assumptions and evaluation of several metrics that relate to landslide morphology, evacuation, deposit stability, and the strength of roots.

Also, the reviewer is completely correct – the association with landslide size has a causation or correlation element that likely muddles the observations of this study. Thus, we have chosen to focus on landslide thickness as this is more intuitively connected to consideration of with trends of weathering at depth.

Major comments

R2.1 The paper needs a title that is more in line with the paper's claims and a more coherent argument to support these main claims. The paper isn't really about size it is about rock/soil strength back calculated from observed landslides. That doesn't make it unimportant! It is still a very interesting paper and its findings are surprising and therefore important. The claims are clearly outlined in the final quarter of the abstract but I get lost in the text. I need more help following your logic if I am to be convinced that the claims of your abstract are justified by your data.

This is a fair comment, and throughout writing this manuscript, something we found challenging: that is, causation versus correlation. We have instead slightly modified the

focus of this work to describe trends in strength as associated with landslide thickness and weathering. We have instead focused on the nonlinearity of failure envelopes as related to normal stress (and implicitly landslide thickness), exchange of cohesive strength proportion with landslide thickness as a proxy for weathering, and changing strength properties versus landslide thickness as a potential manifestation of weaknesses. We have also added further justification of the methods used. The paper has been modified extensively throughout to reflect these objectives – we appreciate the guidance. The title has been revised as:

“Inversions of Landslide Strength as a Proxy for Subsurface Weathering”

The abstract has been revised as:

“Distributions of landslide size are hypothesized to reflect hillslope strength, and consequently weathering patterns. However, the association of weathering and critical zone architecture with mechanical strength properties of parent rock and soil are poorly-constrained. This knowledge gap persists because inversions of strength at field scales are typically indeterminate and strength properties are inherently variable at a landscape level. That is, typical inversions have relied on using simplified landslide geometry while holding cohesion and/or frictional strength properties constant, precluding insights into the potential association between subsurface strength and patterns of weathering. Here we use a three-dimensional stability model to analyze over 7,300 landslides in western Oregon to infer combinations of strength - friction angles and cohesion - through analysis of both failed and reconstructed landslide terrain. Our results demonstrate that the failure envelope that relates shear strength and normal stress at a landscape scale is nonlinear. Further, despite the variability in material strength at large scales, we observe trends in strength associated with landslide thickness that may serve as a proxy for subsurface weathering. We show that frictional resistance is the dominant strength control for most landslides in the study area although the proportion of contribution from cohesion increases with landslide thickness, consistent with deep-seated slope failures, typically in bedrock, that occur on gentle inclinations. We posit that the observed relationships between strength and landslide thickness are associated with the coalescence of zones of low shear strength driven by weathering, which constitutes a first-order control on the mechanical behavior of underlying soil and rock mass.”

R2.2: The paper needs more detail on and a stronger justification for the method of estimating friction angle. The method used to calculate unique values for both friction angle and cohesion involves two back calculation procedures for each landslide. Both involve interpolating surfaces from sparse marginal data. My main concern with the paper is the first back calculation that is used to calculate friction angle under the assumption that deposits within the scar are in limiting equilibrium (along the pre-existing failure mechanism) and are cohesionless. My concerns are: 1) that for many of the landslides I have seen (in person or in pictures) deposits are either patchy or absent in the scar and I don't know how you can apply your procedure in either of these cases; 2) a significant fraction of the pre-existing failure surface will now be the ground surface and shouldn't be included in the analysis (I guess it isn't but this wasn't clear to me from the paper); 3) the deposits that remain in the scar are not necessarily in limiting equilibrium,

they can't be unstable but we have no indication of how stable they might be, as a result the first back calculation is a lower limit on friction angle but one that is also very strongly influenced by the scar topography (i.e. deposit angles do not reflect their friction angle but simply the angle of the scar within which they find themselves). This final point is the one I'm most concerned about because it would introduce a spurious correlation between landslide slope and friction angle. This would propagate through the known (and observed here) negative correlation between landslide slope and landslide depth to generate the relationship that you observe where friction declines with depth.

To me the depth dependent reduction in friction angle is the most important finding of the paper and the basis for the other ideas that follow so it is really important that your method to estimate friction angle is clear and that you deal with these concerns around possible errors in the estimate. The magnitude of the reduction in friction angle with depth is really surprising and since the reduction is most rapid at shallow depths it is shallow landslides that should demonstrate this behaviour most clearly.

In the landscapes I have worked in 90% of the landslide scars were empty of deposit. You would be unable to resolve friction angle for any of these slides but if you assumed they were cohesionless that would also be incorrect, roots typically provide considerable additional strength. Perhaps the types of landslides that you are examining here are different to those I have in mind. This would be important to state in your early definition of landslides and in your explanation of what is and is not included in your landslide inventories.

In the explanation of your back-calculation method, more information is needed on: 1) how you identify deposits within the scar and define the potential failure surfaces associated with those deposits; and 2) how many scars do not contain identifiable deposits in the scar, and their characteristics (e.g. size, depth, slope).

The authors appreciate these excellent comments. First, a brief clarification - we do not include the daylighted slip surface topography reflecting the failure surface in our back-analysis of the landslide deposits. We have attempted to make this clearer in the *Methods* section with this revised text:

“The approach was applied using the rupture surface determined from the thin-plate spline, and first applied to existing landslide topography (i.e., deposits at rest, Figure SI.4b) to determine the friction angle for each landslide in absence of cohesion (e.g., bonds of cohesion have been broken), excluding the headscarp. To obtain unique combinations of friction angle and cohesion, we then rerun the back analysis on all landslides using (1) the same rupture surface (Figure SI.4d), (2) the friction angle determined from the existing landslide deposit (Figure SI.4b) and (3) reconstructed landslide topography (Figure SI.4c) to determine the cohesion that yields equilibrium.”

The reviewer raises several points that are justified, and to some level, insufficient information is available to “prove” the level of evacuation of landslides within the given inventory. That is, event-based inventories or inventories derived from vertical differencing (e.g. Massey et al. 2018, Bernard et al. 2020) provide more conclusive,

quantitative evidence for the magnitude and distribution of landslide evacuation. However, many landslide inventories rely on expert mapping and are not necessarily associated with singular destabilizing events, but rather reflect the best interpretation of landslide characteristics; in this case, these landslides were mapped with bare earth lidar by expert geologists. For those landslides not associated *or associated* with known events, should landslides always be catastrophically evacuative? Isn't it true that many landslides "creep" or move slowly downhill in a residual state at modest rates (if active)? For example, Lacroix et al. (2020) describes that 'most slow-moving landslides typically occur in weak materials, either Quaternary soils or highly damaged sedimentary layers with gentle slope angles (<20°...). Both materials often have interbedded clay-rich layers that host the sliding surface or failure zone...' Within the analyzed inventory, 81% of the analyzed landslides are in sedimentary or quaternary deposits. The median slope of the inventoried landslides is 21.3°, with approximately 90% of landslides having mean surface slopes less than 32°, respectively. Of course, the slow-moving characteristics of some landslides does not preclude catastrophic failure (as described nicely in Lacroix et al. 2020), but it does suggest that while the suggested bias towards runout may exist, the treatment of some proportion of landslides as non-evacuative is not farfetched.

Considering this important potential bias, we reran all analyses to account for spatial overlap of deposits and source area, as well as other sensitivity analyses suggested by reviewers (e.g. root "cohesion"). We now consider the proportional spatial overlap (OL) of modeled landslide source areas (A_{source} , reconstructed area on the fitted slip surface with nonzero thickness) and landslide deposits (A_{deps} , existing deposit area on the fitted rupture surface with nonzero thickness) versus the entire area of both the headscarp and deposit (A_{total}), or:

$$OL = \frac{A(A_{source} \cup A_{deps})}{A_{total}}$$

The median proportional overlap of source landslide area and deposit landslide area to the total landslide area from the model is approximately 0.58. Approximately 90% of the landslides have proportional spatial overlap greater than 0.21. We choose an arbitrary threshold of 25% overlap to be considered non-evacuative. As shown below, the results are not overly sensitive to these constraints, which is perhaps sensible as the controls on catastrophic failure extend far beyond shear strength alone (e.g. pore pressure response, topography, etc.). Below are the strength exchange relationships for OL greater than 0.1 (~7600 landslides), 0.25 (~7300 landslides), and 0.5 (~6000 landslides). Once again, the same overall exchange in strength is present.

We have modified the discussion text to reflect these constraints.

"As rupture surfaces are a modeling output and ultimately, a first-order estimate of landslide geometry used to evaluate strength controls on instability, there is no direct means of directly isolating the observed trends in strength to ignore a bias from landslides with extensive evacuation and runout. We consider simple criteria by which landslides are evacuated, including the proportional overlapping deposit and modeled source area versus total landslide area (deposit and headscarp). We observe that the landslide inventory herein has a median areal overlap of 58% between source and deposit, which

suggests that full evacuation for the landslides used in this analysis may not exert a strong bias associated with long runout landslides. We select a 25% overlap as an lower bound on non-evacuative landslides (approximately the 10th percentile for overlap).”

Strength relationships between proportional overlap of deposit and source areas. The left panel represents 50% minimum overlap. The middle panel represents 25% minimum overlap (same as in the narrative). The right row represents 10% minimum overlap. Minimal difference is observed between results.

As to the level of equilibrium of the landslide deposit, we acknowledge this is a source of uncertainty. However, for landslides that are not completely evacuated and are (or were) active, wouldn't the topography of the deposits indeed represent a state close to a metastable condition? To illustrate this, let us use the classical limit equilibrium definition of a “Factor of Safety,” or FS , where for purely frictional conditions (such as that assumed in the failed deposits), FS is effectively introduced as a means of reducing strength to yield global equilibrium as:

$$\tan(\phi'_m) = \frac{\tan(\phi')}{FS}$$

where ϕ'_m is the mobilized, effective friction angle – i.e. a proportion of the actual effective friction angle (ϕ'). Using the current approach, FS is assumed to be unity and both ϕ'_m and ϕ' are equivalent, reflecting full mobilization of friction within the landslide deposit (i.e. metastable conditions). If we suppose that the mobilized friction for equilibrium of the landslide deposit is only a proportion of the full frictional strength, FS may be used to explore the level of stability of the landslide deposits and any bias this disequilibrium would have on the systematic exchange of strength. This may be introduced by applying defining $\tan(\phi')$, which is used to back out cohesion from the source area, as:

$$\tan(\phi') = FS \tan(\phi'_m)$$

A sensitivity analysis for FS of 1.05, 1.10 and 1.25 is performed using this relationship and added to the SI. The latter value ($FS=1.25$) is commonly used in the design of stable,

engineered slopes. As shown, the same general trends hold, but of course, the proportional and absolute contribution of friction is enhanced.

Figure SI.16. Relationships between mean landslide thickness and strength for different factors of safety (FS) applied to the friction angle of landslide deposits. This sensitivity analysis tests the possibility that landslide deposits are not unstable and consequently not in a state of equilibrium. As shown, increased FS applied to deposit friction angles yields lower cohesion values and offsets the exchange between cohesive and frictional resistance, but an exchange is still present.

We evaluate deposit stability from a different perspective. Supposing $m=0.5$ conditions for simplicity, let's presume that all resistance (both source and deposit) stem from friction. Thus, we can back-analyze source areas for ϕ'_{source} , and then determine deposit stability based on its back-analyzed friction angle as:

$$FS = \frac{\phi'_{source}}{\phi'_{deposit}}$$

Under this set of conditions, we observe a median FS of 1.26 for the entire inventory of landslide deposits. This FS value is commonly applied in design of engineered slopes (that is, slopes *designed and constructed* to be stable). Landslide deposits may not be unstable, but they are often considered to be areas of significant hazard, and for good reason: these deposits are frequently unstable. While there is no way to ascertain the stability of a deposit, it would be difficult to imagine that a majority of the time, these deposits are at higher factors of safety compared to slopes that are engineered to be stable. Landslide deposits are often considered to be unstable and are frequently sources for additional landslides (Temme et al. 2020). These results have been added to the SI.

Figure SI.17. Distribution of factor of safety (FS) considering no cohesion at the source area at the time of failure, simplified under purely frictional conditions as $FS = \tan(\phi'_{source})/\tan(\phi'_{deposit})$. This distribution shows that the median FS for deposits would be ≈ 1.26 , which on par with the stability of engineered slopes. As landslide deposits are often prone to continued failure from being placed in a precarious state (Temme et al. 2020), such a high FS is possible, perhaps not always likely. This suggests that back-analysis of friction from deposits and subsequent use for back-analysis of cohesion from source areas, while an assumption, may serve as a reasonable technique to glean first order estimates of unique cohesion and friction pairs for landslide inventories.

We have added sensitivity analyses to the SI, and the following text to the discussion.

“The role of evacuation may control the level of stability of landslide deposits, which are used to determine the friction angle in this study. That is, landslide deposits are treated to be a state of limiting equilibrium associated with continued activity. This is a significant and necessary assumption for the proposed inversions but has a basis in reality as landslide deposits (even relict deposits) are often prone to reactivation and continued instability (Temme et al. 2020). We perform analyses to evaluate the level of stability of deposits considering back-analyzed friction angles from source areas using a factor of safety (FS), which for purely frictional materials

reduces to $FS = \tan(\phi'_{source}) / \tan(\phi'_{deposit})$. Under these conditions, the median FS is 1.26, almost equivalent to specified FS values of designed, engineered slopes ($FS \approx 1.25-1.3$, Figure SI.16). We also investigate strength trends through application of a FS to deposit friction angles and subsequent recalculation of landslide source cohesion (Figure SI.16, Figure SI.17). We observe similar exchanges in proportional friction and cohesion, although the exchanges are offset.”

Of course, this comparison is reliant on modeled inputs/outputs. Nonetheless, these sensitivity analyses do demonstrate that stable deposits would diminish the observed strength trends, but not entirely. There is indirect evidence, albeit somewhat speculative, that many of the landslides analyzed herein are not fully evacuative owing to (1) the inventorying belying reliable mapping of a headscarp, toe and deposit for a given landslide feature, and (2) these morphological features that do not suggest full evacuation. Using screening of various features as described, the systematic exchange in strength controls with landslide size are still apparent.

Further, it is sensible that there would be a bias between slope and landslide size when considering deposits. However, we note that through examining mean landslide inclination of landslide source areas, a similar systematic decrease is observed, as shown below.

Mean landslide inclination of landslide source areas.

Numerous works for simplified slope stability analyses – both two- and three-dimensional - have demonstrated that both (1) a transition from friction-dominated to cohesion-dominated strength and (2) a transition from steep to more gentle slope inclinations will result in the deepening of the critical failure surface from shallow to deep-seated failure (e.g. Sun and Zhao 2013, Gao et al. 2013, Michalowski 2002, Taylor 1937). Klar et al. (2011) even explores slope stability techniques to demonstrate the theoretical role of strength on scaling laws, showing that the landslide length and thickness scales of landslide thickness are sensitive to friction and slope inclination (Figure 7 in Klar et al., 2011), although they do not have a very pronounced influence on area-volume scaling. These behaviors are consistent with regional-scale observations shown herein. Such conditions could theoretically be driven by the increasing dominance of cohesive strength.

Further, the consideration of decreasing friction angle is supported by a variety of failure criteria. For example, various failure criteria for rock mass (both intact and fractured) emphasize a nonlinear relationship between normal stress and friction (Barton 2006, Hoek and Brown 1980).

This behavior has also been observed in rockfill (Leps 1970, Barton 2016) and as stated before, soil (Rowe 1962). As we would expect larger effective normal stresses at a landslide rupture surface with increasing mean thickness (and further amplified by decreasing inclination), it is sensible that this decreasing effective friction angle is observed at these scales.

The narrative has been revised substantially to reflect these concepts:

Figure 2. Inversions of shear strength ($m=0.5$ conditions) and associated stresses based on landslide thickness ($n=7331$ landslides). (a) Effective normal stress (σ') versus shear stress (τ) for landslide inventories colored by mean landslide thickness. Black lines represent moving median and 10% and 90% quantiles. Power law regressions are fit to the moving quantiles for a 1% moving window to represent plausible landslide failure envelopes. The insets show the nonlinearity of failure envelopes at low and high effective stresses, primarily owing to sensitivity of friction with normal stress. (b) Moving quantiles for both friction angle and cohesion are shown versus normal stress, fitted with power law functions.

“We generalize strength properties by creating a landscape-scale failure envelope, where mean effective normal stress (σ') and commensurate mean limiting shear strength (τ) are compared (Figure 2a). Power-law fits to moving 10th, 50th, and 90th percentiles (1% moving window) demonstrate that the failure envelopes diverge from a linear relationship between effective normal stress and limiting shear strength. That is, while individual landslide strengths are back-analyzed using Mohr-Coulomb criteria (an implicitly linear failure envelope), the landscape-scale failure envelope obtained from analyzing normal stresses and limiting shear strength for all slides in our landslide inventory is nonlinear. A comparison of effective normal stress and both friction angle and cohesion demonstrates the source of the nonlinearity (Figure 2b). There is an inverse correlation between σ' and ϕ' , and thus, a strong positive correlation between σ' and c' . The median friction angles associated at very low σ' are approximately $\sim 28^\circ$ and have small cohesion (≈ 0.5 kPa) for $m=0.5$ conditions. These low normal stresses include shallow failures that tend to be associated with steep terrain (Figure 1a), further decreasing effective normal stress but may also include relatively gentle failures (e.g., small earthflows, weak remobilized landslide debris). On the contrary, deep, presumably bedrock landslides demonstrate decreased frictional resistance, decreased landslide inclinations, and higher normal stresses in comparison to smaller, shallower landslides, but have significantly more cohesion. This trend of decreasing landslide inclination (Figure 1) with

increasing landslide thickness results in two interdependent stress conditions associated with back-analyzed, incipient failure from a landslide source – (1) frictional resistance must be sufficiently reduced as to result in an excess of shear stress that is accommodated by cohesion, and (2) shear stresses must be sufficiently large to overcome the cohesive resistance associated within the rock mass, requiring sufficient landslide thickness. Under these conditions, and particularly with more gentle landslide inclinations (which decreases shear stress), an increasingly large thickness is required to overcome cohesive strength (e.g., Frattini and Crosta 2013). If purely frictional conditions are considered (i.e., no cohesion is considered for back-analysis at the source area), a nonlinear envelope also occurs although friction angles are larger, as expected (Figure SI.13).

The nonlinearity of the inventory-derived failure envelopes, either for cohesive-frictional or purely frictional conditions is consistent with behavior of soils (Rowe 1962), rockfills (Leps 1970), and fractured or intact rock (Barton 1976, Barton 2016). Friction may be enhanced in soils (i.e., regolith) under low confining stresses owing to dilation and hardening stemming from grain interlocking (Rowe 1962). In rockfills, which are effectively a fractured rock mass similar to saprolite, prior research has described a decrease in effective friction angle that spans several orders of magnitude of increasing normal stress (Barton 1976, Leps 1970, Xiao et al. 2014) and is a function of density, grain angularity, crushing of asperities, and suppression of rolling resistance under shear. For rock masses, numerous studies have demonstrated a nonlinearity in failure envelopes (Barton 2006, Barton 2016, Hoek and Brown 1997). A nonlinear failure envelope exists for a variety of rock mass states (e.g., jointed, fractured, intact), owing to infilling of joints with weak suffused materials, diminished dilation along roughened joints, and crushing of asperities in joints (Barton 2016). For the presented conditions, the 90th percentile, median and 10th percentile failure envelopes generally follow representative nonlinear strength envelopes (Barton 2006) of fractured rock, jointed rock, and rock with clay-filled discontinuities, respectively. The attenuation of frictional strength with normal stress (and indirectly, depth) may serve as a proxy for a variety of mechanisms: namely, (1) brittle rock may mobilize cohesive strength before fully mobilizing friction (Barton 2013), (2) deep discontinuities are prone to suffusion of clay minerals that exhibit low frictional strength (Tembe et al. 2010) and/or agglomeration of weaknesses (Milledge et al. 2014, Bellugi et al. 2021), and (3) large effective stresses suppress frictional behavior (Barton 2006, Renani and Martin 2018). “

Klar, A., Aharonov, E., Kalderon-Asael, B., & Katz, O. (2011). Analytical and observational relations between landslide volume and surface area. *Journal of Geophysical Research: Earth Surface*, 116(F2).

Sun, J., & Zhao, Z. (2013). Stability charts for homogenous soil slopes. *Journal of geotechnical and geoenvironmental engineering*, 139(12), 2212-2218.

Taylor, D. W. (1937). Stability of earth slopes. *J. Boston Soc. Civil Engineers*, 24(3), 197-247.

Michalowski, R. L. (2002). Stability charts for uniform slopes. *Journal of Geotechnical and Geoenvironmental Engineering*, 128(4), 351-355.

Gao, Y., Zhang, F., Lei, G. H., Li, D., Wu, Y., & Zhang, N. (2013). Stability charts for 3D failures of homogeneous slopes. *Journal of Geotechnical and Geoenvironmental Engineering*, 139(9), 1528-1538.

R2.3: For friction (L390-91): ‘Approximately half of the observed exchange between frictional and cohesive strength occurs within depths considered to be soil (<1.9 m)’. From the Figures it looks as though estimated friction angle decreases by about (40 – 30 deg) as mean depth increases from 0.1 to 1 m, which is a huge reduction in friction angle with depth. The only studies I have come across that discuss depth dependent friction

angle in soil actually suggest the opposite relationship - i.e. friction angle increases with depth (Lu and Godt, 2008; and later papers by these authors). They argue that porosity decreases with depth (Selby, 1993; Cornforth, 2005) and friction angle decreases with porosity (Rowe, 1969; Marachi et al., 1969; Cornforth, 1973, 2005) thus friction angle should increase with depth (references all from: Lu, N. and Godt, J., 2008. Infinite slope stability under steady unsaturated seepage conditions. *Water Resources Research*, 44(11)).

You find that cohesion increases with depth in the soil (e.g. L417-8, Fig 3b). Again, I do not know of any research suggesting that cohesion should increase with depth in the soil, though I do know of several papers that suggest the opposite, usually on the basis that both root density and root diameter distributions decrease with depth (e.g. Bischetti et al., 2007, Montgomery et al., 2009).

Bischetti, G.B., Chiaradia, E.A., Simonato, T., Speziali, B., Vitali, B., Vullo, P. and Zocco, A., 2007. Root strength and root area ratio of forest species in Lombardy (Northern Italy). In *Eco-and ground bio-engineering: The use of vegetation to improve slope stability* (pp. 31-41). Springer, Dordrecht.

Montgomery, D.R., Schmidt, K.M., Dietrich, W.E. and McKean, J., 2009. Instrumental record of debris flow initiation during natural rainfall: Implications for modeling slope stability. *Journal of Geophysical Research: Earth Surface*, 114(F1).

These differences to the existing literature do not demonstrate that your findings are incorrect. In fact they show how important the findings are if correct. However, because they are so surprising, different and important, the method that underpins them is a particular focus for scrutiny.

You do try to provide some support from the literature for the idea that friction declines with depth but when I followed up these citations it wasn't obvious that their findings supported your claims. You cite Alejano and Alonso (2005) as evidence that soil friction angles decline with depth but their observations are at larger confining stresses >100 kPa and show much smaller reduction in friction angle and much larger friction angle than observed here (67-50 deg across the range 100-1000 kPa). You say (L407) 'reflect a known sensitivity of friction angle to confining pressures' but this needs supporting with citations. You say (L409) 'With decreasing normal stress, many soils may have enhanced friction angles stemming from dilation and interlocking of grains within its matrix (Townsend et al. 2020). However, Townsend et al. don't mention dilation, or interlocking and don't appear to show results in support of the statement that friction angles increase with decreasing normal stress. I couldn't find any reference to depth-dependent friction angles in Bellugi et al. (2021) either (L409-10). You make an argument around colluvial materials on L410-11, but geomorphologists typically count the entire soil mantle as colluvial, particularly in places like the Oregon Coast Range. Are you working from an alternative definition?

We appreciate these comments – we also agree that much of the description could use more clarity around potential controls on decreased friction angles. We have completely

reworked relevant text to highlight that (1) cohesion is still present in the soil column, and (2) there is a suite of research that emphasizes the relationship between normal stresses decreasing friction angles. Much of the text surrounding these references has been reworked or better contextualized. We have also rerun the model with landslides that are considered non-evacuative, with or without roots, under a suite of groundwater conditions modified from before (i.e. consideration of slope-parallel seepage), so some of this prior text is not as relevant. We have modified the statement regarding colluvial soils as shown in the text below.

We do not dispute that cohesion within soil may be significant, either mineralogical or derived from roots, and it is present in our back-analyses of soil. In fact, for some landslides with mean thicknesses in range of what might be considered soil, cohesion may account for upwards of 50% of strength. However, the influence of roots is spatially and temporally variable (Sakals and Sidle 2004, Roering et al. 2004), and its treatment as an apparent cohesion may result in an overestimate of total contributing strength (upwards of 75% in some cases; Pollen et al. 2005, Cohen et al. 2011, Giarossich et al. 2019). We have performed sensitivity analyses with root strengths from Roering et al. (2003) (Figure SI.15) and see that the presence of root cohesion in the upper portion of the soil mantle does control back-analyzed cohesion values at shallow depths, but does not have a major effect at depths associated with bedrock. Within bedrock, rock bridges in rock mass may impart a form of cohesive strength (Kemeny et al. 2003; although most likely tensile in nature). Most importantly, the observed strength patterns shown in this study do not refute that soil may have large real or apparent cohesion or that bedrock may have large friction angles; rather, these relationships reflect the localized, weakened conditions that yield failure at a large scale.

Further, there is published research that conceptually supports decreasing friction angles with increasing normal stress, and we have attempted to work this into the study. For these analyses, we observe an increase in proportional friction of approximately 6%, while approximately 12% occurs at greater depths (Figure 3). The moving median friction angle does decrease by about 4° for $m=0.5$ conditions for depths associated with less than a meter of soil (and drops of 13° and 2° for 90th and 10th percentile moving windows), and continues to diminish with larger normal stresses/thicknesses. These diminished friction angles could be associated with dilative behavior in soil (Rowe 1962), but even moreso in rock mass or rockfill (fractured or partially-fractured, Leps 1970, Barton 2016), and the decreased friction angles continue for normal stress ranges that span orders of magnitude.

We have added and revised text in several sections to better clarify our observations and limitations.

“We generalize strength properties by creating a landscape-scale failure envelope, where mean effective normal stress (σ') and commensurate mean limiting shear strength (τ') are compared (Figure 2a). Power-law fits to moving 10th, 50th, and 90th percentiles (1% moving window) demonstrate that the failure envelopes diverge from a linear relationship between effective normal stress and limiting shear strength. That is, while individual landslide strengths are back-analyzed

using Mohr-Coulomb criteria (an implicitly linear failure envelope), the landscape-scale failure envelope obtained from analyzing normal stresses and limiting shear strength for all slides in our landslide inventory is nonlinear. A comparison of effective normal stress and both friction angle and cohesion demonstrates the source of the nonlinearity (Figure 2b). There is an inverse correlation between σ' and ϕ' , and thus, a strong positive correlation between σ' and c' . The median friction angles associated at very low σ' are approximately $\sim 28^\circ$ and have small cohesion (≈ 0.5 kPa) for $m=0.5$ conditions. These low normal stresses include shallow failures that tend to be associated with steep terrain (Figure 1a), further decreasing effective normal stress but may also include relatively gentle failures (e.g., small earthflows, weak remobilized landslide debris). On the contrary, deep, presumably bedrock landslides demonstrate decreased frictional resistance, decreased landslide inclinations, and higher normal stresses in comparison to smaller, shallower landslides, but have significantly more cohesion. This trend of decreasing landslide inclination (Figure 1) with increasing landslide thickness results in two interdependent stress conditions associated with back-analyzed, incipient failure from a landslide source – (1) frictional resistance must be sufficiently reduced as to result in an excess of shear stress that is accommodated by cohesion, and (2) shear stresses must be sufficiently large to overcome the cohesive resistance associated within the rock mass, requiring sufficient landslide thickness. Under these conditions, and particularly with more gentle landslide inclinations (which decreases shear stress), an increasingly large thickness is required to overcome cohesive strength (e.g., Frattini and Crosta 2013). If purely frictional conditions are considered (i.e., no cohesion is considered for back-analysis at the source area), a nonlinear envelope also occurs although friction angles are larger, as expected (Figure SI.13).

The nonlinearity of the inventory-derived failure envelopes, either for cohesive-frictional or purely frictional conditions is consistent with behavior of soils (Rowe 1962), rockfills (Leps 1970), and fractured or intact rock (Barton 1976, Barton 2016). Friction may be enhanced in soils (i.e., regolith) under low confining stresses owing to dilation and hardening stemming from grain interlocking (Rowe 1962). In rockfills, which are effectively a fractured rock mass similar to saprolite, prior research has described a decrease in effective friction angle that spans several orders of magnitude of increasing normal stress (Barton 1976, Leps 1970, Xiao et al. 2014) and is a function of density, grain angularity, crushing of asperities, and suppression of rolling resistance under shear. For rock masses, numerous studies have demonstrated a nonlinearity in failure envelopes (Barton 2006, Barton 2016, Hoek and Brown 1997). A nonlinear failure envelope exists for a variety of rock mass states (e.g., jointed, fractured, intact), owing to infilling of joints with weak suffused materials, diminished dilation along roughened joints, and crushing of asperities in joints (Barton 2016). For the presented conditions, the 90th percentile, median and 10th percentile failure envelopes generally follow representative nonlinear strength envelopes (Barton 2006) of fractured rock, jointed rock, and rock with clay-filled discontinuities, respectively. The attenuation of frictional strength with normal stress (and indirectly, depth) may serve as a proxy for a variety of mechanisms: namely, (1) brittle rock may mobilize cohesive strength before fully mobilizing friction (Barton 2013), (2) deep discontinuities are prone to suffusion of clay minerals that exhibit low frictional strength (Tembe et al. 2010) and/or agglomeration of weaknesses (Milledge et al. 2014, Bellugi et al. 2021), and (3) large effective stresses suppress frictional behavior (Barton 2006, Renani and Martin 2018)..”

And:

“To express the relative contributions of friction and cohesion with landslide thickness, we normalize frictional and cohesive strength to total shear strength (Figure 3b). A transition in frictional to cohesive resistance is observed with increasing landslide thickness. At depths associated with shallow landslides that typically occur in soil (<1 m), the median contribution of

cohesion is approximately 11-17% of total strength, potentially reflecting factors that may appear as cohesion, such as roots and partial saturation (or the lack thereof). Beyond these depths in materials that would be comprised of saprolite, weathered bedrock and fresh bedrock (Rempe et al. 2014), the increase in median proportional cohesive strength continues to approximately 29%. The apparent plateau in this proportional exchange in cohesive strength results from the bounding values of our moving window and the large uncertainty reflects relatively few landslides having mean thicknesses beneath 0.5 m or beyond 10 m. Nonetheless, we anticipate that these trends are likely to persist, perhaps in a nonlinear fashion (similar relationships using binned data are shown in Figure SI.13). The known decreasing frequency of landslides with area and particularly depth may be a product of the trajectories of these strength exchanges, where significant cohesion (i.e., more intact rock) may preclude instability. Most observed landslides tend to occur within the range of landslide mean thicknesses where the strength transition is most pronounced, but the role of censoring from mapping cannot be excluded although it is not likely (i.e., larger landslides follow the expected negative power law for area-frequency). Nonetheless, the observed tradeoff in governing strength with landslide thickness may reflect the diminishing presence of weathering and fracture formation that control mechanical strength properties at depth. The exchange of strength with increasing thickness may suggest that there is a potential depth threshold where weathering diminishes or manifests at much lower rates (e.g., fresh bedrock, Rempe and Dietrich 2014). The observed transition in strength behavior tends to cover upper bounds of weathered bedrock depths observed in Oregon (~9 m, Anderson et al. 2002) and northern California (~20 m, Salve et al. 2014) although this thickness varies with channel distance and incision.

And:

“The observed nonlinearity in landscape-scale failure envelopes, suggests a depth-dependent control on landslide initiation and indirectly, frequency. The observed inverse relationship between landslide inclination and landslide thickness does not preclude the potential for large and deep bedrock landslides to occur in steep terrain of significant relief; rather, it suggests that sufficiently diminished frictional resistance is requisite for the onset of failure in bedrock, at least at broad scales. These diminished frictional strengths may stem from clay-filled discontinuities, groundwater conditions, or suppressed friction with high normal stress, among other factors. In order to have sufficient stress to overcome rock mass strength, there must be sufficient relief and thickness (Schmidt and Montgomery 1995; Frattini and Crosta 2013), large normal stresses as to suppress friction (Barton 2016), and sufficient weaknesses, fractures, and discontinuities where rock mass cohesion is reduced significantly. However, we note that cohesion values for bedrock landslides are still modest. This is perhaps expected, as numerous studies have described large discrepancies between rock mass strength at hillslope and laboratory scales (Schmidt and Montgomery 1995, Gallen et al. 2015, Medwedeff et al. 2020), which have commonly been attributed to weakness or discontinuities in bedrock that are not captured when testing small specimens for strength. These discontinuities are often filled with suffused clay minerals (Barton 2016), and groundwater (Moon et al. 2017), which in combination with large suppressed dilation from large overburden stresses, act to diminish frictional resistance. It is plausible that coherent, unfailed rock may have appreciably larger strength values than those observed from this inversion (Figure 4c). However, the strength values described herein represent incipient failure; thus, strength values in bedrock are only representative of the weakened strength of bedrock at failure that presumably occurs from weathering. The observed decrease in friction and increase in proportional cohesive resistance with depth serve as a potential proxy for a weathering profile.”

And finally:

” In the absence of strengthening from vegetation, shallow landslides in soil and/or bedrock tend to be dominated by frictional strength with small magnitudes of cohesion, if present. This frictional strength is associated with a relatively incoherent matrix of weathered material (Figure 4b). The elevated friction angles for small and/or steep landslides – and proportionally shallow depths - reflect a known sensitivity of friction angle to confining pressures (Figure 2a). With decreasing normal stress, many soils and fractured rock mass may have enhanced friction angles resulting from dilation and interlocking of grains within its matrix (Leps. 1970, Barton 2006, Townsend et al. 2020). Materials found at lower elevations on hillslopes may have diminished frictional strength from continued weathering, loosening during transport and higher contents of clay minerals (Moon et al. 2017) – this behavior is reflected by the observation that not all shallow landslides have high friction angles (Figure 2b). While many shallower landslides (e.g., $D < 1$ m) have limited cohesion, some cohesive strength is often still present. In many instances, cohesion accommodates over 50% of shear strength in some instances of small landslides ($D < 1$ m, Figure 3). However, this proportion of strength does not reflect large magnitudes of cohesion owing to small thicknesses and shear stresses. A mean cohesion of 0.83, 4.93 and 25.2 kPa is associated with thicknesses of $D < 1$ m, $1 \leq D < 10$ m, and $D \geq 10$ m, respectively. At shallow landslide depths, these cohesive strengths may result from the influence of lateral root reinforcement. As cohesion is back-analyzed uniformly over the entire rupture surface in this analysis, the influence of possible vegetation is evaluated with back-analyses with a variety of root cohesion at shallow depths (0.5 m) conditions reflecting forest management practices from Schmidt et al. (2001), shown in Figure SI.15. The presence of lateral root cohesion greatly reduces back-analyzed mineral cohesion at modest depths ($D < 1$ m), but much less so for moderate and only modestly so for landslides with significant thickness (i.e., bedrock landslides). This suggests that that equivalent cohesion values back-analyzed at larger depths or thicknesses (applied over the entire shear surface) are greater than equivalent root cohesion only applied to the near-surface. At shallow depths (i.e., $D < 1$ m), equivalent cohesion values are at the lower-bound of “root cohesion” values observed in the Oregon Coast Range (Schmidt et al., 2001) and may owe to the aforementioned censoring of smaller landslides; however, it has been suggested that underlying assumptions that describe root cohesion values may overestimate the true strength from roots by as much as 75% (Pollen et al. 2005, Cohen et al. 2011, Giarossich et al. 2019), and potentially even more when considering sparse vegetation and the spatial heterogeneity of roots (Sakals and Sidle 2004).

The presented trends do not contradict that mineral and root cohesion can be large in the soil mantle, and that friction may increase with landslide thickness; rather, they reflect a first-order set of trends that reflect the conditions that represent failure and the associated localized weaknesses that yield failure. For example, studies have made interesting observations that when a threshold slope gradient is exceeded in soil-mantled landscapes, there is less frequent shallow landsliding owing to cohesive boundary forces stemming from vegetation and/or mineral cohesion (Prancevic et al. 2020). It is suggested that beyond these threshold slope gradients, soil erosion in these environments may owe more to creep, which prevents requisite thickening of the soil mantle for landsliding. The censoring of small landslides in these inventories may indeed show a strong cohesive control at very shallow depths. Other studies have described that friction angle may increase with depth, at least in the soil mantle (e.g., Lu and Godt 2008). These strengthening factors may be dominant in many landscapes, including the areas described herein. However, the landslide behaviors presented in this study do not refute those observations; rock mass may have very large friction angles, cohesion stemming from mineral behavior, roots, and capillary stresses may be significant at shallow depths in soil. This study focuses on landslides that have occurred and do not necessarily represent complete landscape strength conditions; rather, the landslides used to construct relationships here may reflect the exception: localized strength conditions that result in landsliding (i.e., low root density or mineral cohesion,

weakened and slickensided fractures in soft rock mass with partially intact rock bridges – all of which result in landslide erosion). Still, this analysis does provide perspective as to the exchange of strength within a gradient of weathering at depths reflective of soil and particularly bedrock.”

- Kemeny, J. (2003). The time-dependent reduction of sliding cohesion due to rock bridges along discontinuities: a fracture mechanics approach. *Rock Mechanics and Rock Engineering*, 36(1), 27-38.
- Roering, J. J., Schmidt, K. M., Stock, J. D., Dietrich, W. E., & Montgomery, D. R. (2003). Shallow landsliding, root reinforcement, and the spatial distribution of trees in the Oregon Coast Range. *Canadian Geotechnical Journal*, 40(2), 237-253.
- Sakals, M. E., & Sidle, R. C. (2004). A spatial and temporal model of root cohesion in forest soils. *Canadian Journal of Forest Research*, 34(4), 950-958.
- Schmidt, K. M., Roering, J. J., Stock, J. D., Dietrich, W. E., Montgomery, D. R., & Schaub, T. (2001). The variability of root cohesion as an influence on shallow landslide susceptibility in the Oregon Coast Range. *Canadian Geotechnical Journal*, 38(5), 995-1024.
- Leps, T. M. (1970). Review of shearing strength of rockfill. *Journal of the Soil Mechanics and Foundations Division*, 96(4), 1159-1170.
- Giadrossich, F., Cohen, D., Schwarz, M., Ganga, A., Marrosu, R., Pirastru, M., & Capra, G. F. (2019). Large roots dominate the contribution of trees to slope stability. *Earth Surface Processes and Landforms*, 44(8), 1602-1609.
- Barton, N. (1976, September). The shear strength of rock and rock joints. In *International Journal of rock mechanics and mining sciences & Geomechanics abstracts* (Vol. 13, No. 9, pp. 255-279). Pergamon.
- Barton, N. (2006). *Rock quality, seismic velocity, attenuation and anisotropy*. CRC press.
- Barton, N. (2013). Shear strength criteria for rock, rock joints, rockfill and rock masses: Problems and some solutions. *Journal of Rock Mechanics and Geotechnical Engineering*, 5(4), 249-261.
- Barton, N. (2016). Non-linear shear strength for rock, rock joints, rockfill and interfaces. *Innovative Infrastructure Solutions*, 1(1), 1-19.

Specific comments

R2.4. L15: The 1st sentence, is stated as a fact but I think this is really a hypothesis.

A fair point – many thanks. This has been revised as:

“Distributions of landslide size are hypothesized to reflect hillslope strength, and consequently weathering patterns.”

R2.5. L29-30: do gentle inclinations accompanied with ‘proportionally lower frictional resistance’ guarantee a need for ‘compensation from cohesion’. The sentence reads as though one is a logical consequence of the other.

Agreed, this series of statements is indeed rather dense. We have broken this up and clarified the text to address this comment as well as the following two comments. Thank you for the suggested clarification – it has been revised as:

“We show that frictional resistance is the dominant strength control for most landslides in the study area although the proportion of contribution from cohesion increases with landslide thickness, consistent with deep-seated slope failures, typically in bedrock, that occur on gentle inclinations. We posit that the observed relationships between strength and landslide thickness are associated with the coalescence of zones of low shear strength driven by weathering, which constitutes a first-order control on the mechanical behavior of underlying soil and rock mass.”

R2.6 L30: ‘requiring’ what is it required in order to achieve? Stability?

This has been reworded – please see previous response.

R2.7 L29-31: The argument of the sentence appears circular, it starts with landslide depth, associates depth with properties then finishes by saying that these properties result in the observed depth.

Please see the response before the previous comment for the rewording.

R2.8 L32: not clear which particular ‘strength relationships’ you mean.

Agreed. Revised as:

“We posit that the observed relationships between strength and landslide thickness are associated with the coalescence of zones of low shear strength driven by weathering, which constitutes a first-order control on the mechanical behavior of underlying soil and rock mass.”

R2.9 L33: not clear what you mean by ‘geomorphic scaling laws’.

Apologies for any lack of clarity. When we use the expression “Geomorphic scaling laws,” we intend to describe scaling laws between landslide area and volume or landslide magnitude-frequency relationships. All descriptions of “geomorphic scaling laws” has been revised so they are clearer.

R2.10 L38: ‘landslide size distributions are a reflection of weathering patterns at a landscape scale’. This is too strong. It is stated as a fact but there is not consensus on this. Tanyas (2018) gives a nice review of many of the possible explanations.

Indeed, this is a fair statement. This introductory sentence has been revised as:

“Landslide size distributions are fundamental for evaluating erosion patterns, landscape evolution, soil production, transport and sequestration of organic carbon, and hazard prediction (Clarke and Burbank 2010, Gallen et al. 2015, Medwedeff et al. 2020; Li and Moon 2021, Frith et al. 2018; Korup et al. 2007; Larsen, Montgomery, and Korup 2010).”

R2.11 L38: It would be useful to start by defining what you mean by landslides in this paper what are the limits to your definition in terms of speed, size, and movement mechanics? This will be important when you seek to claim that your landslide inventories are representative.

An excellent suggestion – we have added the following to the introduction:

“The wide array of proposed scaling relationships reflect a variety of definitions (e.g., scar, deposit, both), including for different landslide mechanisms, and movement mechanics – in this study, we use both inferred source and deposit area as metrics for inversion of combined strength properties and scaling of inventoried landslides. ”

and:

“Landslides in the inventories are classified using the Varnes (1978) system, where approximately 19.8% of features were classified as being comprised of soil and/or debris (herein termed “soil” for the rest of this study; 5.6% as earth/debris translational or rotational slides, 14.2% as earth flows), 44.5% classified as bedrock (36.9% as rock translational or rotational slides, 7.6% as rock flows), and 39.5% were categorized with complex landslide movements. ”

R2.12 L38: When you talk about landslide size do you mean scar alone or scar and runout combined?

A fair point – clarified in the previous comment.

R2.13 L58-63: Some researchers continue to argue that censoring is responsible for the rollover.

An excellent point – this element should certainly not be ignore. Revised on L57-62:

“Thus, prior research has tended towards exploring various combinations of cohesion and friction (Katz and Aharonov 2006; Stark and Guzzetti 2009; Medwedeff et al. 2020, Bunn et al. 2020), disturbance and forcing conditions (Pelletier et al. 1997; Gallen et al. 2015), censoring and mapping subjectivity (Tanyas et al. 2019, Bernard et al. 2021), local geomorphic features (Schmidt and Montgomery 1995; Larsen et al. 2010; Frattini and Crosta 2013; Jeandet et al. 2019; Tanyas et al. 2021), as they relate to characteristic distributions of landslide size and frequency.”

And L136-139:

“The presence of distinct rollover points may reflect censoring of smaller landslides (typically soil), as described in several relevant studies (e.g., Bernard et al. 2021, Tanyas et al. 2019); this censoring is likely reflected in some of the relatively modest morphological differences between landslides classified as “soil” or “bedrock” in this study.”

R2.14 L67: ‘long been associated with...’ I think this claim needs citations to support it.

This has been revised with a citation as:

“Trends in scaling laws, typically landslide area-volume (A-V) and frequency-area relationships, have associated with specific parent materials (e.g. Larsen et al. 2010)...”

R2.15 L71-74: I don’t think this detail is necessary.

This sentence has been shortened as:

“A variety of scaling parameters have been proposed for specific landslide settings and parent materials (e.g., Hovius et al. 1997; Korup 2006, Guzzetti et al. 2009, ten Brink et al. 2009).”

R2.16 L74-75: This clause is out of place, it relates to the next sentence.

Appertaining to the previous comment, this statement has been removed.

R2.17 L77-78: How many of the distributions cited here are for landslide scars only?

A fair question, again emphasizing the importance of distinguishing the specific representation of landslide area (i.e. deposit/runout only, deposit/runout+scar, or scar only). It seems that some of these studies are based predominantly on landslide scars (e.g. Hovius et al. 1997, seemingly Star and Hovius 2001) while others consider scar and deposit (Malamud et al. 2004) or just deposits. Based on this excellent point of clarification, we have rerun the analysis and used only the scar area (called landslide source area, which represents the unevacuated, reconstructed landslide mass placed into the representative area that would be a scar during full evacuation), to be consistent with other studies (e.g. Malamud et al. 2004, Larsen et al. 2010 which compiled databases of deposits, scars and deposits+scars). Revised on L75-78:

“...in this study, we use both inferred source and deposit area as metrics for inversion of combined strength properties and scaling of inventoried landslides. .”

And L156-158:

“As conventional scaling laws are typically based on data either from deposits or scars, we choose source areas (i.e., landslides reconstructed to their unevacuated scars with inferred rupture surfaces) to evaluate relationships between landslide thickness and strength throughout this study.”

R2.18 L84: what does relevant mean in this context?

This is indeed unclear - many thanks for the suggestion. This sentence has been revised as:

“We posit that there is an association between landslide thickness and the patterns of governing mechanical strength properties of soil, saprolite, and rock, which reflect the rate and magnitude of dominant subsurface weathering processes within a landscape.”

R2.19 L85: mapped landslides: what are the mapping criteria and methods used to map these landslides? What is the definition of landslide used by the mappers? What do the shapes outline (scar or scar and runout)? These details will influence your sample of landslides, the particular properties of this sample is very likely to exert a strong influence on your back-calculated strength-depth relationships.

This certainly should be clarified – thank you for the suggestion. The landslides were inventoried as polygons of headscarp (i.e., the daylighted portion of a landslide scar) and flanks and deposits using bare earth lidar hillshade, DEM, and a variety of DEM derivatives as per the guidelines of Burns and Madin (2009). Basic features used in mapping include distinct topography associated with a headscarp, internal scarps, material dislocated downslope inferred to be a deposit, and a landslide toe. Landslides in this inventory are classified using the Varnes (1978) classification system and contain a variety of metadata, including landslide type and inferred material (e.g. translational rock landslide, rotational earth landslide, complex movement, etc.), lithology, etc. The following text has been added to the “Methods” section:

“This landslide inventory was mapped by the Oregon Department of Geology and Mineral Industries (DOGAMI) using identification of distinct topography associated with a headscarp, internal scarps, material dislocated downslope (deposits) and a toe based on interpretation of bare earth lidar and its derivatives. A given landslide was mapped using polygons representing the interpreted headscarp and associated deposits. The landslide inventory was classified according to the Varnes (1978) classification system (Burns and Madin 2009). The approximate headscarp height and topographic texture were used by the mappers to define material as rock, debris or earth – classifications consistent with the Varnes classification system. This landslide inventory contains metadata for most mapped landslides, including inferred landslide mechanism, material type, and lithology. More details on the mapping process can be found in the protocols outlined by Burns and Madin (2009). All landslide inventories used in this analysis were mapped using 0.9-meter lidar-derived digital elevation models (DEMs) from the Oregon Lidar Consortium (OLC), and certified by the DOGAMI geologists. Based on the mapping protocols, Burns and Madin (2009) state that the smallest mappable landslide was 100 m² in planform area.”

Burns, W. J., & Madin, I. (2009). Protocol for inventory mapping of landslide deposits from light detection and ranging (LiDAR) imagery.

R2.20 L92-96: This sentence is very difficult to understand it lists multiple interacting components but doesn't explain how they interact. It also relies on results that the reader hasn't yet seen so my suggestion would be to remove it.

Apologies for the lack of clarity – this sentence has been removed.

R2.21 L98: I'm not sure what you mean by 'adverse geologic conditions'.

This is unclear. Many thanks. This has been revised as:

“Steep terrain, strong but infrequent earthquakes, weak geologic conditions (e.g., sedimentary rock), and considerable precipitation result in extensive landsliding in western Oregon, USA (Goldfinger et al. 2012, Lahusen et al. 2020).”

R2.22 L104: How were landslides classified as soil or bedrock?

This is a necessary classification. As briefly described in response R2.19, this inventory was created by expert mappers (DOGAMI), who used the Varnes (1978) landslide classifications (earth, debris and rock) for classification of landslide parent material. These classifications were objectively assigned based on a threshold headscarp height (>4.5m) and subjectively based on deposit and toe texture. In some cases, landslides had a field check to confirm parent material classifications. While the use of headscarp height (and relatively large headscarp heights) as a demarcation between soil and rock landslides is subjective, it does adhere to a consistent procedure (with its biases). In the revised study, we focus less on the subjective classifications and focus more on trends between back-analyzed mean landslide thicknesses likely as a more objective representation of landslide parent material. We consider landslides comprised of “earth” or “debris” to be soil, while “rock” is bedrock.

“The approximate headscarp height and topographic texture were used by the mappers to define material as rock, debris or earth – classifications consistent with the Varnes classification system.”

R2.23 L108: More information is needed on these inventories. What was their definition of a landslide? How small were the smallest landslides they could resolve? How did they deal with scar and runout? How do you deal with rockslide and rockfall? Bernard et al. (2021) show that if these rockfalls may be severely undersampled in existing inventories and can considerably alter the size distribution.

Bernard, T.G., Lague, D. and Steer, P., 2021. Beyond 2D landslide inventories and their rollover: synoptic 3D inventories and volume from repeat lidar data. *Earth Surface Dynamics*, 9(4), pp.1013-1044.

This certainly should be clarified – thank you for the suggestion. The landslides were inventoried as polygons of headscarp and deposits using bare earth lidar hillshade, DEM, and a variety of

DEM derivatives as per the guidelines of Burns and Madin (2009). Basic features used in mapping include distinct topography associated with a headscarp, internal scarps, material dislocated downslope inferred to be a deposit, and a landslide toe. Landslides in this inventory are classified using the Varnes (1978) classification system and contain a variety of metadata, including landslide movement classification and in most instances inferred material (e.g. translational rock landslide, rotational earth landslide, complex movement, etc.), lithology, etc. Classifications such as rockfall, topples, debris flows, and spreads are neglected in these analyses. The following has been added to the “Methods” section:

“This landslide inventory was mapped by the Oregon Department of Geology and Mineral Industries (DOGAMI) using identification of distinct topography associated with a headscarp, internal scarps, material dislocated downslope (deposits) and a toe based on interpretation of bare earth lidar and its derivatives. A given landslide was mapped using polygons representing the interpreted headscarp and associated deposits. The landslide inventory was classified according to the Varnes (1978) classification system (Burns and Madin 2009). The approximate headscarp height and topographic texture were used by the mappers to define material as rock, debris or earth – classifications consistent with the Varnes classification system. This landslide inventory contains metadata for most mapped landslides, including inferred landslide mechanism, material type, and lithology. More details on the mapping process can be found in the protocols outlined by Burns and Madin (2009).”

The smallest landslide that could be resolved as considered to be 100 m² in planform area. The potential censoring of this has been discussed in proximity to figure 1:

“The presence of distinct rollover points may reflect censoring of smaller landslides (typically soil), as described in several relevant studies (e.g., Bernard et al. 2021, Tanyas et al. 2019); this censoring is likely reflected in some of the relatively modest morphological differences between landslides classified as “soil” or “bedrock” in this study.”

And in the methods as:

“Based on the mapping protocols, the smallest mappable landslide was 100 m² in planform area (Burns and Madin 2009).”

R2.24 Fig 1b: There are surprisingly few small landslides in this inventory. How much censoring do you expect in your mapping and at what size?

Many thanks. Please see the above response and revisions.

R2.25 L123: ‘rupture surface fitting’ introduces a dependence on both resolution and precision of topographic data. How does this censor your landslide distribution?

This is an excellent question. DOGAMI limited their minimum mappable landslide size to 100 m². We have rerun the analyses after sensitivity studies with various resolutions, and evaluated sensitivity of back-analyzed strength to resolution. We found that there tends to be limited

sensitivity (less than 0.1° change in back-analyzed friction angles) for small landslides when using the native DEM resolution (0.91m) and at least 100 cells were included in the deposit and headscarp. Thus, our smallest landslide used in the analysis is 121m^2 . Of course, the resolution limits the size of the smallest landslides that can be analyzed. Added to the methods section:

“To adequately reflect various landslide sizes and maintain computational expediency, we use variable resolutions that ensure a minimum of at least 100 cells in the smallest mapped landslide (a sensitivity analysis of back-analyses was observed to be insensitive past this limit) and no smaller than the native DEM resolution (0.91m). Similar tests showed limited back-analysis sensitivity for the biggest landslides when resolution was less than 6m. Based on rupture surface fitting, the smallest landslide deposit area that could be resolved using these techniques was 121m^2 . These limits on landslide area censor smaller landslides and likely influence observed magnitude-frequency relationships as well as the boundaries of observed strength exchange. Examples of landslide geometries of varying sizes are shown in Figure SI.4.”

R2.26 L126: the ‘example of reconstruction’ is useful, you should also include an example for one of the smallest landslides in your inventory to demonstrate (qualitatively) that it is appropriate to apply these techniques at this scale.

We have revised the example figure in SI to have several examples at different landslide sizes.

R2.27 L142: I don't understand what ‘sorted’ means in this context. Do you mean the distributions differ between soil and bedrock landslides? Or are you being more specific e.g. they are shifted?

Apologies for being unclear – this was a poor choice of wording. Based on reviewer comments, this specific sentence has been removed and reworked into the narrative and SI figures.

R2.28 L150: How confident can you be that you have captured the geometry of ‘incipient failure’ for your landslides? Is there a size (and therefore depth) dependence to this confidence? How does uncertainty in incipient failure propagate into your results?

This is an excellent question, and surely deserves clarity. In the absence of a landslide inventory with both pre- and post-event digital elevation models, there is of course exact means of ensuring that the reconstructed geometry is exact. However, as (1) this model is an approximation (like all models), and (2) the presented inventory does not represent failures from a singular event (a limitation of *many* landslide inventories), we rely on indirect metrics for confidence. Namely, we use the observation from Larsen et al. (2010) that scars (i.e. reconstructed areas and volumes) and deposits (i.e. existing areas and volumes) may demonstrate statistically indistinguishable power-law fits for area-volume relationships. Larsen et al. (2010) explicitly state this for bedrock landslides, but also show this for soil landslides (in SI Table 1 from their study). As we have now consider deposit area only (i.e. no headscarp area) in the case of deposits, and scar area only for reconstructed landslides (i.e. area where reconstructed columns have more than zero thickness, effectively the landslide scar), we see statistically indistinguishable fitting exponents between reconstructed landslide scars and deposits ($\gamma=1.46\pm 0.007$ and $\gamma=1.467\pm 0.008$ for scars and reconstructed landslides, respectively) when omitting classification as soil or rock (i.e. “all landslides”). While this is a broad check, we consider this agreement with other findings to suggest that our approximation of reconstructed failure geometry is reasonable.

In these revised analyses, we also consider bulking as a metric for evaluating reasonableness. We consider landslides with ratios of deposit volume to source volume greater than 0.5 (significant erosion of material owing to landslide deposits being potentially old and eroded with subsequent failure and other erosional processes) and 5.5 (the 90% confidence interval for documented bulking from a sensitivity analysis from Larsen et al. 2010) to be reasonable bounds on relevant landslides, leaving approximately 7,300 landslides for analysis with a median bulking ratio of 1.28. While somewhat arbitrary, it does provide a reasonableness check on capturing incipient failure. To test sampling bias, we also show the strength relationships for all data without exclusion of bulking or “evacuation” constraints in Figure SI.20 – the observed trends are similar to those in the narrative.

Added to lines 144-155:

“Source and deposit volumes show similar scaling (Figure 1), maintaining a power-law exponent of $\gamma\approx 1.46$ and 1.47 and slightly different intercepts of $\alpha\approx 0.033$ - 0.036 , respectively. Scars and deposits may have similar scaling exponents, as discussed in previous studies (i.e., Larsen et al. 2010). There is an evident sorting of data by mean landslide inclination (θ), whereas there is an apparent inverse correlation between inclination and area (Figure 1a and 1b) and consequently, thickness. As expected, landslides reconstructed to their source area demonstrate steeper mean inclinations. Power-law fits of area and volume for landslide deposits and sources show different exponents for different movement mechanisms and material classifications (i.e., bedrock landslides with $\gamma\approx 1.40$ - 1.42 ; soil landslides $\gamma\approx 1.39$ - 1.42 , complex movement $\gamma\approx 1.46$ - 1.47 , Figure SI.5). Scaling coefficients for soil

rotational/translational landslides are similar to those observed in Oregon and Washington in previous studies (Larsen et al. 2010). However, the differences in area-volume relationship between landslide materials lie within 95% confidence intervals of one another, precluding differences of statistical significance.”

Added to the methods section:

“Results excluding these values are shown in Figure SI.19 and show similar trends to those in the narrative. We used 7,300 landslides from twelve different inventories that met a variety of criteria necessary for this analysis (Figure SI.1). Namely, this analysis required inventoried landslides that contained headscarp and deposit polygons and that contained sufficient metadata as to exclude falls, topples, and debris flows, which are mechanisms where the proposed methods would not appropriately capture failure kinematics. Further, we exclude data where estimates of volume change between source and deposits are poor – only considering bulking ranging from 0.5 (50% of source material has been eroded, possible in such landscapes) and 5.5 (90% confidence for bulking, Larsen and Montgomery 2010). As many landslides are likely ancient, sufficient erosion (through repeated failure or other geomorphic processes) could result in diminished post-failure volume and served as the lower bound for the bulking threshold. Large bulking can occur from entrainment of downslope debris. We test any bias from potential change in density from bulking in Figure SI.19 and similar trends in strength with thickness hold. As evacuation may be a bias for the trends (i.e., low friction angles for long runout landslides), we place a minimum areal overlap threshold for deposit and source area versus total area (deposit and headscarp are combined) of 25% to reflect the lower bounds of non-evacuative landslides. This approach was deemed reasonable as most landslides did not exhibit complete evacuation based on areal extents (median areal overlap of deposit and source proportional to total area was 58%). To ensure that no sampling bias is introduced, we evaluate the aforementioned strength relationships without sampling thresholds (Figure SI.20), which show similar trends to those already presented. ”

Larsen, I. J., Montgomery, D. R., & Korup, O. (2010). Landslide erosion controlled by hillslope material. *Nature Geoscience*, 3(4), 247-251.

And the following Figure in SI:

Figure SI.20. (a) Failure envelope, (b) strength versus effective normal stress, (c) strength relationships with mean landslide thickness for all landslides without exclusion based on bulking (0.5-5.5) or overlap of source and deposit areas (0.25-1). The observed nonlinearity of the failure envelope and trends between strength and normal stress or landslide thickness are similar to those presented in the paper, suggestive of limited sampling bias.

R2.29 L164: ‘Soil landslides demonstrate median thicknesses of 1.9-2.1m’: These are extremely thick for soil landslides suggesting under-sampling of smaller shallower landslides in the inventory.

This is a fair point, and potentially owes to an assumption by the mappers introduced during inventory creation. The mappers (Oregon Department of Geology and Mineral Industries, Burns and Madin 2009) used several factors for classification of soil or bedrock landslides, including headscarp height (<4.5 m is “soil”), texture and morphology (expert judgment), and in some cases, field verification. We do not use headscarp height as a direct measurement in mapping rupture surfaces or associated mean depth; however, we do rely on the associated Varnes classification used by the mappers to distinguish between soil and bedrock landslides. The relatively deep headscarp height (again, not a measurement of mean depth) that dictates the classifications assigned in the inventory could potentially and indirectly bias the observed mean thicknesses. Many large landslide inventories have subjective elements when it comes to classification (mechanism, parent material) – this inventory is no exception. Further, we dispute how binary any classification system is for landslide inventorying – for example, is saprolite or weathered rock or soil? Many of the landslides in this inventory are in weak, sedimentary rock – such materials exist at the gradient between acting as soil or rock. No matter the inventory, we are betrothed to the subjective interpretation of mapping experts, and such is the basis for numerous studies based on landslide inventories. Lastly, we note that with the new constraints on evacuation, we have median soil thicknesses of 0.9-1.2m.

We have added the following text:

“Median thickness for soil/debris and bedrock translational and rotational landslide sources are 0.92-1.17 m and 2.54-2.59 m, respectively. The observed median soil thicknesses are consistent with field

observations (e.g., Heimsath et al., 2001). Nonetheless, potential censoring of smaller landslide sizes (Hovius et al. 1997, Stark and Hovius 2001, Brardinoni and Church 2004, Tanyas et al. 2019), and/or a bias in the original classification procedure of the landslide inventory (Burns and Madin 2009) is likely present.”

R2.30 L166: ‘friction angles are a strong covariate with mean landslide inclination’. Isn't this unavoidable given the method that you use to estimate friction angle? You should certainly including scatter plots of friction angle and mean landslide inclination.

Perhaps this is an obvious covariate and can certainly be attributed to the back-analysis procedure. However, we disagree that this is far-fetched as (1) this analysis considers three-dimensional effects (e.g. altered boundary resistances owing to forces that do not align with a direction of sliding) that could reduce overall strength needed for stability, and (2) these friction angles can deviate quite a bit from mean slope inclination for deposits and especially when compared to source mean landslide inclinations. The figure below shows this scatter. We have plots of both mean landslide inclination and friction angle in the revised Figure 2.

Left: Mean deposit inclination versus friction angle. Right: Mean landslide source inclination versus friction angle. The black line represents an idealized relationship for infinite slope conditions under $m=0.5$ conditions.

R2.31 L203-204: (e.g. “angle of repose”, Alejano and Alonso 2005). Alejano and Alonso 2005 don’t discuss ‘angle of repose’.

This is a fair point, although this study does describe dilation, which is conceptually similar. We have heavily reworked this section, as shown in response R2.3.

R2.32 L214-215: ‘supporting arguments that’: citations needed for the arguments you refer to here.

Thank you. This sentence has been removed as a result of previous reviewer comments.

R2.33 L248-254: ‘decreasing landslide inclination with larger landslide area results in two interdependent stress requirements for failure’: An increased thickness required to overcome cohesive strength makes sense but it is not clear why this follows from the preceding requirements: the first requirement (frictional resistance reducing so that cohesion can increase) isn’t a requirement it is a restatement of your finding; the second (shear stress being sufficient large to overcome cohesion) is guaranteed from back-calculation.

A fair point – this has been reworded as:

“This trend of decreasing landslide inclination (Figure 1) with increasing landslide thickness results in two interdependent stress conditions associated with back-analyzed, incipient failure from a landslide source – (1) frictional resistance must be sufficiently reduced as to result in an excess of shear stress that is accommodated by cohesion, and (2) shear stresses must be sufficiently large to overcome the cohesive resistance associated within the rock mass, requiring sufficient landslide thickness.”

R2.34 L298: ‘The attenuation of frictional strength with landside area and consequently depth...’: none of the explanations that follow involve area, all involve depth; while you can use area, depth or volume as your independent variable, only depth is a candidate for mechanistic explanations. In your method area controls depth (through spline and inpainting) but you don’t explain the physical connection between area and depth.

Indeed, this statement and other comments have helped us focus this study more on thickness. We have reworked the entire manuscript to focus on mean landslide thickness and depth as it better relates to weathering.

R2.35 L330-331: I suggest removing landslide relief it is clearly a result of the increased length scale over which relief is being measured as area increases (demonstrated by the reduction in inclination with area).

We appreciate this suggestion – we have removed this.

R2.36 L421: It is not clear what you mean by ‘residence times’ here.

Thank you for the suggested clarification. This sentence has been removed as a byproduct of the revisions of the discussion in this section.

R2.37 L427-428: Don’t these ‘other strengthening factors, such as vegetation and partial saturation’ provide strength as cohesion?

Yes, this is true, but the importance of these factors at the time of triggering and at these scales is uncertain. Further, *we do* observe shallow, steep landslides that exhibit cohesion in this database – alternatively, some thick landslides have low proportional cohesion from our analysis. That is, there is no binary association of strength with different landslide sizes, simply regional-scale trends.

Shallow landslides may occur in a partially saturated state (Godt et al. 2009), but they often occur after intense storms and high antecedent moisture, particularly from the presence of a transient, perched water table (Sidle and Swanston 1982, Reid et al. 1998); thus, it is reasonable to expect that for many shallow landslides, ignoring matric suction stemming from partial saturation is a reasonable assumption.

We also explore cohesion magnitudes and rerun models with root cohesion, which may be overestimated, as described in R2.3.

Reid, M. E., Nielsen, H. P., & Dreiss, S. J. (1988). Hydrologic factors triggering a shallow hillslope failure. *Bulletin of the Association of Engineering Geologists*, 25(3), 349-361.

Sidle, R. C., & Swanston, D. N. (1982). Analysis of a small debris slide in coastal Alaska. *Canadian Geotechnical Journal*, 19(2), 167-174.

Godt, J. W., Baum, R. L., & Lu, N. (2009). Landsliding in partially saturated materials. *Geophysical research letters*, 36(2).

R2.38 L456: It is not clear what you mean by flank polygons, it would help to label this on Fig SI4

Apologies for a lack of clarity. These polygons reflect the region of the landslide where the headscarp flanks the landslide deposit. A more straightforward term would be simply “headscarp”. This has been revised throughout the study. We have also revised figure SI.4 as requested.

Figure SI.4. Example of landslide rupture surface and topographic reconstruction technique. This specific example is a landslide with a source (scar) area and volume of approximately $4.81 \times 10^4 \text{ m}^2$ and $4.79 \times 10^5 \text{ m}^3$, respectively. **(Left)** Contours of existing topography (solid black contours), pre-landslide reconstructed topography (red dashed contours), inferred rupture surface geometry (blue dashed contours), and the inventoried landslide extent (thin, black dashed line). Contour elevations are presented in meters above sea level (m.a.s.l.). Existing surface topography with landslide extent marked by a red, dashed line. Reconstructed surface topography using curvature-preserving inpainting technique with landslide extent marked by red, dashed line. Mapped rupture surface topography determined from using a the thin-plate spline technique with landslide extent marked by red, dashed line and the scarp mapped by the green dashed line. **(Right)** Contours for a series of different landslide sizes reflecting existing topography, the inferred rupture surface and reconstructed topography.

R2.39 L457: How were the landslides identified in lidar? Can explain the method or provide a reference?

This certainly should be clarified – thank you for the suggestion. The landslides were inventoried as polygons of headscarp and flanks and deposits using bare earth lidar hillshade, DEM, and a variety of DEM derivatives as per the guidelines of Burns and Madin (2009). Basic features used in mapping include distinct topography associated with a headscarp, internal scarps, material dislocated downslope inferred to be a deposit, and a landslide toe. Landslides in this inventory are classified using the Varnes (1978) classification system and contain a variety of metadata, including landslide type and inferred material (e.g., translational rock landslide, rotational earth landslide, landslide complex, etc.), lithology, etc. The following has been added to the “Methods” section:

“This landslide inventory was mapped by the Oregon Department of Geology and Mineral Industries (DOGAMI) using identification of distinct topography associated with a headscarp, internal scarps,

material dislocated downslope (deposits) and a toe based on interpretation of bare earth lidar and its derivatives. A given landslide was mapped using polygons representing the interpreted headscarp and associated deposits. The landslide inventory was classified according to the Varnes (1978) classification system (Burns and Madin 2009). The approximate headscarp height and topographic texture were used by the mappers to define material as rock, debris or earth – classifications consistent with the Varnes classification system. This landslide inventory contains metadata for most mapped landslides, including inferred landslide mechanism, material type, and lithology. More details on the mapping process can be found in the protocols outlined by Burns and Madin (2009). All landslide inventories used in this analysis were mapped using 0.9-meter lidar-derived digital elevation models (DEMs) from the Oregon Lidar Consortium (OLC), and certified by the DOGAMI geologists. Based on the mapping protocols, Burns and Madin (2009) state that the smallest mappable landslide was 100 m² in planform area.”

Burns, W. J., & Madin, I. (2009). Protocol for inventory mapping of landslide deposits from light detection and ranging (LiDAR) imagery.

R2.40 L457: What metadata were used to exclude landslide complexes?

We apologize – this was actually a mistake in our original interpretation of the landslide classification. We originally interpreted “Complex” to reflect a landslide complex (i.e. an amalgamation/coalescence of multiple landslides). These are actually complex movements (i.e. from the Varnes classification) and thus relevant to this study. Thus, we have updated the paper so that these relevant data are now used in the analysis. This classification is based on expert geologic mapping according to the protocols of Burns and Madin (2009), which was used by the mappers (DOGAMI) to create the inventory.

R2.41 L460: ‘mapped using 0.9-meter lidar-derived DEMs’: What censoring do you expect in your inventory on this basis?

An excellent question – we have added the following:

“Based on the mapping protocols, Burns and Madin (2009) state that the smallest mappable landslide was 100 m² in planform area.”

R2.42 L470: What is the minimum landslide area for which the method can resolve a rupture surface geometry? Did you test this for some of your smallest landslides?

Building on the response to R2.25, we have tested the fit on smaller slides and used other requested metrics (i.e., overlap to evaluate evacuation) to constrain reasonable surfaces. An example is shown in Figure SI.4. Ultimately, this approach provides a first-order estimate and we acknowledge this in the discussion of limitations.

“As rupture surfaces are a modeling output and ultimately, a first-order estimate of landslide geometry...”

R2.43 L481: What length scale is curvature calculated over? I guess uncertainty in this length parameter has little impact on your findings but it should probably be recognised.

The length scale is the same as that of the rupture surface fitting technique (0.91m-3.0m). This has been revised as:

“...under a simple plate constraint that adheres to the diffusion equation ($\partial z^2 / \partial^2 x + \partial z^2 / \partial^2 y = 0$), computed using a finite difference solver over the landslide DEM grid.”

R2.44 L481: Why is preserving and propagating curvature ‘key for infilling landslide scars’?

Thank you for the suggested clarification – we appreciate it. We have reworded this statement as:

“These techniques have shown great potential to preserve and propagate curvature (Chan and Shen 2001), which serves as a means of reconstructing topography based on surrounding, presumably unfailed terrain”

R2.45 L491-492: It is not clear to me that ‘existing landslide topography’ and ‘deposits are rest’ are equivalent. Two examples that I have seen at many landslides are: 1) The landslide topography will include part of the scar that is now free of deposit; 2) the deposit within the scar is discontinuous and patchy; 3) the deposit extends far beyond the scar (e.g. due to avalanche or flow-like runout). How do you handle these situations?

As the reviewer has mentioned, these scenarios would be difficult (if not impossible) to handle using the proposed approach. However, based on indirect consideration a variety of morphological metrics described in previously, we do not have a reason to believe that a large proportion of the analyzed inventory falls into these scenarios.

Added to the limitations and uncertainties section:

“As rupture surfaces are a modeling output and ultimately, a first-order estimate of landslide geometry used to evaluate strength controls on instability, there is no direct means of directly isolating the observed trends in strength to ignore a bias from landslides with extensive evacuation and runout. We consider simple criteria by which landslides are evacuated, including the proportional overlapping deposit and modeled source area versus total landslide area (deposit and headscarp). We observe that the landslide inventory herein has a median areal overlap of 58% between source and deposit, which suggests that full evacuation for the landslides used in this analysis may not exert a strong bias associated with long runout landslides. We select a 25% overlap as an lower bound on non-evacuative landslides (approximately the 10th percentile for overlap). The role of evacuation may control the level of stability of landslide deposits, which are used to determine the friction angle in this study. That is, landslide deposits are treated to be a state of limiting equilibrium associated with continued activity. This is a significant and necessary assumption for the proposed inversions but has a basis in reality as landslide deposits (even relict deposits) are often prone to reactivation and continued instability (Temme et al. 2020). We perform analyses to evaluate the level of stability of deposits considering back-analyzed friction angles from source areas using a factor of safety (FS), which for purely frictional materials reduces to $FS = \tan(\phi_{source}) / \tan(\phi_{deposit})$. Under these conditions, the median FS is 1.26, almost equivalent to specified FS values of designed, engineered slopes ($FS \approx 1.25-1.3$, Figure SI.16). We also investigate strength trends through application of a FS to deposit friction angles and subsequent recalculation of landslide source cohesion (Figure SI.16, Figure SI.17). We observe similar exchanges in proportional friction and cohesion, although the exchanges are offset.”

R2.46 L493: ‘most landslides did not exhibit complete evacuation’: What was your estimate for the proportion of landslides that did exhibit complete evacuation?

There is no direct way of knowing this, but the sensitivity analyses provided earlier in this review (R2.2) hopefully provide some insight towards this.

R2.47 L502: ‘less than 10%’: give the nearest integer value.

As described in the response to the major comment by the reviewer, 90% of the landslides show overlap of scar and deposits of 21% or greater.

L508: Equation 2: What assumptions are required to construct these equations for normal and shear stress? I was expecting a $\cos(\theta)$ in the pore pressure term. Are you assuming hydrostatic rather than slope parallel seepage? If so, why make that choice?

This is a fair point, and the assumptions should be explained more clearly. This text has been modified as:

“As normal and shear stresses may vary throughout the landslide slip surface, for comparative purposes we simplify these stresses into scalar values using mean landslide thickness for comparison on a landscape scale using Mohr-Coulomb relationships based on mean basal stresses. The effective strength parameters are representative of drained conditions (i.e., loading is slow and rate-dependent excess pore pressures are not appreciable, effective stress conditions are maintained). For simplification, scalar representation of landslide effective normal stress (σ') is treated as:

$$\sigma' = (\rho g D - \rho_w g D m) \cos \theta \quad (2)$$

accounting for slope-parallel seepage (per Reid 1997) and pore pressures are applied as a pressure within each landslide column as a function of saturated thickness and the inclination of the surface of each cell.”

We acknowledge that groundwater conditions are a major uncertainty, seepage conditions aside. We have modified the analyses to reflect slope parallel flow conditions, including modification of pore pressure ratios based on the approach proposed by Reid (1997).

Reid, M. E. (1997). Slope instability caused by small variations in hydraulic conductivity. *Journal of Geotechnical and Geoenvironmental Engineering*, 123(8), 717-725.

R2.49 L512: I’m not clear what you mean by ‘arbitrary landslide inclinations’ here.

The word “arbitrary” has been removed to reduce confusion.

Once again, we would like to thank the reviewer for their extremely thorough and constructive review – not only has it helped us contextualize the findings and limitations of this study better, but it has helped us learn. It is very much appreciated.

Reviewer #3

This paper by Alberti et al., "Distributions of Landslide Size Controlled by Patterns in Hillslope Strength" examined the influence of the hillslope strength on the landslide size scaling relation with volume and frequency. They analyzed ~ 8,000 landslides, quantified strength metric – friction angle and cohesion in western Oregon, and showed how the strength estimation change with landslide size. This paper presented the applications of a novel 3D slope stability analysis on large actual dataset of landslides, and provide impressive amount of work on statistical tests and analysis. The 3D slope stability methods are based on the previously published studies but are applied to new large dataset in western Oregon. They show that friction and cohesion dominate soil and bedrock landslides, respectively, and hillslope strength affected by weathering patterns affect the geomorphic scaling law, especially the landslide frequency-area relationship.

The paper is well written and provides important results on demonstrating the weathering controls landslide scaling factors. However, some parts need to be improved by clarifying and presenting slightly differently before publication. The authors can improve the manuscript by 1) analyzing strength results separated for soil and bedrock landslides and 2) revising the discussion to acknowledge limitations of the methods and clarify the influence of "weathering." I recommend considering the publication after the significant revision of the paper. Below are the important comments on the paper.

The authors are sincerely grateful to the reviewer for their comments. They were insightful, constructive, and certainly helped us refine and hopefully improve this work. Of course, there are limitations in models, these analyses notwithstanding. We have attempted to better highlight uncertainties as well as isolate some of the aforementioned trends in context of parent material. Once again, we thank the reviewer for their excellent comments and feedback.

R3.1 First, the authors may consider currently presenting the strength estimation results separately for soil and bedrock landslides. I am wondering whether the relationship between strength and landslide sizes is somewhat influenced by mixing between abundances of soil vs. bedrock landslides for a given landslide size range. Smaller landslides are more soil landslides, and bigger landslides are more bedrock landslides. The increase of landslide sizes tends to have more bedrock landslides, so just higher cohesion values. The smooth transition in Figure 3 may not reflect the bedrock weathering degree difference but may show the relative mixing trends between those two landslide types. If bedrock weathering degree indeed influences the scaling factor, this trend should be evident when analyzed only for bedrock landslides.

Typically, soil landslides tend to occur in the soil-bedrock boundary, so the failure planes may likely represent soil depth variation. The soil depths are probably influenced by in-situ weathering but also can be affected by curvature, soil transport and thus landscape position, etc. If you separate the analysis, you may see some trends differ in bedrock and soil landslides. For example, the influence of cohesion in soil landslides is more significant with shallower depth. Because there may be different potential controls on soil and bedrock landslides, I suggest dividing the results for those and clearly showing the existence of trends in soil and bedrock

landslides separately. Adding the figures for only soil vs only bedrock landslides in the supplementary figure may be sufficient.

This is an excellent suggestion. We have presented trends for individual classifications of landslides, both with soil (i.e. earth or debris classification) and bedrock (i.e. rock) classifications, as shown in figure SI.10. As shown, the trends when isolating bedrock landslides (translation, rotational, and likely larger depths or areas of complex movements) are generally similar, although some minor differences exist owing to number of samples and expected censoring bias of undermapping small landslides (presumably comprised of soil).

Figure SI.10. Relationships between mean landslide thickness and strength for different subjective landslide classifications. Most subjective mechanism classifications demonstrate an exchange between frictional and cohesive strength with mean thickness, although the trends are noisy for the few samples for soil/debris slides. Most mechanisms demonstrate a decrease in friction angle with landslide thickness and a commensurate increase in cohesion. Complex movements tend to have mean thicknesses associated with bedrock (i.e. >1m) and a pronounced decrease in friction angle with thickness.

We have also added the following table SI2.a and SI2.b (please pardon the formatting in portrait page layout):

Table SI.2a. Wilcoxon rank sum test comparing friction angle distributions of different landslide classifications and their level of statistical significance of their differences. A (-) reflects no statistical difference under a 5% significance level, (*) represents statistical differences under a 5% significance level, (**) represents statistical differences under a 1% significance level, (***) represents statistical differences under a 0.1% significance level.

FRICTION ANGLE								
	All Landslides	Bedrock Translational Landslides	Bedrock Rotational Landslides	Soil/Debris Translational Landslides	Soil/Debris Rotational Landslides	Earthflows	Rockflows	Complex Movement
All Landslides	1	3.01×10^{-24} (***)	9.16×10^{-08} (***)	2.91×10^{-02} (*)	7.03×10^{-01} (-)	1.39×10^{-09} (***)	1.02×10^{-02} (*)	2.54×10^{-01} (-)
Bedrock Translational Landslides	-	1	1.03×10^{-32} (***)	4.63×10^{-03} (**)	2.77×10^{-04} (***)	2.66×10^{-30} (***)	7.40×10^{-07} (***)	1.29×10^{-25} (***)
Bedrock Rotational Landslides	-	-	1	3.76×10^{-05} (***)	3.41×10^{-01} (-)	1.07×10^{-01} (-)	3.87×10^{-08} (***)	4.45×10^{-05} (***)
Soil/Debris Translational Landslides	-	-	-	1	1.52×10^{-01} (-)	4.56×10^{-06} (***)	7.21×10^{-01} (-)	1.01×10^{-02} (*)
Soil/Debris Rotational Landslides	-	-	-	-	1	1.50×10^{-01} (-)	2.13×10^{-01} (-)	9.24×10^{-01} (-)
Earthflows	-	-	-	-	-	1	3.62×10^{-10} (***)	2.06×10^{-07} (***)

Rockflows	-	-	-	-	-	-	1	2.43×10-03 (**)
Complex Movement	-	-	-	-	-	-	-	1

Table SI.2b. Wilcoxon rank sum test comparing cohesion distributions of different landslide classifications and their level of statistical significance of their differences. A (-) reflects no statistical difference under a 5% significance level, (*) represents statistical differences under a 5% significance level, (**) represents statistical differences under a 1% significance level, (***) represents statistical differences under a 0.1% significance level.

COHESION								
	All Landslides	Bedrock Translational Landslides	Bedrock Rotational Landslides	Soil/Debris Translational Landslides	Soil/Debris Rotational Landslides	Earthflows	Rockflows	Complex Movement
All Landslides	1	1.08×10-02 (*)	5.16×10-01 (-)	4.92×10-31 (***)	1.17×10-08 (***)	7.30×10-27 (***)	6.11×10-06 (***)	1.68×10-39 (***)
Bedrock Translational Landslides	-	1	7.09×10-03 (**)	1.06×10-20 (***)	1.92×10-06 (***)	3.79×10-10 (***)	3.09×10-02 (*)	7.01×10-25 (***)
Bedrock Rotational Landslides	-	-	1	2.72×10-31 (***)	1.37×10-09 (***)	1.01×10-22 (***)	4.43×10-06 (***)	1.16×10-20 (***)
Soil/Debris Translational Landslides	-	-	-	1	8.63×10-02 (-)	1.03×10-06 (***)	6.96×10-13 (***)	8.77×10-54 (***)
Soil/Debris Rotational Landslides	-	-	-	-	1	9.83×10-02 (-)	3.29×10-04 (***)	2.19×10-17 (***)
Earthflows	-	-	-	-	-	1	1.52×10-03 (**)	6.86×10-68 (***)
Rockflows	-	-	-	-	-	-	1	2.61×10-26 (***)
Complex Movement	-	-	-	-	-	-	-	1

We have also added the following text to the manuscript:

“We find that there are trends in strength properties of landslides when compared to mean thickness, and to a modest level, between subjective landslide classifications. Translational landslides, both in rock and soil/debris, tend to show greater friction angles and lower cohesion than their rotational counterparts (Figure SI.9, SI.10). Bedrock landslides show greater median thickness and cohesion than soil landslides, and although differences of median friction angles when comparing soil/debris and rock classifications are modest, higher percentiles of friction angles of soil/debris landslides are notably larger than their bedrock counterparts (greater by ~2-5°). Landslides with complex movement have the largest median thickness and cohesion, the former suggesting a bias towards being composed of bedrock. Median friction angles vary by mechanism for the given conditions but tends to be highest for translational landslides. Differences in median friction angle between landslide classifications are modest (19-25°) and are a strong covariate with mean landslide surface inclination. However, differences in landslide inclination, thickness and strength are usually more pronounced at the extremes of their distributions (e.g., 90th percentile). For example, at the 90th percentile, bedrock landslides exhibit cohesion values three to four times larger than those associated with soil failures at the same percentile. To further compare differences, we calculate statistical significance under the hypothesis that distributions of strength are different using a Wilcoxon rank sum test (Table SI.2). Distributions of cohesion are different with statistical significance (p-values less than 0.01) between most landslide classifications. Differences in distributions of friction angles show modest

statistical differences in many cases. However, mean landslide thickness may serve as a more objective means of constraining strengths as classification of landslide mechanism (Figure SI.10) is subjective.”

R3.2 Second, the discussion can be improved by a clear explanation of ‘weathering’ and the limitation of the methods. The discussion regarding weathering is somewhat unclear to me. Does "weathering regimes" imply dichotomy boundaries between soil vs. bedrock, or in situ, continuous physical or chemical transition from bedrock, saprolite, to the soil. Those are different, and current analysis based on all landslides together may be hard to distinguish those two. Clarification on the influence 'weathering' – boundary, degree, or others, based on separated analysis (see the first comment) will be helpful. In addition, limitations of methods can be more clearly explained, and some quantitative comparison on strength estimates can be helpful. Although the authors used novel 3D methods for fitting failure planes, the slope stability analysis is simply based on the weight of the vertical column (Methods). This doesn't consider the effective stress from 3D topography or groundwater flow patterns, and the currently estimated cohesion in bedrock may influence by those other factors. In fact, the estimated cohesion in soil or bedrock are quite lower than the values observed by field measurement. Authors already discussion those potential influences qualitatively, but may consider providing a more quantitative comparison of results. It will be interesting to know whether the magnitudes of estimated cohesion differences (10 – 100 kPa) are similar in magnitudes of 3D effective stress variation from topography or groundwater flow.

The authors appreciate this comment – certainly, more clarity regarding the definition of weathering is justified. We have modified the following in the discussion (Lines 421-452) for clarity:

“Weathering governs the strength of rock and soil and may stem from a variety of processes (e.g., hydrologic, tectonic, and chemical weakening) – it has been recognized as an increasingly important factor in earth surface processes (Rempe and Dietrich 2014, Rempe and Dietrich 2018, Riebe et al. 2017), including landsliding (Li and Moon 2021). Recent studies have described the importance – and uncertainty – of the boundary between weathered and “fresh” bedrock stemming from wetting and drying (Rempe and Dietrich 2014), which is considered to be the boundary of the “critical zone.” We expect that while soil-bedrock boundary has often been treated as distinct, there is likely a gradient in the breakdown of bedrock with depth – localized and/or stochastic - that are intrinsically a control on the nonlinear relationship between strength and landslide thickness. The exchange of frictional and cohesive strength with mean landslide thickness may serve as a proxy for the bounds of this breakdown, particularly as extensive weathering is known to disaggregate bedrock (cohesive materials with tensile strength) to a more granular matrix (frictional materials). These weathering limits are particularly evident when comparing mean landslide thickness with the exchange of strength (Figure 3b). When compared with the cumulative distribution of landslide thickness, it is evident that much – but not all – of the exchange of proportional strength occurs in the range of mean thicknesses where landsliding is most frequent (Figure 4d); however, this trend may be muddled by the apparent censoring of smaller landslides, where strong controls on stability, such as vegetation, may persist. Approximately 33% of the observed exchange between frictional and cohesive strength occurs within depths that may reasonably be considered as soil (<1 m from Figure 3). The other portion of this strength exchange occurs within depths associated within saprolite, weathered bedrock and “fresh” bedrock. At the larger end of mean landslide thickness, this exchange in strength becomes less clear as landslide frequency decreases. However, the well-known decreasing frequency of very deep landslides and the trajectories of the strength exchange (i.e., the conjectured dashed lines in

Figure 3b) could indicate that at depth, cohesions associated with more intact rock (i.e., less weathered conditions) potentially reflect the transition or gradient to fresh bedrock that has undergone limited subaerial weathering (e.g., wetting and drying) and is governed by slower weathering processes (e.g., tectonic, chemical weathering). Because layering and interfaces between strata are often assumed to be a control on landsliding, transitions in mechanical properties with depth may be gradual. Weathering beneath the critical zone is much slower than in the near-surface environment (St. Clair et al. 2015), but is likely the most rapid along discontinuities. Sufficient weakening of rock mass in discontinuities may take orders of magnitude more time than rapid weathering and production of regolith (Larsen et al. 2010), resulting in the associated infrequency of large, deep bedrock landslides and sufficient loss of cohesive strength for yield (Figures SI.5, 4d). ”

This likely owes to a lack of clarity on the authors’ part, but we do account for localized topographic stresses relating to the thickness and basal dip slope of the soil column (described below); however, we do acknowledge that we do not account for far-field tectonic stresses, although this may be important. The following has been added to the “Limitations and Uncertainties” section:

“We acknowledge that while the three-dimensional slope stability analysis does account for localized topographic stresses within the landslide soil columns, it does not account for other stress controls, such as that of far-field tectonic stresses (e.g., Li and Moon 2021). More robust back-analyses may better incorporate how fractures and stresses stemming from tectonic strains might affect observed strength trends, as well as direct influences of discontinuities and stratigraphy.”

Regarding the slope stability analysis, it is a three-dimensional force equilibrium analysis that accounts for driving and resisting forces of all three-dimensional columns, evaluated both transverse and longitudinal to a direction of sliding – x and y , respectively (Hung et al. 1989, Bunn et al. 2020b). For force equilibrium, a factor of safety (FS) of unity is solved iteratively (i.e. equilibrium and incipient failure) for a suite of potential directions, where:

$$FS = \frac{\sum [c' A \cos \alpha_y + (N - uA) \tan \phi' \cos \alpha_y]}{\sum N \cos \gamma_z \tan \alpha_y}$$

$$N = \frac{W - \frac{c' A \sin \alpha_y}{FS} + \frac{u A \tan \phi' \sin \alpha_y}{FS}}{m_\alpha}$$

$$\cos \gamma_z = \sqrt{\frac{1}{\tan^2 \alpha_y + \tan^2 \alpha_x}}$$

$$m_\alpha = \cos \gamma_z \left(1 + \frac{\sin \alpha_y \tan \phi'}{FS \cos \gamma_z} \right)$$

Where A is the basal area of a column, α_y is the basal slope in the direction of sliding, α_x is the basal slope transverse to the direction of sliding, $\cos \gamma_z$ is the cosine of the maximum basal dip slope of a column, N is resultant normal force of a column, W is the weight of column, c' is

the effective cohesion, ϕ' is the effective friction angle, and u is the basal pore water pressure. As a landslide mass consists of many columns, convergence is achieved when summed forces in all columns for both x and y directions are zero. A maximization procedure is used to evaluate the back-analyzed shear strength that yields equilibrium for a range of potential directions of sliding, where the final back-analyzed shear strength for a given landslide is the largest determined for each direction. As back-analysis procedures for Mohr-Coulomb criteria are indeterminate to determine both cohesion and friction angle, we analyze deposits only for friction angle (ϕ') while setting $c'=0$. This reflects a mature slip surface where the bonds of cohesion have been broken. Finally, reconstructed landslide source areas use the same friction angle, but are analyzed using the same procedure to solve for cohesion. Topographic stresses (at least if we are understanding correctly) are being accounted for as each column has different weights, pore pressures, basal slopes, etc. In summary, a three-dimensional force equilibrium slope stability analysis that accounts for variation in topography, as described in Bunn et al. (2020) is used for the back-analysis, but it is performed for each landslide twice – both for deposit and landslide source – as a means of gleaning a first-order estimate of both cohesion and friction angle. Two parts of the “Methods” section have been modified for clarity:

*“**Back-Analysis and Stability.** Back-analyzed slope stability was evaluated using a three-dimensional adaptation of the three-dimensional force equilibrium method of columns approach (Hungry et al. 1989, Bunn et al. 2020b) which evaluates the sum of forces in the direction of sliding and transverse to the direction of sliding. We used a modification of this approach with a rotational correction procedure to account for the governing direction of sliding in context the asymmetry and complex shapes of natural landslides and the convergence of force equilibrium conditions while maximizing strength properties. We chose a column-based slope stability model for its straightforward implementation on a gridded digital elevation model, enabling efficient performance at a regional scale and direct incorporation of topographic variation of the digital elevation model.”*

*“**Landslide Stresses.** We use mean landslide thickness (D , m), mean reconstructed landslide inclination (θ , °), proportionally saturated thickness (m , dimensionless), mass density of soil/rock (ρ , 2,040 kg/m³), mass density of water (ρ_w , 1,000 kg/m³), gravity (g , 9.81 m/s²), and back-analyzed landslide effective cohesion (c' , kPa) and effective friction angle (ϕ' , °) to determine effective normal and shear strength for failure envelopes. As normal and shear stresses may vary throughout the landslide slip surface, for comparative purposes we simplify these stresses into scalar values using mean landslide thickness for comparison on a landscape scale using Mohr-Coulomb relationships based on mean basal stresses.”*

The simplifications, as shown in the methods are simply to be able to describe the overall characteristics of each landslide (mean thickness, mean shear stress, mean normal effective stress).

As to the influence of seepage-induced effective stress changes, we describe related revisions and discussion of limitations in a later comment.

Detailed comments are below.

R3.3 Line 38. Consider adding references to Clarke and Burbank 2010

Clarke, B. A., and D. W. Burbank (2010), Bedrock fracturing, threshold hillslopes, and limits to the magnitude of bedrock landslides, *Earth Planet. Sci. Lett.*, 297(3–4), 577–586, doi:10.1016/j.epsl.2010.07.011.

An excellent reference and suggestion – done.

R3.4 Line 43. Consider adding references to Hovius et al. (1997)

Hovius, Niels, Colin P. Stark, and Philip A. Allen. "Sediment flux from a mountain belt derived by landslide mapping." *Geology* 25.3 (1997): 231-234.

Again, an excellent suggestion. We have added this reference.

R3.5 Line 49. Clarke and Burbank's 2010 and 2011 papers

Done – many thanks.

R3.6 Line 53. three-dimensional geometry of "failure plane?"

Good suggestion – this could be clarified. Modified as:

“...that neglect the potentially strong influence of three-dimensional rupture surface...”

R3.7 Line 100 high-resolution topography means what resolution?

Apologies for any confusion. We have reworded this as follows:

“We use twelve inventories consisting of 7,300 landslides mapped with high-resolution topographic data (0.91 m bare earth lidar, Burns and Madin 2009)...”

We also note that we have used only 7,300 landslides in this revised analysis as a means of constraining the potential evacuation of landslides based on other reviewer comments.

R3.8 Lines 219 – 221. The range of measured lateral root cohesion in soil landslides in Oregon varies from 6.8 to 94.3 kPa (primarily measured < 1 m, Schmidt et al. 2001), which seem to be higher than the estimate presented here. Why do you think there are some differences?

This is an excellent question with several plausible reasons for significant differences. The seminal works (Waldron 1977, Wu et al. 1979, Schmidt et al. 2001) focused on lateral root cohesion are based on a variety of assumptions that may amplify the perceived stabilizing effects of root structures. One potential source of overestimating the role of root reinforcement *likely* owes to the assumption is that all roots fail at once. Another source of overestimated root strength is the assumption that all roots break at failure, as opposed to pullout of sufficiently strong but weakly anchored roots. However, numerous more recent studies (e.g. Pollen et al.

2005, Cohen et al. 2011, Cronkite-Ratliff et al. 2018, Giadrossich et al. 2019) have demonstrated that classical root reinforcement models may overestimate root lateral strength by 50-80%. Further, root lateral reinforcement is a strong function of root density, which decays with distance from the stem of a tree (Sakals and Sidle 2004, Roering et al. 2003). Studies using the “root cohesion” metric for stability (e.g. Wu et al. 1979) but evaluating the spatial distribution of roots (e.g. Sakals and Sidle 2004) suggest that root cohesion can regularly be in the range of ~1kPa, particularly in managed forestland (as is much of the area of these studies). Further, as shown in Table 2 and Table 4 (literature synthesis) of Schmidt et al. 2001), root cohesion values can regularly be in the range of ~2kPa (or less) at modest depths (~1m) *without accounting for* the spatial distribution of roots, pullout, or the progressive breakage of different size roots at failure. All of these elements suggest that the traditional definition of root cohesion may potentially be overestimated by 1-2 orders of magnitude and generally consistent with plausible ranges suggested in this study.

Maybe more relevant is that besides the depth scale associated with the reinforcing effects of roots, shallow landslides may actually nucleate from locations between trees or other significant vegetation where the stabilizing forces of roots is minimized.

We have added the following to the manuscript (lines 504-547) to expand on this point:

“Shallow landslides in soil and/or bedrock tend to be dominated by frictional strength with small magnitudes of cohesion, if present. This frictional strength is associated a relatively incoherent matrix of weathered material (Figure 4b). The elevated friction angles for small and/or steep landslides – and proportionally shallow depths - reflect a known sensitivity of friction angle to confining pressures (Figure 2a). With decreasing normal stress, many soils and fractured rock mass may have enhanced friction angles resulting from dilation and interlocking of grains within its matrix (Leps. 1970, Barton 2006, Townsend et al. 2020). Materials found at lower elevations on hillslopes may have diminished frictional strength from continued weathering, loosening during transport and higher contents of clay minerals (Moon et al. 2017) – this behavior is reflected by the observation that not all shallow landslides have high friction angles (Figure 2b). While many shallower landslides (e.g. $D < 1$ m) have limited cohesion, some cohesive strength is often still present. In many instances, cohesion accommodates over 50% of shear strength in some instances of small landslides ($D < 1$ m, Figure 3). However, this proportion of strength does not reflect large magnitudes of cohesion owing to small thicknesses and shear stress. For mean landslide thickness, a mean cohesion of 0.83, 4.93 and 25.2 kPa is associated with thicknesses of $D < 10^0$ m, $10^0 \leq D < 10^1$ m, and $D \geq 10^1$ m, respectively. At shallow landslide depths, these cohesive strengths may result from the influence of lateral root reinforcement. These values are at the lower-bound of “root cohesion” values observed in the Oregon Coast Range (Roering et al. 2003); however, it has been suggested that underlying assumptions in the methods that describe root cohesion values in in the Oregon Coast Range may overestimate the true strength from roots by as much as 75% (Pollen et al. 2005, Cohen et al. 2011, Giarossich et al. 2019), and potentially even more when considering the spatial heterogeneity of roots (Sakals and Sidle 2004). To test this, we performed back-analyses with a variety of root cohesion conditions reflecting forest management practices from Roering et al. (2004), shown in SI.15. As shown, the nonlinearity of the exchange in proportional resistance from mineral cohesion and friction is only amplified, and only modestly so for landslides with significant thickness. This suggests that the back-analyzed cohesions are potentially underestimates of root reinforcement. Consideration of potential overestimates in the equivalence of “root cohesion” suggest that the back-analyzed cohesion values for shallow soils are of reasonable magnitudes, at least as a lower bound. Studies have made the observation that when a threshold slope gradient is exceeded in soil-mantled landscapes, there is less frequent shallow landsliding owing to cohesive boundary forces stemming from vegetation and/or mineral

cohesion (Prancevic et al. 2020). It is suggested that beyond these threshold slope gradients, soil erosion in these environments may owe more to creep, which prevents requisite thickening of the soil mantle for landsliding. Other studies have described that friction angle may increase with depth, at least in the soil mantle (e.g. Lu and Godt 2008). These strengthening factors may be dominant in many landscapes, including the areas described herein. However, the landslide behaviors presented in this study do not refute those observations; rock mass may have very large friction angles, cohesion stemming from mineral behavior, roots, and capillary stresses may be significant at shallow depths. This study focuses on landslides that have occurred and do not necessarily represent complete landscape strength conditions; rather, the landslides used to construct relationships here may reflect the exception: localized strength conditions that result in landsliding, i.e. low root density or mineral cohesion, weakened and slickensided fractures in soft rock mass with partially intact rock bridges. The heavy censoring of small landslides from resolution and mapping limits in this study may indeed show a strong cohesive control at very shallow depths, still this analysis does provide some perspective as to the exchange of strength within a gradient of weathering of plausible soil and particularly bedrock depths.”

- Waldron, L. J. (1977). The shear resistance of root-permeated homogeneous and stratified soil. *Soil Science Society of America Journal*, 41(5), 843-849.
- Wu, T. H., McKinnell III, W. P., & Swanston, D. N. (1979). Strength of tree roots and landslides on Prince of Wales Island, Alaska. *Canadian Geotechnical Journal*, 16(1), 19-33.
- Schwarz, M., Cohen, D., & Or, D. (2012). Spatial characterization of root reinforcement at stand scale: theory and case study. *Geomorphology*, 171, 190-200.
- Pollen, N., & Simon, A. (2005). Estimating the mechanical effects of riparian vegetation on stream bank stability using a fiber bundle model. *Water Resources Research*, 41(7).
- Cohen, D., Schwarz, M., & Or, D. (2011). An analytical fiber bundle model for pullout mechanics of root bundles. *Journal of Geophysical Research: Earth Surface*, 116(F3).
- Cronkite-Ratcliff, C., Schmidt, K. M., & Wirion, C. (2018, December). Revisiting apparent root cohesion at the Coos Bay, Oregon landslide: A comparison of three models. In *AGU Fall Meeting Abstracts* (Vol. 2018, pp. NH21B-0820).
- Giadrossich, F., Cohen, D., Schwarz, M., Ganga, A., Marrosu, R., Pirastru, M., & Capra, G. F. (2019). Large roots dominate the contribution of trees to slope stability. *Earth Surface Processes and Landforms*, 44(8), 1602-1609.
- Sakals, M. E., & Sidle, R. C. (2004). A spatial and temporal model of root cohesion in forest soils. *Canadian Journal of Forest Research*, 34(4), 950-958.
- Roering, J. J., Schmidt, K. M., Stock, J. D., Dietrich, W. E., & Montgomery, D. R. (2003). Shallow landsliding, root reinforcement, and the spatial distribution of trees in the Oregon Coast Range. *Canadian Geotechnical Journal*, 40(2), 237-253.

R3.9 Line 225 – 226. This is slight confusion to me. To see whether this trend is due to the in-situ weathering process within bedrock, I think that it will be helpful to see the analysis done only using bedrock landslides, not from combined soil and bedrock landslides. Currently, the results are shown for both soil and bedrock landslides, so it is hard to tell whether the transition is due to variations of soil thickness, the difference in relative abundances of landslides, or the in-situ weathering differences.

A fair point – we thank the reviewer for the excellent suggestion. We now present these relationships for both soil and bedrock in the SI, as described in the response R3.1. We also acknowledge the potential censoring of smaller landslides (presumably soil). We acknowledge that it is almost convention to consider the interface between soil and bedrock as being a distinct transition, but it is plausible that this variation with depth may be more of a gradient owing to how weathering changes the form of rock mass (i.e. weathered rock may adhere more to

frictional behavior being comprised of cobbles/boulders; more intact rock relies on friction, but may have intact rock bridges or asperities that must be sheared to induce failure).

R3.10 Line 306. Are groundwater depths consistent with all surface? In reality, it may be deep under the ridge and shallow in the channel and 3D flows. Can this 3D effect be significant enough to change the trends?

This is a fair criticism – we fully agree that this could potentially affect the results. We would very much hope to evaluate the overall landscape groundwater regime, both in terms of seepage stresses and groundwater depth, to better reflect topographic controls. One would expect more depth to groundwater under ridgelines and less so in valleys and convergent terrain (i.e. “Dupuit-like” flow). Unfortunately, evaluating these controls would require data outside of what is realistically available to us with any level of confidence (e.g. regional-scale hydraulic conductivities, recharge rates, boundary conditions). Thus, in order to estimate the potential effects, we rely on sensitivity analyses relating to the saturated thickness of landslide columns (m), and possible groundwater depths. As shown in figures SI.7, SI.8, SI.11 and SI.12, these assumptions *do* influence the magnitude of changing friction angle and cohesion, but the proportional exchange in strength with landslide thickness to these assumptions is less sensitive. Thus, we completely agree that this would be an important step for future work, but unfortunately do not feel that we have a confident means of doing so. We have added some discussion throughout the manuscript related to this comment and limitation:

Lines 179-189: *“A larger uncertainty in this analysis owes to variability in groundwater flow-fields; thus, various groundwater conditions were evaluated as frictional strength is directly influenced by effective stress conditions associated with saturation (Figure SI.7, SI.8) and described briefly herein, but “half-saturated” conditions (i.e., the thickness of saturated soil and/or rock for the landslide is half of total landslide thickness on a column-by-column basis, $m=0.5$) and slope-parallel seepage are used for analyses described in the main text. We choose this condition as a conceptual “average” hydrological control as it is plausible to have perched groundwater in overlying soil or weathered rock stemming from intense rainfall, but saturated conditions are more likely to be persist deeper in bedrock (Salve et al. 2012). The sensitivity of varied groundwater conditions is explored in the Figures SI.11 and SI.12, but observed trends in comparative proportional frictional and cohesive hillslope strength are relatively insensitive to these conditions (e.g., <5% change in proportional strength).”*

Lines 255-294: *“The attenuation of frictional strength with normal stress (and indirectly, depth) may serve as a proxy for a variety of mechanisms: namely, (1) brittle rock may mobilize cohesive strength before fully mobilizing friction (Barton 2013), (2) deep discontinuities are prone to suffusion of clay minerals that exhibit low frictional strength (Tembe et al. 2010) and/or agglomeration of weaknesses (Milledge et al. 2014, Bellugi et al. 2021), and (3) large effective stresses suppress frictional behavior (Barton 2006, Renani and Martin 2018).*

Another plausible cause is a decrease in frictional resistance (but not friction angle) in the presence of saturation in bedrock but not in soil (e.g., frictional resistance is affected by pore water pressure, Eqn. 1). This sensitivity of our landscape-scale, nonlinear failure envelope is tested for both fixed groundwater depths and a range of saturation ratios (Figures SI.11 and SI.12), whereas most scenarios still maintain a nonlinear failure envelope of different magnitudes, although fixed, landscape-level groundwater depths of ≈ 5 -20 m demonstrate approximately linear behavior and diminished changes in friction angle with effective normal stress. Nonetheless, there is still an increase in cohesion with mean landslide thickness and

increasing normal stress. While persistent saturation may occur in bedrock at large depths, intense rainfall is more likely to result in more rapid (and destabilizing) pore pressure changes in the near surface (i.e., within soil, saprolite and to some extent, weathered rock; Salve et al. 2012). As contrasts in hydraulic conductivity at the transition to bedrock may result in a distinct aquifer (e.g., perched groundwater) within regolith that is disconnected from seasonal groundwater variations, saturation ratios may serve as a more plausible assumption in comparison to constant groundwater depth. As groundwater conditions in bedrock may be extremely heterogenous (e.g., Lovill et al. 2018), it is possible that variable groundwater conditions over the large area considered in this study may dampen the observed shifts in friction angle with depth, but nonetheless, we attribute the nonlinearity in landscape-level failure envelopes to a sensitivity of governing strength with landslide thickness as the observed nonlinearity in failure envelopes is consistent with observations of rock mass behavior (i.e., Hoek and Brown 1997, Barton 2006). An approximation of landscape-scale regional flow patterns would enable more confident trends in shear strength as controlled by groundwater flow. However, such models rely on (1) unknown regional-scale boundary conditions and parameters (e.g., hydraulic conductivity), and (2) and homogenous conditions – at this stage, there is insufficient data as to apply such conditions for our analysis. Thus, we evaluate the sensitivity and bounds of groundwater assumptions ranging from dry to fully saturated conditions, observing that the exchange of shear strength magnitude is sensitive to groundwater assumptions; however, the proportional exchange in strength with landslide thickness (described below) is less sensitive to groundwater assumptions. Seepage stresses may cause localized instability – for example, the near-surface (e.g., regolith and saprolite) is more sensitive than deeper flow regimes to changes in pore water pressures owing to rainfall infiltration and higher variations in seepage stresses are observed at shallow depths (Reid and Iverson, 1992). At greater depths, changes in effective stress fields induced by seepage are less pronounced and also exist at depths where cohesion may be appreciable, adding a nontrivial length scale to stability (i.e., seepage fields may be less important when cohesion plays a significant role in stability). At more shallow depths (the top few meters) where friction is slightly more dominant, these seepage fields and their influence on effective stresses may be more important, which supports the known sensitivity of shallower landslides to groundwater changes stemming from infiltration.”

R3.11 Line 310. Groundwater flow itself can generate differences in effective stress (Iverson and Reid, 1992, Reid and Iverson 1992). May check the magnitude of stress changes can induced by this effect.

As stated in the previous comment, the potential effects stemming from groundwater are a major uncertainty source. As shown in the groundwater sensitivity analysis, we simply do not have sufficient information to evaluate groundwater seepage fields and associated stresses on this scale. Thus, we are beholden to simplified assumptions about groundwater and investigating the sensitivity of model outputs to various assumptions of saturation, including pore pressure ratios ($m=0$ for dry, $m=0.5$ for half-saturated columns, and $m=1$ for completely saturated columns) and groundwater depths ($D_{GW}=1, 2.5, 10, 20\text{m}$). As shown in SI.7, SI.8, SI.11 and SI.12, there is a sensitivity of the magnitude of strength properties (both cohesion and primarily friction angle), but generally, the exchange of proportional shear strength is still observed. We have rerun our analyses to reflect slope-parallel seepage based on the relationship proposed by Reid (1997), where we account for fixed groundwater depths or saturation ratios (m) based on the inclination of cell itself. However, this still cannot account for the stress fields as shown by Iverson and Reid (1992a, 1992b), which demonstrated a proxy for the ratio of shear stress to normal under cohesionless conditions (please see figure below). Under cohesionless conditions, seepage fields will play a larger role in destabilizing conditions, particularly at more significant depths where cohesion, which is really a simplification for the tensile strength of rock, adds a non-trivial

length scale to stability (i.e. it would reduce the influence of pore pressures stemming from seepage fields). We have added the following to the limitations section:

“Seepage stresses may cause localized instability – for example, the near-surface (e.g., regolith and saprolite) is more sensitive than deeper flow regimes to changes in pore water pressures owing to rainfall infiltration and higher variations in seepage stresses are observed at shallow depths (Reid and Iverson, 1992). At greater depths, changes in effective stress fields induced by seepage are less pronounced and also exist at depths where cohesion may be appreciable, adding a nontrivial length scale to stability (i.e., seepage fields may be less important when cohesion plays a significant role in stability). At more shallow depths (the top few meters) where friction is slightly more dominant, these seepage fields and their influence on effective stresses may be more important, which supports the known sensitivity of shallower landslides to groundwater changes stemming from infiltration.”

FIG. 6. Elastic Effective Stress in a Homogeneous Hillslope: (a) Contours of Stress Ratio Φ in a Dry Hillslope (Shaded Regions Have Φ Greater Than 0.8); (b) Contours of Φ in a Saturated Hillslope and Normalized Seepage-Force Vectors; (c) Percentage of Change in Φ between Saturated and Dry Homogeneous Hillslopes

Effective stresses in a homogenous hillslope from Reid (1997). The seepage fields for saturated conditions are shown in Figure b with effective stress ratio contours shown. The variation and magnitude of percentage change in effective stress ratio are most concentrated at relatively shallow depths, which is where greater proportion of cohesive strength is expected.

Reid, M. E. (1997). Slope instability caused by small variations in hydraulic conductivity. *Journal of Geotechnical and Geoenvironmental Engineering*, 123(8), 717-725.

Iverson, R. M., and Reid, M. E. (1992), Gravity-driven groundwater flow and slope failure potential: 1. Elastic Effective-Stress Model, *Water Resour. Res.*, 28(3), 925– 938, doi:10.1029/91WR02694.

Reid, M. E., and Iverson, R. M. (1992), Gravity-driven groundwater flow and slope failure potential: 2. Effects of slope morphology, material properties, and hydraulic heterogeneity, *Water Resour. Res.*, 28(3), 939– 950, doi:10.1029/91WR02695.

R3.12 Line 313: "but more likely simply add noise to the observed trends." Why? Is it due to the stress magnitude being small?

Apologies – we agree that this sentence is confusing. Our intention is to say that groundwater conditions are very poorly-constrained at the scales described in this study, and thus consideration of these seemingly heterogenous hydrogeologic conditions may increase *or* decrease back-analyzed friction angles of individual landslides, but likely not change the overall observations, as shown for the variety of groundwater conditions in Figure SI.6. We have removed this part of the statement and modified as:

“While persistent saturation may occur in bedrock at large depths, intense rainfall is more likely to result in more rapid (and destabilizing) pore pressure changes in the near surface (i.e., within soil, saprolite and to some extent, weathered rock; Salve et al. 2012). As contrasts in hydraulic conductivity at the transition to bedrock may result in a distinct aquifer (e.g., perched groundwater) within regolith that is disconnected from seasonal groundwater variations, saturation ratios may serve as a more plausible assumption in comparison to constant groundwater depth. As groundwater conditions in bedrock may be extremely heterogenous (e.g., Lovill et al. 2018), it is possible that variable groundwater conditions over the large area considered in this study may dampen the observed shifts in friction angle with depth, but nonetheless, we attribute the nonlinearity in landscape-level failure envelopes to a sensitivity of governing strength with landslide thickness as the observed nonlinearity in failure envelopes is consistent with observations of rock mass behavior (i.e., Hoek and Brown 1997, Barton 2006). An approximation of landscape-scale regional flow patterns would enable more confident trends in shear strength as controlled by groundwater flow. However, such models rely on (1) unknown regional-scale boundary conditions and parameters (e.g., hydraulic conductivity), and (2) and homogenous conditions – at this stage, there is insufficient data as to apply such conditions for our analysis.”

R3.13 Line 353 delete large

Done – thank you for catching the typo.

R3.14 Line 419-420 The cohesion values (10 – 100 ka ranges) in the Roering et al. 2003 and Schmidt et al. 2001 seem quite higher than the presented.

We thank the reviewer for this suggestion. As described in Comment R3.8, the actual magnitudes of root “cohesion” are potentially significantly overestimated in these works owing to underlying assumptions of the breakage model (e.g. Pollen et al. 2005, Cohen et al. 2011). As shown above, this text has been modified as:

“In the absence of strengthening from vegetation, shallow landslides in soil and/or bedrock tend to be dominated by frictional strength with small magnitudes of cohesion, if present. This frictional strength is associated with a relatively incoherent matrix of weathered material (Figure 4b). The elevated friction angles for small and/or steep landslides – and proportionally shallow depths - reflect a known sensitivity of friction angle to confining pressures (Figure 2a). With decreasing normal stress, many soils and fractured rock mass may have enhanced friction angles resulting from dilation and interlocking of grains within its matrix (Leps. 1970, Barton 2006, Townsend et al. 2020). Materials found at lower elevations on hillslopes may have diminished frictional strength from continued weathering, loosening during transport and higher contents of clay minerals (Moon et al. 2017) – this behavior is reflected by the observation that

not all shallow landslides have high friction angles (Figure 2b). While many shallower landslides (e.g., $D < 1$ m) have limited cohesion, some cohesive strength is often still present. In many instances, cohesion accommodates over 50% of shear strength in some instances of small landslides ($D < 1$ m, Figure 3). However, this proportion of strength does not reflect large magnitudes of cohesion owing to small thicknesses and shear stresses. A mean cohesion of 0.83, 4.93 and 25.2 kPa is associated with thicknesses of $D < 1$ m, $1 \leq D < 10$ m, and $D \geq 10$ m, respectively. At shallow landslide depths, these cohesive strengths may result from the influence of lateral root reinforcement. As cohesion is back-analyzed uniformly over the entire rupture surface in this analysis, the influence of possible vegetation is evaluated with back-analyses with a variety of root cohesion at shallow depths (0.5 m) conditions reflecting forest management practices from Schmidt et al. (2001), shown in Figure SI.15. The presence of lateral root cohesion greatly reduces back-analyzed mineral cohesion at modest depths ($D < 1$ m), but much less so for moderate and only modestly so for landslides with significant thickness (i.e., bedrock landslides). This suggests that that equivalent cohesion values back-analyzed at larger depths or thicknesses (applied over the entire shear surface) are greater than equivalent root cohesion only applied to the near-surface. At shallow depths (i.e., $D < 1$ m), equivalent cohesion values are at the lower-bound of “root cohesion” values observed in the Oregon Coast Range (Schmidt et al., 2001) and may owe to the aforementioned censoring of smaller landslides; however, it has been suggested that underlying assumptions that describe root cohesion values may overestimate the true strength from roots by as much as 75% (Pollen et al. 2005, Cohen et al. 2011, Giarossich et al. 2019), and potentially even more when considering sparse vegetation and the spatial heterogeneity of roots (Sakals and Sidle 2004).”

As stated above, we have also performed back-analyses with lateral root cohesion values from Roering et al. (2003) applied to potential failure surfaces, as shown in Figure SI.15. As shown, this primarily diminishes back-analyzed cohesion at landslide sizes associated with soil, but much less so at sizes associated with bedrock landslides.

Figure SI.15. Relationships between mean landslide thickness and strength for different levels of lateral root cohesion (c_r) representative of different forest management conditions in the Oregon Coast Range (Roering et al. 2003). For management conditions of *No Roots*, *Clearcut*, *Industrial Forest*, and *Natural Forest*, root cohesions were 0, 3.35, 7.66, and 54.89 kPa, respectively. Lateral root cohesion was only applied to the area of the rupture surface with depths of 0.5m or less, consistent with the general shallow rooting depths described by Roering et al. (2003). These root cohesion values have been conjectured to possibly overestimate the mechanical reinforcement of roots (by as much as 75%) owing to their progressive breakage and pullout (e.g. Pollen et al. 2005, Cohen et al. 2011, Giarossich et al. 2019), thus reflect an upper bound for the proposed management conditions. As shown, stronger root cohesions amplify the proposed exchange between frictional and cohesive resistance with landslide thickness as much of the back-analyzed mineral cohesion is now supported by lateral root cohesion.

Once again, we thank the reviewer for their excellent comments and time dedicated towards helping us improve this work. It is sincerely appreciated.

REVIEWERS' COMMENTS

Reviewer #1 (Remarks to the Author):

Dear authors

congratulations with the revised manuscript. It was a pleasure reading and I found you method sound and the results convincing and again intriguing but with the second version also well explainable. The discussion on the effect of groundwater has been addressed and indeed, discussion on that topic will remain possible as you write as well but will not anything to add GW modelling for your analysis.

I have no remaining issues to discuss and am happy with this version. In short, I advice the publish this significant piece of work.

Kind regards

Thom Bogaard

Delft University of Technology

Reviewer #2 (Remarks to the Author):

The authors have addressed almost all my previous concerns and I recommend that the paper is published subject to some further minor revisions (see my attached report).

My major concern was “that the friction angle estimates rely on a method that generates spurious correlations with landslide inclination.” This is still a concern but now much less of a concern. The authors now provide clear and sufficient information in the paper to fully understand the method for friction angle estimation. They have also undertaken very extensive sensitivity testing, which is well designed, executed and reported. As a result, while I might disagree with some of the choices or interpretations and do still think that on balance the effects they find could be due to a methodological bias I think the paper should be published in a form very near to its current form.

Major comments

R2.1 RE title & abstract addressed, these are now excellent as is the match-up between abstract content and narrative and that in the main paper.

R2.2: The paper needs more detail on and a stronger justification for the method of estimating friction angle. My main concern with the paper is the first back calculation that is used to calculate friction angle under the assumption that deposits within the scar are in limiting equilibrium (along the preexisting failure mechanism) and are cohesionless. My concerns are: 1) that for many of the landslides I have seen (in person or in pictures) deposits are either patchy or absent in the scar and I don't know how you can apply your procedure in either of these cases; 2) a significant fraction of the pre-existing failure surface will now be the ground surface and shouldn't be included in the analysis (I guess it isn't but this wasn't clear to me from the paper);

These two concerns have been addressed in the revised manuscript.

3) the deposits that remain in the scar are not necessarily in limiting equilibrium, they can't be unstable but we have no indication of how stable they might be, as a result the first back calculation is a lower limit on friction angle but one that is also very strongly influenced by the scar topography (i.e. deposit angles do not reflect their friction angle but simply the angle of the scar within which they find themselves). This final point is the one I'm most concerned about because it would introduce a spurious correlation between landslide slope and friction angle. This would propagate through the known (and observed here) negative correlation between landslide slope and landslide depth to generate the relationship that you observe where friction declines with depth.

This is much more completely dealt with now and although I still differ from the authors I think the methods and results are sufficiently reported that my difference of opinion shouldn't prevent publication.

In the explanation of your back-calculation method, more information is needed on: 1) how you identify deposits within the scar and define the potential failure surfaces associated with those deposits; and 2) how many scars do not contain identifiable deposits in the scar, and their characteristics (e.g. size, depth, slope).

These two points are now well addressed in the paper. I do have a couple of minor comments below but these should be easily addressed with minor text modifications.

First, a brief clarification - we do not include the daylighted slip surface topography reflecting the failure surface in our backanalysis of the landslide deposits. We have attempted to make this clearer in the *Methods* section with this revised text:

“The approach was applied using the rupture surface determined from the thin-plate spline, and first applied to existing landslide topography (i.e., deposits at rest, Figure SI.4b) to determine the friction angle for each landslide in absence of cohesion (e.g., bonds of cohesion have been broken), excluding the headscarp. To obtain unique combinations of friction angle and cohesion, we then rerun the back analysis on all landslides using (1) the same rupture surface (Figure SI.4d), (2) the friction angle determined from the existing landslide deposit (Figure SI.4b) and (3) reconstructed landslide topography (Figure SI.4c) to determine the cohesion that yields equilibrium.”

In Figure SI.4 it is difficult to see whether the rupture surface follows the existing or reconstructed topographic surface. Could you move the rupture surface contours up as a layer so that they are visible over the existing topography contours. That would then make it clear that the rupture surface in the scar follows the existing rather than reconstructed topography (which is what I expect to be happening).

The reviewer raises several points that are justified, and to some level, insufficient information is available to “prove” the level of evacuation of landslides within the given inventory. For those landslides not associated *or associated* with known events, should landslides always be catastrophically evacuative? Isn’t it true that many landslides “creep” or move slowly downhill in a residual state at modest rates (if active)? ... Of course, the slow-moving characteristics of some landslides does not preclude catastrophic failure (as described nicely in Lacroix et al. 2020), but it does suggest that while the suggested bias towards runout may exist, the treatment of some proportion of landslides as non-evacuative is not farfetched.

Is it fair to use your approach for “creeping” landslides and earth flows? It would be useful to add a sentence or two on this in the conceptual setup for the approach. You take two snapshots in time (present and pre-failure) and apply cohesion to the later and no cohesion to the former. Is that appropriate for slow moving landslides? Perhaps it is, but you should be explicit about the implications of your approach and include a justification.

Considering this important potential bias, we reran all analyses to account for spatial overlap of deposits and source area, as well as other sensitivity analyses suggested by reviewers (e.g. root “cohesion”). ... The median proportional overlap of source landslide area and deposit landslide area to the total landslide area from the model is approximately 0.58. Approximately 90% of the landslides have proportional spatial overlap greater than 0.21.

This is useful, the vast majority of landslides have considerable overlap between scar and deposit. The new text addresses my concern around the influence of fully evacuated scars.

As to the level of equilibrium of the landslide deposit, we acknowledge this is a source of uncertainty. However, for landslides that are not completely evacuated and are (or were) active, wouldn’t the topography of the deposits indeed represent a state close to a metastable condition?

Yes, I agree if they ‘are’ or ‘were’ active in their current configuration but not if they were deposited and have not moved since. The sensitivity analysis applying a FS to the friction angle is useful in

demonstrating what properties of your relationships are more or less sensitive to the friction angle estimate. However, it doesn't account for potential correlation with slope inclination i.e. deposits on steeper slopes are more likely to be closer to limiting equilibrium than those on gentler slopes and would therefore need a slope dependent FS. I recognise though that it isn't feasible or perhaps even possible to do this. The analysis you've added is a useful if partial demonstration of the impact of your assumptions on your findings.

The two different FS approaches here are both useful but the new text in the paper is confusing because you discuss them in the same paragraph with little explanation of what your objective is for introducing them or what you have done to generate each, and without explaining the difference between the two.

“As rupture surfaces are a modeling output and ultimately, a first-order estimate of landslide geometry used to evaluate strength controls on instability, there is no direct means of directly isolating the observed trends in strength to ignore a bias from landslides with extensive evacuation and runout.”

I think you need to explain why this might introduce a bias. I agree that estimating friction strength from deposits likely introduces a bias due to the potential for enhanced landslide runout mobility (e.g. due to momentum). I can imagine an argument for why partial vs extensive evacuation of the scar might indicate landslides with lower runout mobility. But I think you need to lay that argument out in a sentence or two.

“The role of evacuation may control the level of stability of landslide deposits”

As above, I think you need a clearer explanation here of how you think partial evacuation (and specifically >25% source deposit overlap) can be used as an indicator of low mobility landslides (where dynamic forces play a smaller role in runout and deposition).

“That is, landslide deposits are treated to be a state of limiting equilibrium associated with continued activity. This is a significant and necessary assumption for the proposed inversions but has a basis in reality as landslide deposits (even relict deposits) are often prone to reactivation and continued instability (Temme et al. 2020).”

It seems almost guaranteed that this assumption of limiting equilibrium introduces a bias (because the dynamic forces that are neglected are destabilizing forces) and the major unknown is how large the bias is. What is the impact of the assumption? If runout mobility caused runout to stop under conditions within static stability rather than conditions of limiting equilibrium then the friction angle for these materials would be underestimated in a purely static analysis.

“We perform analyses to evaluate the level of stability of deposits considering back-analyzed friction angles from source areas using a factor of safety (FS), which for purely frictional materials reduces to $FS = \tan(\phi_{source}') / \tan(\phi_{deposit}')$. Under these conditions, the median FS is 1.26, almost equivalent to specified FS values of designed, engineered slopes ($FS \approx 1.25-1.3$, Figure SI.16).”

“It isn't clear what point you are trying to make here or why this would be an appropriate FS comparable to the FS values that would be generated for engineered slopes. It also seems to run counter to your argument that the deposits are in limiting equilibrium if you are here trying to associate them with engineered slopes with an FS 1.25-1.3.” I wrote these notes before reading the rebuttal and they suggest that it is difficult for a reader to understand what you mean here without

additional information. Once I'd read the rebuttal this became clear but I recommend you edit this text for clarity.

"We also investigate strength trends through application of a FS to deposit friction angles and subsequent recalculation of landslide source cohesion (Figure SI.16, Figure SI.17)."

What FS do you apply and why would you do this? As above, this became clear after reading the rebuttal but was not clear when reading the paper in isolation.

Of course, this comparison is reliant on modeled inputs/outputs. Nonetheless, these sensitivity analyses do demonstrate that stable deposits would diminish the observed strength trends, but not entirely. There is indirect evidence, albeit somewhat speculative, that many of the landslides analyzed herein are not fully evacuative owing to (1) the inventorying belying reliable mapping of a headscarp, toe and deposit for a given landslide feature, and (2) these morphological features that do not suggest full evacuation. Using screening of various features as described, the systematic exchange in strength controls with landslide size are still apparent

This is a very helpful paragraph and should be worked into your paper text

Numerous works for simplified slope stability analyses – both two- and three-dimensional - have demonstrated that both (1) a transition from friction-dominated to cohesion-dominated strength and (2) a transition from steep to more gentle slope inclinations will result in the deepening of the critical failure surface from shallow to deep-seated failure (e.g. Sun and Zhao 2013, Gao et al. 2013, Michalowski 2002, Taylor 1937). ... Such conditions could theoretically be driven by the increasing dominance of cohesive strength.

They could be driven by increasing dominance of cohesive strength but the dominance of cohesive strength could also be driven by the dependence of landslide area on landslide deposit slope, which reduces friction angle (in your method). But, we can agree to differ here.

The narrative has been revised substantially to reflect these concepts:

Figure 2. Inversions of shear strength ($m=0.5$ conditions) and associated stresses based on landslide thickness ($n=7331$ landslides). **(a)** Effective normal stress (σ') versus shear stress (τ') for landslide inventories colored by mean landslide thickness. Black lines represent moving median and 10% and 90% quantiles. Power law regressions are fit to the moving quantiles for a 1% moving window to represent plausible landslide failure envelopes. The insets show the nonlinearity of failure envelopes at low and high effective stresses, primarily owing to sensitivity of friction with normal stress. **(b)** Moving quantiles for both friction angle and cohesion are shown versus normal stress, fitted with power law functions.

What are the open circles in Fig 2b? This is not clear from the caption nor the methods text.

“The median friction angles associated at very low σ' are approximately $\sim 28^\circ$ and have small cohesion (≈ 0.5 kPa) for $m=0.5$ conditions.

It would be worth including some indication of variability around these medians e.g. +/- IQR or P10 & P90.

Friction may be enhanced in soils (i.e., regolith) under low confining stresses owing to dilation and hardening stemming from grain interlocking (Rowe 1962).

How large can this enhancement be?

For the presented conditions, the 90th percentile, median and 10th percentile failure envelopes generally follow representative nonlinear strength envelopes (Barton 2006) of fractured rock, jointed rock, and rock with clay-filled discontinuities, respectively.

What does this mean? Are you claiming agreement with the detail of strength envelopes in Barton? What aspects of their strength envelopes do you reproduce?

R2.3: For friction (L390-91): ‘Approximately half of the observed exchange between frictional and cohesive strength occurs within depths considered to be soil (<1.9 m)’. From the Figures it looks as though estimated friction angle decreases by about (40 – 30 deg) as mean depth increases from 0.1 to 1 m, which is a huge reduction in friction angle with depth. The only studies I have come across that discuss depth dependent friction angle in soil actually suggest the opposite relationship. ... You find that cohesion increases with depth in the soil (e.g. L417-8, Fig 3b). Again, I do not know of any research suggesting that cohesion should increase with depth in the soil, though I do know of several papers that suggest the opposite, usually on the basis that both root density and root diameter distributions decrease with depth (e.g. Bischetti et al., 2007, Montgomery et al., 2009). ... You do try to provide some support from the literature for the idea that friction declines with depth but when I followed up these citations it wasn’t obvious that their findings supported your claims.

We have completely reworked relevant text to highlight that (1) cohesion is still present in the soil column, and (2) there is a suite of research that emphasizes the relationship between normal stresses decreasing friction angles.

Most importantly, the observed strength patterns shown in this study do not refute that soil may have large real or apparent cohesion or that bedrock may have large friction angles; rather, these relationships reflect the localized, weakened conditions that yield failure at a large scale.

This is a key modifier to your conclusions that should be reflected in your headline/abstract statements and in your suggestions for how the findings can be applied in future.

Further, there is published research that conceptually supports decreasing friction angles with increasing normal stress, and we have attempted to work this into the study. For these analyses, we observe an increase in proportional friction of approximately 6%, while approximately 12% occurs at greater depths (Figure 3). The moving median friction angle does decrease by about 4° for $m=0.5$ conditions for depths associated with less than a meter of soil (and drops of 13° and 2° for 90th and 10th percentile moving windows), and continues to diminish with larger normal stresses/thicknesses. These diminished friction angles could be associated with dilative behavior in soil (Rowe 1962), but even moreso in rock mass or rockfill (fractured or partially-fractured, Leps 1970, Barton 2016), and the decreased friction angles continue for normal stress ranges that span orders of magnitude.

This seems more reasonable, I also note the shape of the relationship at shallow depths is much changed and you place limits on the depth range in which you consider your results to be reliable. This is all useful. In the top meter though, you are surely almost always dealing with failures in soil? Does Rowe provide support for the 13 degree drop in friction angle you see for 90th percentile moving window?

“The apparent plateau in this proportional exchange in cohesive strength results from the bounding values of our moving window and the large uncertainty reflects relatively few landslides having mean thicknesses beneath 0.5 m or beyond 10 m.”

This statement is useful and important

“Nonetheless, we anticipate that these trends are likely to persist, perhaps in a nonlinear fashion (similar relationships using binned data are shown in Figure SI.13).”

I’m not convinced that you can claim this. On what basis do you expect the trends to persist?

“With decreasing normal stress, many soils and fractured rock mass may have enhanced friction angles resulting from dilation and interlocking of grains within its matrix (Leps. 1970, Barton 2006, Townsend et al. 2020). Materials found at lower elevations on hillslopes may have diminished frictional strength from continued weathering, loosening during transport and higher contents of clay minerals (Moon et al. 2017) – this behavior is reflected by the observation that not all shallow landslides have high friction angles (Figure 2b).

But these friction angles are still surprisingly low, you should comment on the level of agreement between your friction angle estimates and those for soils from testing.

This suggests that that equivalent cohesion values back-analyzed at larger depths or thicknesses (applied over the entire shear surface) are greater than equivalent root cohesion only applied to the near-surface.

Typo in repeat of “that”.

This study focuses on landslides that have occurred and do not necessarily represent complete landscape strength conditions; rather, the landslides used to construct relationships here may reflect the exception: localized strength conditions that result in landsliding (i.e., low root density or mineral cohesion, weakened and slickensided fractures in soft rock mass with partially intact rock bridges – all of which result in landslide erosion). Still, this analysis does provide perspective as to the exchange of strength within a gradient of weathering at depths reflective of soil and particularly bedrock.”

This is a very important modifier to your findings. If you do not think your findings are reflective of landscapes in general but only of landslide producing locations in landscapes and you have good reason to believe that the behavior in these regions may be considerably different to complete

landscape conditions this is something that should appear in your headlines and conclusions. For example, your final sentence: “Thus, we posit that landscape-scale models that rely on strength of materials may account for the nonlinearity in strength criteria, and thus serve to link landslide thickness and weathering.” (L514-515) **I don’t think you can claim this if the behavior you identify is specific to landslide generating locations rather than the landscape as a whole.**

Specific comments

R2.43 L481: What length scale is curvature calculated over? I guess uncertainty in this length parameter has little impact on your findings but it should probably be recognised.

The length scale is the same as that of the rupture surface fitting technique (0.91m-3.0m). This has been revised as:

“...under a simple plate constraint that adheres to the diffusion equation ($\partial z^2/\partial^2 x + \partial z^2/\partial^2 y = 0$), computed using a finite difference solver over the landslide DEM grid.”

You should probably tell the reader the length scale over which this is calculated.

New Specific Comments - these are largely due to new text associated with groundwater assumptions

L263-267: This is an excellent piece of analysis but is not clearly reported at present. I also think that a constant depth of ground water is as plausible as a ratio of the failure surface so the results from this analysis are as likely to be the ‘true’ results as those you present in the main paper.

L269: Why should more rapid pore pressure changes be the destabilizing changes? Previous work on earth flows in Oregon has shown that longer term seasonal rainfall and its associated gradual changes to pore pressure can prompt acceleration of these deeper landslides and earth flows.

L270-273: this argument is problematic because most of your landslides are not shallow landslides. If the water tables are perched they would not influence these deeper failure surfaces, which would instead respond to the deeper aquifer. I don’t think this argument holds up as a justification of the half saturated assumption. This is to me simply an arbitrary choice in the absence of any information to direct you one way or another. You are right to test the sensitivity to the assumption but I don’t think you can make any statements about one groundwater scenario (and thus one set of results) being more plausible than any other.

L283-285: “however, the proportional exchange in strength with landslide thickness (described below) is less sensitive to groundwater assumptions” I don’t see these graphs in the SI. I am surprised that the proportional exchange is insensitive given the quite large changes in friction angle behavior between scenarios. You should certainly note in the text that the relationship between friction angle and effective normal stress reverses under approximately half of your fixed depth groundwater scenarios (e.g. GW depth = 2 and 20 m causes a switch for median and 10th percentile; depth = 5 and 10 m causes a switch for median and both percentiles; depth = 1 m causes a switch for only 10th percentile). These are quite large changes and they result from scenarios that are as plausible as the m=0.5 that you focus on.

Reviewer #3 (Remarks to the Author):

This paper by Alberti et al., "Inversions of Landslide Strength as a Proxy for Subsurface Weathering" applied 3D slope stability methods over 7,300 mapped landslides in Western Oregon, USA, to quantify unique contributions from cohesion and friction strength. They find that the proportion of contribution from cohesion increases with landslide thickness, implying the increased cohesive strengths associated with less weathered rock at depths.

I have reviewed the earlier version of the paper (R3), and I applaud the authors' efforts in making this revision. The authors addressed my concerns well by including additional analysis (e.g., separating bedrock and soil landslides, groundwater effects) and providing more in-depth discussion, including the limitations of the study. I also have reviewed other reviewers' comments, which are constructive and detailed. All reviewers pointed out that the assumptions of groundwater effect need clarification or modification. The authors now included detailed explanations and additional sensitivity analysis using varying saturation ratios or depth to groundwater (Figure SI7, 11, 12). They explained that they could not examine the variable groundwater condition (e.g., regional-scale ground flow patterns) due to the limitation of data/information. However, they included a discussion on the justification and limitation of their approach.

R2 made good suggestions and critical comments on the methods to estimate the friction angle from deposits and possible errors in the estimate. The methods of friction angle measured from the slope of deposits may not be accurate because the deposits may not be at rest, may be evacuated fully, and be part of landslide scar topography. This bias may provide a negative correlation between landslide slope and depth, resulting in observed trends of friction declines with landslide depths. I agree that this is an important point and needs clarification. The authors made efforts to clarify their methods, add additional analysis using landslides with varying thresholds of deposit/source overlaps, and examine the level of equilibrium of landslide deposits. The authors showed that the main results (e.g., proportional frictional and cohesive hillslope strength change with landslide thickness) are consistent regardless of those assumptions.

The authors made a significant improvement to the paper, including performing multiple sensitivity tests, clarifying their methods, and discussing their assumptions, choices, and limitations. The finding of this paper will be an interest of the broader audience in Nature Communications, and I recommend the publication of the paper.

Below are minor comments

Line 143. Inclination means slope inclination?

Line 162. 7300 or 7330? I notice that the numbers changing as 7300, 7330, and 7331. Please check and revise them if needed.

Line 179. A separate paragraph starting from "A larger uncertainty "

Figure 2. $n = 7331$? Not 7330?

Missing color bar for landslide thickness

Not sure what you mean "1% moving window". Is it moving window to include 1% of dataset (~73 landslides)?

Line 241 in Figure SI.14?

Line 294. Add reference (maybe, Montgomery et al 2009)?

Lines 573. Please a, b, c, d in figure SI.4.

Line 578. 7300 or 7730?

General: It is hard to see some of the SI figures because they are too small. Please make it bigger so we can see the difference better (Figure SI. 10)

Guide:

Reviewer comments are in blue.

Author responses are in green.

REVIEWER 1

Dear authors

congratulations with the revised manuscript. It was a pleasure reading and I found you method sound and the results convincing and again intriguing but with the second version also well explainable. The discussion on the effect of groundwater has been addressed and indeed, discussion on that topic will remain possible as you write as well but will not anything to add GW modelling for your analysis.

I have no remaining issues to discuss and am happy with this version. In short, I advice the publish this significant piece of work.

Thom Bogaard

Delft University of Technology

The authors very much appreciate the feedback and constructive comments on this study – it has certainly helped us improve this work.

REVIEWER 2

The authors have addressed almost all my previous concerns and I recommend that the paper is published subject to some further minor revisions (see my attached report).

Once again, we appreciate the reviewer's diligence, open-mindedness and constructive comments on this study. Without a doubt, it has helped us make this work more concise, focused, and meaningful.

My major concern was *“that the friction angle estimates rely on a method that generates spurious correlations with landslide inclination.”* **This is still a concern but now much less of a concern. The authors now provide clear and sufficient information in the paper to fully understand the method for friction angle estimation. They have also undertaken very extensive sensitivity testing, which is well designed, executed and reported. As a result, while I might disagree with some of the choices or interpretations and do still think that on balance the effects they find could be due to a methodological bias I think the paper should be published in a form very near to its current form.**

We appreciate the reviewer's open-mindedness and also recognize that this is largely an under-constrained problem. Nonetheless, we believe that approaches of this nature could be further refined to provide further insight towards weathering and landslide processes on a landscape scale.

Major comments

R2.1 RE title & abstract **addressed, these are now excellent as is the match-up between abstract content and narrative and that in the main paper.**

We thank the reviewer for their excellent suggestions.

R2.2: The paper needs more detail on and a stronger justification for the method of estimating friction angle. My main concern with the paper is the first back calculation that is used to calculate friction angle under the assumption that deposits within the scar are in limiting equilibrium (along the preexisting failure mechanism) and are cohesionless. My concerns are:

1) that for many of the landslides I have seen (in person or in pictures) deposits are either patchy or absent in the scar and I don't know how you can apply your procedure in either of these cases;

2) a significant fraction of the pre-existing failure surface will now be the ground surface and shouldn't be included in the analysis (I guess it isn't but this wasn't clear to me from the paper); **These two concerns have been addressed in the revised manuscript.**

Many thanks.

3) the deposits that remain in the scar are not necessarily in limiting equilibrium, they can't be unstable but we have no indication of how stable they might be, as a

result the first back calculation is a lower limit on friction angle but one that is also very strongly influenced by the scar topography (i.e. deposit angles do not reflect their friction angle but simply the angle of the scar within which they find themselves). This final point is the one I'm most concerned about because it would introduce a spurious correlation between landslide slope and friction angle. This would propagate through the known (and observed here) negative correlation between landslide slope and landslide depth to generate the relationship that you observe where friction declines with depth.

This is much more completely dealt with now and although I still differ from the authors I think the methods and results are sufficiently reported that my difference of opinion shouldn't prevent publication.

Again, we appreciate the comment, and recognize that inversions of this kind are largely unconstrained. We hope that the uncertainty of inversion techniques of this nature might be improved in the future with analyses accounting for landslide runout dynamics and more focused on disturbance events where triggering is better constrained.

In the explanation of your back-calculation method, more information is needed on: 1) how you identify deposits within the scar and define the potential failure surfaces associated with those deposits; and 2) how many scars do not contain identifiable deposits in the scar, and their characteristics (e.g. size, depth, slope).

These two points are now well addressed in the paper. I do have a couple of minor comments below but these should be easily addressed with minor text modifications.

We thank the reviewer again for the excellent comments – we have performed all of the requested modifications.

Minor Comments:

First, a brief clarification - we do not include the daylighted slip surface topography reflecting the failure surface in our back analysis of the landslide deposits. We have attempted to make this clearer in the *Methods* section with this revised text:

“The approach was applied using the rupture surface determined from the thin-plate spline, and first applied to existing landslide topography (i.e., deposits at rest, Figure SI.4b) to determine the friction angle for each landslide in absence of cohesion (e.g., bonds of cohesion have been broken), excluding the headscarp. To obtain unique combinations of friction angle and cohesion, we then rerun the back analysis on all landslides using (1) the same rupture surface (Figure SI.4d), (2) the friction angle determined from the existing landslide deposit (Figure SI.4b) and (3) reconstructed landslide topography (Figure SI.4c) to determine the cohesion that yields equilibrium.”

In Figure SI.4 it is difficult to see whether the rupture surface follows the existing or reconstructed topographic surface. Could you move the rupture surface contours up as a layer so that they are visible over the existing

topography contours. That would then make it clear that the rupture surface in the scar follows the existing rather than reconstructed topography (which is what I expect to be happening).

Indeed, the reviewer's assessment is correct. The contour layers have been adjusted accordingly.

The reviewer raises several points that are justified, and to some level, insufficient information is available to “prove” the level of evacuation of landslides within the given inventory. For those landslides not associated *or associated* with known events, should landslides always be catastrophically evacuative? Isn't it true that many landslides “creep” or move slowly downhill in a residual state at modest rates (if active)? ... Of course, the slow-moving characteristics of some landslides does not preclude catastrophic failure (as described nicely in Lacroix et al. 2020), but it does suggest that while the suggested bias towards runout may exist, the treatment of some proportion of landslides as non-evacuative is not farfetched. **Is it fair to use your approach for “creeping” landslides and earth flows? It would be useful to add a sentence or two on this in the conceptual setup for the approach. You take two snapshots in time (present and pre-failure) and apply cohesion to the later and no cohesion to the former. Is that appropriate for slow moving landslides? Perhaps it is, but you should be explicit about the implications of your approach and include a justification.**

A fair point. We have added the following text:

“For some slow-moving failures that tend to be in a persistent residual state and generally be devoid of cohesive strength over long timescales, back-analyzed cohesion estimates may be overestimated. However, the genesis of these failures (which is approximated from this analysis) may have had significant cohesion from reconsolidation or lithification despite their post-failure creeping behavior.”

Considering this important potential bias, we reran all analyses to account for spatial overlap of deposits and source area, as well as other sensitivity analyses suggested by reviewers (e.g. root “cohesion”). ... The median proportional overlap of source landslide area and deposit landslide area to the total landslide area from the model is approximately 0.58. Approximately 90% of the landslides have proportional spatial overlap greater than 0.21.

This is useful, the vast majority of landslides have considerable overlap between scar and deposit. The new text addresses my concern around the influence of fully evacuated scars.

Many thanks.

As to the level of equilibrium of the landslide deposit, we acknowledge this is a source of uncertainty. However, for landslides that are not completely evacuated and are (or were)

active, wouldn't the topography of the deposits indeed represent a state close to a metastable condition?

Yes, I agree if they 'are' or 'were' active in their current configuration but not if they were deposited and have not moved since. The sensitivity analysis applying a FS to the friction angle is useful in demonstrating what properties of your relationships are more or less sensitive to the friction angle estimate. However, it doesn't account for potential correlation with slope inclination i.e. deposits on steeper slopes are more likely to be closer to limiting equilibrium than those on gentler slopes and would therefore need a slope dependent FS. I recognise though that it isn't feasible or perhaps even possible to do this. The analysis you've added is a useful if partial demonstration of the impact of your assumptions on your findings.

The two different FS approaches here are both useful but the new text in the paper is confusing because you discuss them in the same paragraph with little explanation of what your objective is for introducing them or what you have done to generate each, and without explaining the difference between the two.

We appreciate the comment – it has been addressed below.

“As rupture surfaces are a modeling output and ultimately, a first-order estimate of landslide geometry used to evaluate strength controls on instability, there is no direct means of directly isolating the observed trends in strength to ignore a bias from landslides with extensive evacuation and runout.”

I think you need to explain why this might introduce a bias. I agree that estimating friction strength from deposits likely introduces a bias due to the potential for enhanced landslide runout mobility (e.g. due to momentum). I can imagine an argument for why partial vs extensive evacuation of the scar might indicate landslides with lower runout mobility. But I think you need to lay that argument out in a sentence or two.

An excellent suggestion. We have revised the introductory paragraph sentences as follows:

“As rupture surfaces are a modeling output and ultimately, a first-order estimate of landslide geometry used to evaluate strength controls on instability, there is no direct means of directly isolating the observed trends in strength to ignore a bias from landslides with extensive evacuation and runout. Extensive runout or complete evacuation of scar could result in deposition that is largely a function of momentum. In such an instance, deposits could be more stable than a state of limiting equilibrium.”

“The role of evacuation may control the level of stability of landslide deposits”

As above, I think you need a clearer explanation here of how you think partial evacuation (and specifically >25% source deposit overlap) can be used as an indicator of low mobility landslides (where dynamic forces play a smaller role in runout and deposition).

We have modified this sentence as:

“We select a 25% overlap as a lower bound on non-evacuative landslides (approximately the 10th percentile for overlap) as it suggests that the area of deposition is not overly large in comparison to scar area (a consequence of runout).”

“That is, landslide deposits are treated to be a state of limiting equilibrium associated with continued activity. This is a significant and necessary assumption for the proposed inversions but has a basis in reality as landslide deposits (even relict deposits) are often prone to reactivation and continued instability (Temme et al. 2020).”

It seems almost guaranteed that this assumption of limiting equilibrium introduces a bias (because the dynamic forces that are neglected are destabilizing forces) and the major unknown is how large the bias is. What is the impact of the assumption? If runout mobility caused runout to stop under conditions within static stability rather than conditions of limiting equilibrium then the friction angle for these materials would be underestimated in a purely static analysis.

We agree, and this is the basis for exploring this bias through application of FS, as described in the text outlined in the next few comments.

“We perform analyses to evaluate the level of stability of deposits considering back-analyzed friction angles from source areas using a factor of safety (FS), which for purely frictional materials reduces to FS

= $\tan(\phi_{source}) / \tan(\phi_{deposit})$. Under these conditions, the median FS is 1.26, almost equivalent to specified FS values of designed, engineered slopes (FS ≈ 1.25-1.3, Figure SI.16).”

“It isn’t clear what point you are trying to make here or why this would be an appropriate FS comparable to the FS values that would be generated for engineered slopes. It also seems to run counter to your argument that the deposits are in limiting equilibrium if you are here trying to associate them with engineered slopes with an FS 1.25-1.3.” I wrote these notes before reading the rebuttal and they suggest that it is difficult for a reader to understand what you mean here without additional information. Once I’d read the rebuttal this became clear but I recommend you edit this text for clarity.

Many thanks – please see the next comment.

“We also investigate strength trends through application of a FS to deposit friction angles and subsequent recalculation of landslide source cohesion (Figure SI.16, Figure SI.17).”

What FS do you apply and why would you do this? As above, this became clear after reading the rebuttal but was not clear when reading the paper in isolation.

This is an excellent comment – we neglected to provide any sort of introductory statement as to why we were performing the FS analysis. We have added a sentence to provide context to these sensitivity analyses. Many thanks.

“To explore the potential bias of deposits not being in a state of limiting equilibrium from extensive evacuation or runout, we perform analyses to evaluate the level of stability of deposits considering back-analyzed friction angles from source areas using a factor of safety (FS), which for purely frictional materials reduces to $FS = \tan(\phi'_{source}) / \tan(\phi'_{deposit})$.”

Of course, this comparison is reliant on modeled inputs/outputs. Nonetheless, these sensitivity analyses do demonstrate that stable deposits would diminish the observed strength trends, but not entirely. There is indirect evidence, albeit somewhat speculative, that many of the landslides analyzed herein are not fully evacuative owing to (1) the inventorying belying reliable mapping of a headscarp, toe and deposit for a given landslide feature, and (2) these morphological features that do not suggest full evacuation. Using screening of various features as described, the systematic exchange in strength controls with landslide size are still apparent

This is a very helpful paragraph and should be worked into your paper text

We appreciate the comment – we feel that this text has been worked into the manuscript on Lines 345-366:

“As rupture surfaces are a modeling output and ultimately, a first-order estimate of landslide geometry used to evaluate strength controls on instability, there is no direct means of directly isolating the observed trends in strength to ignore a bias from landslides with extensive evacuation and runout. We consider simple criteria by which landslides are evacuated, including the proportional overlapping deposit and modeled source area versus total landslide area (deposit and headscarp). We observe that the landslide inventory herein has a median areal overlap of 58% between source and deposit, which suggests that full evacuation for the landslides used in this analysis may not exert a strong bias associated with long runout landslides. We select a 25% overlap as a lower bound on non-evacuative landslides (approximately the 10th percentile for overlap). The role of evacuation may control the level of stability of landslide deposits, which are used to determine the friction angle in this study. That is, landslide deposits are treated to be a state of limiting equilibrium associated with continued activity. This is a significant and necessary assumption for the proposed inversions but has a basis in reality as landslide deposits (even relict deposits) are often prone to reactivation and continued instability (Temme et al.

2020). We perform analyses to evaluate the level of stability of deposits considering back-analyzed friction angles from source areas using a factor of safety (FS), which for purely frictional materials reduces to $FS = \tan(\phi'_{source}) / \tan(\phi'_{deposit})$. Under these conditions, the median FS is 1.26, almost equivalent to specified FS values of designed, engineered slopes ($FS \approx 1.25-1.3$, Figure SI.16). We also investigate strength trends through application of a FS to deposit friction angles and subsequent recalculation of landslide source cohesion (Figure SI.16, Figure SI.17). We observe similar exchanges in proportional friction and cohesion, although the exchanges are offset. For a given landslide, the inversion of strength treats the landslide mass as a homogenous material, which is a simplification with potential significance in layered materials or interfaces between strata (e.g., soil-bedrock boundaries).”

Numerous works for simplified slope stability analyses – both two- and three-dimensional - have demonstrated that both (1) a transition from friction-dominated to cohesion-dominated strength and (2) a transition from steep to more gentle slope inclinations will result in the deepening of the critical failure surface from shallow to deep-seated failure (e.g. Sun and Zhao 2013, Gao et al. 2013, Michalowski 2002, Taylor 1937). ... Such conditions could theoretically be driven by the increasing dominance of cohesive strength.

They could be driven by increasing dominance of cohesive strength but the dominance of cohesive strength could also be driven by the dependence of landslide area on landslide deposit slope, which reduces friction angle (in your method). But, we can agree to differ here.

Much appreciated – there are uncertainties at such scales, but some research has shown that cohesion does have a control on potential landslide depth (e.g. Klar et al. 2011).

Klar, A., Aharonov, E., Kalderon-Asael, B., & Katz, O. (2011). Analytical and observational relations between landslide volume and surface area. *Journal of Geophysical Research: Earth Surface*, 116(F2).

The narrative has been revised substantially to reflect these concepts:

Figure 2. Inversions of shear strength ($m=0.5$ conditions) and associated stresses based on landslide thickness ($n=7331$ landslides). (a) Effective normal stress (σ') versus shear stress (τ) for landslide inventories colored by mean landslide thickness. Black lines represent moving median and 10% and 90% quantiles. Power law regressions are fit to the moving quantiles for a 1% moving window to represent plausible landslide failure envelopes. The insets show the nonlinearity of failure envelopes at low and high effective stresses, primarily owing to sensitivity of friction with normal stress. (b) Moving quantiles for both friction angle and cohesion are shown versus normal stress, fitted with power law functions.

What are the open circles in Fig 2b? This is not clear from the caption nor the methods text.

Agreed - this should be made clearer. We have added the following to the caption:

“...**(b)** Moving quantiles for both friction angle and cohesion are shown versus normal stress, fitted with power law functions to median strength values for each percentile of binned effective normal stress, shown with solid and open circles.”

“The median friction angles associated at very low σ' are approximately $\sim 28^\circ$ and have small cohesion (≈ 0.5 kPa) for $m=0.5$ conditions.

It would be worth including some indication of variability around these medians e.g. +/- IQR or P10 & P90.

We have added the following text:

“At low normal stresses, 10th and 90th percentiles for friction angle and cohesion are 14° and 42° , and 0.1 and 2 kPa, respectively.”

Friction may be enhanced in soils (i.e., regolith) under low confining stresses owing to dilation and hardening stemming from grain interlocking (Rowe 1962).

How large can this enhancement be?

The degree of enhancement is largely dependent on the material. Rowe shows these enhancements in the order of degrees, while work on rockfill has shown changes well over 15 degrees (e.g. Duncan et al. 2014, Leps 1970).

Duncan, J. M., Wright, S. G., & Brandon, T. L. (2014). *Soil strength and slope stability*. John Wiley & Sons.

For the presented conditions, the 90th percentile, median and 10th percentile failure envelopes generally follow representative nonlinear strength envelopes (Barton 2006) of fractured rock, jointed rock, and rock with clay-filled discontinuities, respectively.

What does this mean? Are you claiming agreement with the detail of strength envelopes in Barton? What aspects of their strength envelopes do you reproduce?

A fair point – this statement is likely a distraction and has been removed.

R2.3: For friction (L390-91): ‘Approximately half of the observed exchange between frictional and cohesive strength occurs within depths considered to be soil (<1.9 m)’. From the Figures it looks as though estimated friction angle decreases by about (40 – 30 deg) as mean depth increases from 0.1 to 1 m, which is a huge reduction in friction angle with depth. The only studies I have come across that discuss depth dependent friction angle in soil actually suggest the opposite relationship. ... You find that cohesion increases with depth in the soil (e.g. L417-8, Fig 3b). Again, I do not know of any research suggesting that cohesion should increase with depth in the soil, though I do know of several papers that suggest the opposite, usually on the basis that both root density and root diameter distributions decrease with depth (e.g. Bischetti et al., 2007, Montgomery et al., 2009). ... You do try to provide some support from the literature for the idea that friction declines with depth but when I followed up these citations it wasn’t obvious that their findings supported your claims.

We have completely reworked relevant text to highlight that (1) cohesion is still present in the soil column, and (2) there is a suite of research that emphasizes the relationship between normal stresses decreasing friction angles.

Most importantly, the observed strength patterns shown in this study do not refute that soil may have large real or apparent cohesion or that bedrock may have large friction angles; rather, these relationships reflect the localized, weakened conditions that yield failure at a large scale.

This is a key modifier to your conclusions that should be reflected in your headline/abstract statements and in your suggestions for how the findings can be applied in future.

Many thanks. We have modified this in several locations, including the abstract as follows:

“Our results demonstrate that the failure envelope that relates shear strength and normal stress in landslide terrain is nonlinear.”

Further, in our prior revisions, we have some discussion related to this specific uncertainty:

“The presented trends do not contradict that mineral and root cohesion can be large in the soil mantle, and that friction may increase with landslide thickness; rather, they reflect a first-order set of trends that reflect the conditions that *represent failure* and the associated localized weaknesses that yield failure.”

Further, there is published research that conceptually supports decreasing friction angles with increasing normal stress, and we have attempted to work this into the study. For these analyses, we observe an increase in proportional friction of approximately 6%, while approximately 12% occurs at greater depths (Figure 3). The moving median friction angle does decrease by about 4° for $m=0.5$ conditions for depths associated with less than a meter of soil (and drops of 13° and 2° for 90th and 10th percentile moving windows), and continues to diminish with larger normal stresses/thicknesses. These diminished friction angles could be associated with dilative behavior in soil (Rowe 1962), but even more so in rock mass or rockfill (fractured or partially-fractured, Leps 1970, Barton 2016), and the decreased friction angles continue for normal stress ranges that span orders of magnitude.

This seems more reasonable, I also note the shape of the relationship at shallow depths is much changed and you place limits on the depth range in which you consider your results to be reliable. This is all useful. In the top meter though, you are surely almost always dealing with failures in soil? Does Rowe provide support for the 13 degree drop in friction angle you see for 90th percentile moving window?

Much appreciated. While Rowe does not show a 13 degree drop in friction angle, other relationships (e.g. Leps 1970 and similar work) show similar decreases (or larger drops), as alluded to in the text. We have also added the following line for relating to shallow landslide depths:

“Further, there are still uncertainties as to whether landslides with small mean thicknesses are purely in soil.”

“The apparent plateau in this proportional exchange in cohesive strength results from the bounding values of our moving window and the large uncertainty reflects relatively few landslides having mean thicknesses beneath 0.5 m or beyond 10 m.”

This statement is useful and important

Agreed.

“Nonetheless, we anticipate that these trends are likely to persist, perhaps in a nonlinear fashion (similar relationships using binned data are shown in Figure SI.13).”

I’m not convinced that you can claim this. On what basis do you expect the trends to persist?

We have modified this as:

“The apparent plateau in this proportional exchange in cohesive strength results from the bounding values of our moving window and the large uncertainty reflects relatively few landslides having mean thicknesses beneath 0.5 m or beyond 10 m, but is less defined using binned data (shown in Figure SI.13).”

“With decreasing normal stress, many soils and fractured rock mass may have enhanced friction angles resulting from dilation and interlocking of grains within its matrix (Leps. 1970, Barton 2006, Townsend et al. 2020). Materials found at lower elevations on hillslopes may have diminished frictional strength from continued weathering, loosening during transport and higher contents of clay minerals (Moon et al. 2017) – this behavior is reflected by the observation that not all shallow landslides have high friction angles (Figure 2b).

But these friction angles are still surprisingly low, you should comment on the level of agreement between your friction angle estimates and those for soils from testing.

We have revised this text as:

“While observed friction angles are rather low, materials found at lower elevations on hillslopes may have diminished frictional strength from continued weathering, loosening during transport and higher contents of clay minerals (Moon et al. 2017) – this behavior is reflected by the observation that not all shallow landslides have high friction angles (Figure 2b).”

This suggests that that equivalent cohesion values back-analyzed at larger depths or thicknesses (applied over the entire shear surface) are greater than equivalent root cohesion only applied to the near-surface.

Typo in repeat of “that”.

Fixed – many thanks.

This study focuses on landslides that have occurred and do not necessarily represent complete landscape strength conditions; rather, the landslides used to construct relationships here may reflect the exception: localized strength conditions that result in landsliding (i.e., low root density or mineral cohesion, weakened and slickensided fractures in soft rock mass with partially intact rock bridges – all of which result in landslide erosion). Still, this analysis does provide

perspective as to the exchange of strength within a gradient of weathering at depths reflective of soil and particularly bedrock.”

This is a very important modifier to your findings. If you do not think your findings are reflective of landscapes in general but only of landslide producing locations in landscapes and you have good reason to believe that the behavior in these regions may be considerably different to complete landscape conditions this is something that should appear in your headlines and conclusions. For example, your final sentence: “Thus, we posit that landscape-scale models that rely on strength of materials may account for the nonlinearity in strength criteria, and thus serve to link landslide thickness and weathering.” (L514-515) I don’t think you can claim this if the behavior you identify is specific to landslide generating locations rather than the landscape as a whole.

This is fair. We have revised the text in several locations to be more accurate; for example:

“Thus, we posit that models that rely on strength of subsurface materials should account for the nonlinearity in strength criteria, and that analysis of landslide terrain may serve as a proxy for landslide thickness and weathering.”

Specific comments

R2.43 L481: What length scale is curvature calculated over? I guess uncertainty in this length parameter has little impact on your findings but it should probably be recognised.

The length scale is the same as that of the rupture surface fitting technique (0.91m-3.0m). This has been revised as:

“...under a simple plate constraint that adheres to the diffusion equation ($\partial z^2/\partial^2x + \partial z^2/\partial^2y = 0$), computed using a finite difference solver over the landslide DEM grid.”

You should probably tell the reader the length scale over which this is calculated.

Done. Many thanks.

New Specific Comments - these are largely due to new text associated with groundwater assumptions

L263-267: This is an excellent piece of analysis but is not clearly reported at present. I also think that a constant depth of ground water is as plausible as a ratio of the failure surface so the results from this analysis are as likely to be the ‘true’ results as those you present in the main paper.

Much appreciated – we have modified the text to highlight these uncertainties:

“As contrasts in hydraulic conductivity at the transition to bedrock may result in a distinct aquifer (e.g., perched groundwater) within regolith that is disconnected from seasonal groundwater variations, we assume that saturation ratios are a more representative assumption in comparison to constant groundwater depth. Nonetheless, as groundwater conditions in bedrock may be extremely heterogenous (e.g., Lovill et al. 2018) and most of the analyzed landslides are deep-seated, it is possible that variable groundwater conditions over the large area considered in this study may dampen the observed shifts in friction angle with depth.”

L269: Why should more rapid pore pressure changes be the destabilizing changes?

Previous work on earth flows in Oregon has shown that longer term seasonal rainfall and its associated gradual changes to pore pressure can prompt acceleration of these deeper landslides and earth flows.

This sensitivity to infiltration and infiltration-driven porewater pressures are most acute in the near surface (i.e. the soil mantle). We have revised this as:

“While persistent saturation may occur in bedrock at large depths, intense rainfall is more likely to result in perched groundwater in the near surface”

L270-273: this argument is problematic because most of your landslides are not shallow landslides. If the water tables are perched they would not influence these deeper failure surfaces, which would instead respond to the deeper aquifer. I don't think this argument holds up as a justification of the half saturated assumption. This is to me simply an arbitrary choice in the absence of any information to direct you one way or another. You are right to test the sensitivity to the assumption but I don't think you can make any statements about one groundwater scenario (and thus one set of results) being more plausible than any other.

Ultimately, landscape-scale analyses of groundwater flow (also an indeterminate problem) would be necessary to describe the triggering groundwater conditions accurately. However, perched water is not necessarily constrained to only shallow failures, but may exist at the weathering front in in bedrock (deeper slides, Lebedeva and Brantley 2020). Thus, we appreciate this point, but still believe that the assumptions are reasonable but should be stated as an uncertainty. We have modified the text as follows:

“As contrasts in hydraulic conductivity at the transition to bedrock may result in a distinct aquifer within regolith or the weathering front of bedrock (e.g., perched groundwater, Lebedeva and Brantley 2020) that is disconnected from seasonal groundwater variations, we assume that saturation ratios are a representative assumption in comparison to constant groundwater depth. Nonetheless, as groundwater conditions in bedrock may be extremely heterogenous (e.g., Lovill et al. 2018) and most of the analyzed landslides are deep-seated, it is possible that variable groundwater conditions over the large area considered in this study may dampen the observed shifts in friction angle with depth.”

Lebedeva, M. I., & Brantley, S. L. (2020). Relating the depth of the water table to the depth of weathering. *Earth Surface Processes and Landforms*, 45(9), 2167-2178.

L283-285: *“however, the proportional exchange in strength with landslide thickness (described below) is less sensitive to groundwater assumptions”* **I don’t see these graphs in the SI. I am surprised that the proportional exchange is insensitive given the quite large changes in friction angle behavior between scenarios. You should certainly note in the text that the relationship between friction angle and effective normal stress reverses under approximately half of your fixed depth groundwater scenarios (e.g. GW depth = 2 and 20 m causes a switch for median and 10th percentile; depth = 5 and 10 m causes a switch for median and both percentiles; depth = 1 m causes a switch for only 10th percentile). These are quite large changes and they result from scenarios that are as plausible as the $m=0.5$ that you focus on.**

We appreciate the comment, but the proportional exchanges are shown in Figures SI.11 and SI.12, as noted in the text. Further, the sensitivity of friction angles to fixed groundwater depths are also noted in the text:

“This sensitivity of our failure envelope is tested for both fixed groundwater depths and a range of saturation ratios (Figures SI.11 and SI.12), whereas most scenarios still maintain a nonlinear failure envelope of different magnitudes, although fixed, landscape-level groundwater depths of ≈ 5 -20 m demonstrate approximately linear behavior and diminished changes in friction angle with effective normal stress.”

REVIEWER 3

This paper by Alberti et al., “Inversions of Landslide Strength as a Proxy for Subsurface Weathering” applied 3D slope stability methods over 7,300 mapped landslides in Western Oregon, USA, to quantify unique contributions from cohesion and friction strength. They find that the proportion of contribution from cohesion increases with landslide thickness, implying the increased cohesive strengths associated with less weathered rock at depths.

I have reviewed the earlier version of the paper (R3), and I applaud the authors’ efforts in making this revision. The authors addressed my concerns well by including additional analysis (e.g., separating bedrock and soil landslides, groundwater effects) and providing more in-depth discussion, including the limitations of the study. I also have reviewed other reviewers’ comments, which are constructive and detailed. All reviewers pointed out that the assumptions of groundwater effect need clarification or modification. The authors now included detailed explanations and additional sensitivity analysis using varying saturation ratios or depth to groundwater (Figure SI7, 11, 12). They explained that they could not examine the variable groundwater condition (e.g., regional-scale ground flow patterns) due to the limitation of data/information. However, they included a discussion on the justification and limitation of their approach.

R2 made good suggestions and critical comments on the methods to estimate the friction angle from deposits and possible errors in the estimate. The methods of friction angle measured from the slope of deposits may not be accurate because the deposits may not be at rest, may be evacuated fully, and be part of landslide scar topography. This bias may provide a negative correlation between landslide slope and depth, resulting in observed trends of friction declines with landslide depths. I agree that this is an important point and needs clarification. The authors made efforts to clarify their methods, add additional analysis using landslides with varying thresholds of deposit/source overlaps, and examine the level of equilibrium of landslide deposits. The authors showed that the main results (e.g., proportional frictional and cohesive hillslope strength change with landslide thickness) are consistent regardless of those assumptions.

The authors made a significant improvement to the paper, including performing multiple sensitivity tests, clarifying their methods, and discussing their assumptions, choices, and limitations. The finding of this paper will be an interest of the broader audience in Nature Communications, and I recommend the publication of the paper.

The authors sincerely appreciate the constructive feedback of the reviewer. It has greatly helped us improve this work and we are very grateful for the reviewer sharing their time and expertise.

Below are minor comments

Line 143. Inclination means slope inclination?

Indeed – this has been corrected.

Line 162. 7300 or 7330? I notice that the numbers changing as 7300, 7330, and 7331. Please check and revise them if needed.

Apologies – it should be 7330. Corrected.

Line 179. A separate paragraph starting from “A larger uncertainty ”

A good suggestion – many thanks.

Figure 2. n = 7331? Not 7330?

This has been fixed – many thanks.

Missing color bar for landslide thickness

Done – many thanks.

Not sure what you mean “1% moving window”. Is it moving window to include 1% of dataset (~ 73 landslides?)

Apologies – this should be made clear. Revised as:

“...values for each percentile of binned effective normal stress, shown with solid and open circles.”

Line 241 in Figure SI.14?

The reviewer is correct – many thanks!

Line 294. Add reference (maybe, Montgomery et al 2009)?

An excellent reference and suggestion – it has been added.

Lines 573. Please a, b, c, d in figure SI.4.

Done – excellent suggestion.

Line 578. 7300 or 7730?

Fixed – thanks!

**General: It is hard to see some of the SI figures because they are too small.
Please make it bigger so we can see the difference better (Figure SI. 10)**

Yes, we apologize for this, however it is difficult to make these larger in an efficient manner. Nevertheless, we have attempted to make the figures that are particularly small a bit larger and more legible.